# Linear Causal Bandits: Unknown Graph and Soft Interventions

**Zirui Yan**
Rensselaer Polytechnic Institute
yanz11@rpi.edu

**Ali Tajer**
Rensselaer Polytechnic Institute
tajer@ecse.rpi.edu

## Abstract

Designing causal bandit algorithms depends on two central categories of assumptions: (i) the extent of information about the underlying causal graphs and (ii) the extent of information about interventional statistical models. There have been extensive recent advances in dispensing with assumptions on either category. These include assuming known graphs but unknown interventional distributions, and the converse setting of assuming unknown graphs but access to restrictive hard/$\mathrm{do}$ interventions, which removes the stochasticity and ancestral dependencies. Nevertheless, the problem in its general form, i.e., *unknown* graph and *unknown* stochastic intervention models, remains open. This paper addresses this problem and establishes that in a graph with $N$ nodes, maximum in-degree $d$ and maximum causal path length $L$, after $T$ interaction rounds the regret upper bound scales as $\tilde{\mathcal{O}}((cd)^{L-\frac{1}{2}}\sqrt{T} + d + RN)$ where $c > 1$ is a constant and $R$ is a measure of intervention power. A universal minimax lower bound is also established, which scales as $\Omega(d^{L-\frac{3}{2}}\sqrt{T})$. Importantly, the graph size $N$ has a diminishing effect on the regret as $T$ grows. These bounds have matching behavior in $T$, exponential dependence on $L$, and polynomial dependence on $d$ (with the gap $d$ ). On the algorithmic aspect, the paper presents a novel way of designing a computationally efficient CB algorithm, addressing a challenge that the existing CB algorithms using soft interventions face.

## 1 Motivation & Overview

Causal bandits (CBs) provide a formal framework for the sequential design of experiments over a network of agents with *causal* interactions. The objective of CBs is to identify an experiment that maximizes a notion of utility over the causal network. CB settings are specified by three elements: (i) a causal graphical model that defines the topological ordering of the causal variables and their probabilistic relationships; (ii) a set of structural equation models (SEMs) that specify their cause-effect dependencies among the variables; and (iii) *intervention* models that specify the extent of exogenous variations imposed on the causal interactions by an external force. By interpreting the set of interventions as the set of *arms* and the decision quality (utility) as the rewards, CBs' objective is to maximize the cumulative reward by strategically selecting the sequence of interventions that optimize a notion of cumulative utility [1, 2]. CBs have a broad range of applications [3–5].

The recent advances in CBs can be grouped based on three assumption dimensions: (i) the assumptions on the extent of information available about the causal graph structure, (ii) the assumptions about pre- and post-intervention statistical models, and (iii) the nature of the SEMs. There have been significant advances in understanding CBs when the *causal graphs are known*. The most relevant studies include those that started with analyzing $do$ interventions as the simplest form of interventions [1, 2, 6, 7] and have progressed toward the more complex stochastic interventions (hard and soft). These studies have investigated various linear and non-linear SEMs. Specifically, the studies in [8–13] assume that the pre- and post-interventional statistical distributions are known. The study in [14] further advances the results by assuming that these distributions are known only partially, and finally, the studies in [15–18] entirely dispense with all the assumptions about the interventions' statistical models.

In sharp contrast, when the causal graph is *unknown*, the problem is far less investigated and open in its general form. The lack of topology knowledge makes the problem substantially more complex, since the graph's topology captures all the conditional independence information about the random variables in the system. Hence, when the graph is known, it is unnecessary to learn the conditional independencies; however, when it is unknown, all the conditional independencies should be learned.

The notable results under unknown graphs include [19], which assumes that all interventional distributions are *fully known*. Dispensing with the assumption of interventional distributions with a focus on $do$ is investigated in [19–25]. $do$ interventions are generally more amenable to tractable analysis because of the analytical simplifications they enable. A $do$ intervention at a node sets the random value of that node to a pre-specified fixed value. This results in (i) removing all the causal dependence of that node on its ancestors and (ii) removing the randomness of the data generated by that node. In sharp contrast, *stochastic soft interventions* are the more general and realistic forms of interventions that retain all the ancestral dependencies and the probabilistic nature of the model. A soft intervention, specifically, changes the pre-intervention statistical models to other distinct models.

Table 1: Cumulative instance-independent regrets for linear CBs.

| Algorithm | Regret bound | Intervention | Scalable | Lower bound |
|---|---|---|---|---|
| Unknown Graph | | | | |
| CN-UCB[20] | $\tilde{\mathcal{O}}\left(\sqrt{KT} + Kd\right)$ | do | ✓ | $\Omega(\sqrt{NKT})$[1] |
| GA-LCB (This paper) (Theorem 2) | $\tilde{\mathcal{O}}\left((\kappa d)^{L-\frac{1}{2}}\sqrt{T} + d + RN\right)$ | soft | ✓ | - |
| Known Graph | | | | |
| C-UCB[6] | $\tilde{\mathcal{O}}\left(\sqrt{K^d T}\right)$ | do | ✓ | - |
| LinSEM-UCB[15] | $\tilde{\mathcal{O}}\left(d^{2L-\frac{3}{2}}\sqrt{NT}\right)$ | soft | ✗ | $\Omega\left(d^{L-\frac{3}{2}}\sqrt{T}\right)$ (Theorem 3) |
| GCB-UCB[18] | $\tilde{\mathcal{O}}\left(d^{2L-1}\sqrt{T}\right)$ | soft | ✗ | |
| GA-LCB (this paper) (Corollary 1) | $\tilde{\mathcal{O}}\left(d^{L-\frac{1}{2}}\sqrt{T}\right)$ | soft | ✓ | |

[1] under weaker assumptions.

**Contributions.** We establish upper and lower regret bounds for the CB problem under *unknown* graphs, *unknown* pre- and post-intervention statistical models, and *soft* stochastic interventions. Furthermore, we also provide a novel approach to algorithm design and regret analysis. The main assumptions and contributions of the paper are as follows.

- **Topology:** We assume to know only the number of the nodes on the graph and the in-degree of the causal graph.[1]
- **Statistical model:** We assume that all pre- and post-intervention statistical models are unknown.

- **Regret bounds:** We characterize almost matching upper and lower bounds on the regret as a function of the time horizon and graph topology parameters. Specifically, we show the achievable regret of $\tilde{\mathcal{O}}\left((\kappa d)^{L-\frac{1}{2}}\sqrt{T} + d + N\right)$ where $\kappa > 1$ is a constant, $N$ is the number of graph nodes, $d$ is the maximum in-degree of the graph, $L$ is the maximum causal depth, and $T$ is the time horizon. We also establish the minimax regret lower bound of $\Omega(d^{L-\frac{3}{2}}\sqrt{T})$.
- **Tightness of the bounds:** The dependence of the achievable regret on $N$ is diminishing as $T$ grows. Therefore, the mismatch of the achievable and the minimax regrets is on the order of $d$ and a constant $\kappa^{L-\frac{1}{2}}$.
- **Special cases:** Our general regret bounds provide improvements for the known special cases. In particular, we show that when the graph becomes known, our achievable regret becomes $\tilde{\mathcal{O}}(d^{L-\frac{1}{2}}\sqrt{T})$, which is tighter than the best known results $\tilde{\mathcal{O}}(d^{2L-1}\sqrt{T})$ [18].
- **Scalabe algorithm:** We introduce a novel CB algorithm under soft interventions. We note that the existing algorithms for soft algorithms are based on the upper confidence bound (UCB) principle, and they are generally not scalable due to the intractable optimization problem pertinent to maximizing the UCBs. In our algorithm, we circumvent his issue, resulting in a scalable algorithm as the graph size grows.

---

[1] We note that assuming only an upper bound on the in-degree is sufficient to achieve the same regret bound.

**Notations.** For a positive integer $N \in \mathbb{N}$, we define $[N] \triangleq \{1, \cdots, N\}$. Random variables and their realizations are represented by upper- and lower-case letters, respectively. Matrices and vectors are represented by bold upper- and lower-case letters. The $i$-th element of vector $\mathbf{x}$ is denoted by $x_i$. The $i$-th column vector of matrix $\mathbf{A}$ is denoted by $[\mathbf{A}]_i$ and $[\mathbf{A}]_{i,j}$ denotes the $(i, j)$ element of $\mathbf{A}$. $\mathbf{A}^n$ denotes the $n$-th power of matrix $\mathbf{A}$ for $n \in \mathbb{N}$. $\mathbb{1}$ denotes the indicator function. Sets and events are denoted by calligraphic letters. The cardinality of set $\mathcal{A}$ is denoted by $|\mathcal{A}|$. For any set $\mathcal{S} \subseteq [N]$, $\mathbb{1}\{\mathcal{S}\} \in \{0, 1\}^N$ is specified such that its elements at the coordinates included in $\mathcal{S}$ are set to 1, and the rest are 0. For a vector $\mathbf{x}$ and positive semidefinite matrix $\mathbf{A}$, we define $\|\mathbf{x}\|_{\mathbf{A}} = \sqrt{\mathbf{x}^\top \mathbf{A} \mathbf{x}}$ as the weighted $\ell_2$ norm. The $\ell_1$-norm and $\ell_2$-norm of vector $\mathbf{x} \in \mathbb{R}^d$ are denoted by $\|\mathbf{x}\|_1$ and $\|\mathbf{x}\|_2$, respectively. The notation $\tilde{\mathcal{O}}$ is an order notation that ignores constant and poly-logarithmic factors.

## 2 Causal Bandit Problem Setup

**Causal graph.** Consider an *unknown* directed acyclic graph (DAG) $\mathcal{G} = \{\mathcal{V}, \mathcal{E}\}$ in which $\mathcal{V} = [N]$ is the set of nodes and $\mathcal{E}$ is the set of directed edges. A directed edge from node $i$ to node $j$ is denoted by the ordered tuple $(i, j)$. The set of parents of node $i \in \mathcal{V}$ is denoted by $\mathsf{pa}(i)$. Similarly, the sets of ancestors and descendants of node $i$ are denoted by $\mathsf{an}(i)$ and $\mathsf{de}(i)$, respectively. We define the *causal depth* of node $i$, denoted by $L_i$, as the length of the longest directed causal path that ends at node $i \in [N]$. According, we denote the maximum causal depth of the graph by $L \triangleq \max_{i \in [N]} L_i$ and denote the *maximum in-degree* of the graph by $d \triangleq \{\max_{i \in [N]} |\mathsf{pa}(i)|\}$.

**Data model.** DAG $\mathcal{G}$ represents a Bayesian network, in which we denote the causal random variable associated with node $i \in \mathcal{V}$ by $X_i$. Accordingly, we define the random vector $X \triangleq (X_1, \ldots, X_N)^\top$. For any $A \subseteq \mathcal{V}$, $X_A$ denotes the vector formed by $\{X_i : i \in A\}$. The extents of the cause-effect relationships among the causal variables $X$ are specified by the following linear SEMs:

$$X = \mathbf{B}^\top X + \epsilon, \tag{1}$$

where $\mathbf{B} \in \mathbb{R}^{N \times N}$ is the edge weights matrix and $\epsilon \triangleq (\epsilon_1, \ldots, \epsilon_N)^\top$ denotes the model noises. It is noteworthy that the element $[\mathbf{B}]_{j,i}$ is non-zero if and only if $j \in \mathsf{pa}(i)$. We denote the conditional distribution of $X_i$ given its parents by $\mathbb{P}(X_i \mid X_{\mathsf{pa}(i)})$.

**Soft stochastic interventions.** We use *soft* interventions as the most general form of intervention. A *soft* intervention on node $i$ retains the ancestral dependence of $X_i$ on $X_{\mathsf{pa}(i)}$ and its probabilistic nature. Specifically, a soft intervention on node $i$ changes the conditional probability $\mathbb{P}(X_i \mid X_{\mathsf{pa}(i)})$ to a distinct one denoted by $\mathbb{Q}(X_i \mid X_{\mathsf{pa}(i)})$. In a linear SEM, the impact of a soft intervention on node $i$ can be abstracted by a change in the vector $[\mathbf{B}]_i$. We denote the post-intervention vector by $[\mathbf{B}^*]_i$. We refer to $\mathbf{B}$ and $\mathbf{B}^*$ as the observational and interventional weights matrices, respectively. We allow multiple nodes to be intervened simultaneously and denote the space of possible interventions by $\mathcal{A} \triangleq 2^{[N]}$. For a specific intervention $\mathbf{a} \in \mathcal{A}$, we define $\mathbf{B_a}$ as the post-intervention weight matrix specified by

$$[\mathbf{B_a}]_i \triangleq \mathbb{1}\{i \in \mathbf{a}\}[\mathbf{B}^*]_i + \mathbb{1}\{i \notin \mathbf{a}\}[\mathbf{B}]_i. \tag{2}$$

**Causal bandit – problem statement.** In causal bandit, a learner performs a sequence of interventions to optimize a reward measure. Each unique set of interventions $\mathbf{a} \in \mathcal{A}$ is represented by an arm. Following the CB's convention, we designate node $N$ as the reward node and its associated value $X_N$ as the reward variable. We denote the post-intervention probability measure of $X$ induced by intervention $\mathbf{a}$ by $\mathbb{P_a}$, and the associated expectation by $\mathbb{E_a}$. Subsequently, we denote the expected value of variable $X_i$ under intervention $\mathbf{a} \in \mathcal{A}$ by

$$\mu_{i,\mathbf{a}} \triangleq \mathbb{E_a}[X_i], \tag{3}$$

Accordingly, we denote the optimal intervention by $\mathbf{a}^*$, which is specified by

$$\mathbf{a}^* \triangleq \arg\max_{\mathbf{a} \in \mathcal{A}} \mu_{N,\mathbf{a}}. \tag{4}$$

The sequence of interventions over time is denoted by $\{\mathbf{a}(t) \in \mathcal{A} : t \in \mathbb{N}\}$. Upon intervention $\mathbf{a}(t)$ in round $t$, the learner observes $X(t) \triangleq (X_1(t), \ldots, X_N(t))^\top$ and collects the reward $X_N(t)$. The learner's objective is to minimize the regret that it incurs with respect to an omniscient oracle that has

access to the best intervention $\mathbf{a}^*$. Hence, the average regret incurred at time $t$ is $r(t) = \mu_{\mathbf{a}^*} - \mu_{\mathbf{a}}$. Accordingly, the average cumulative regret over horizon $T$ is given by

$$\mathbb{E}[\mathcal{R}(T)] \triangleq T\mu_{\mathbf{a}^*} - \sum_{t=1}^{T} \mu_{\mathbf{a}(t)} . \tag{5}$$

We list the set of assumptions that we make about the SEMs.

**Assumption 1** (Unknown graph). *We assume that the skeleton and orientation of the edges in graph $\mathcal{G}$ are* unknown. *We assume the number of nodes $N$ and degree $d$ are known.*

This assumption is in contrast to all the existing studies on soft intervention [15–18].

**Assumption 2** (Unknown conditional distributions). *We assume that all observational and interventional conditional distributions $\{\mathbb{P}(X_i \mid X_{\mathsf{pa}(i)}) : i \in [N]\}$ and $\{\mathbb{Q}(X_i \mid X_{\mathsf{pa}(i)}) : i \in [N]\}$ are* unknown.

**Assumption 3** (Weight matrices). *The interventional and observational matrices $\mathbf{B}$ and $\mathbf{B}^*$ are* unknown. *We assume that the range of weight matrix elements is* known, *i.e., there exists a known $m_{\mathbf{B}} \in \mathbb{R}_+$ such that $|[\mathbf{B}]_{j,i}| \leq m_{\mathbf{B}}$ and $|[\mathbf{B}^*]_{j,i}| \leq m_{\mathbf{B}}$ for all $i, j \in [N]$.*

**Assumption 4** (Noise model). *We assume that the noise statistical model is* unknown. *The expected noise value $\boldsymbol{\nu} \triangleq \mathbb{E}[\epsilon]$ is known. We assume the noise terms are independent and bounded, i.e., there exists $m_\epsilon \in \mathbb{R}_+$ such that $|\epsilon_i(t)| \leq m_\epsilon$ for all $i \in [N]$ and $t \in [T]$.*

We note the assumption that knowing $d$ can be replaced by knowing an upper bound, and the requirement of expected noise value $\boldsymbol{\nu}$ can be removed by the re-arrange method in [15]. To ensure the bounded stability of the system, the bounded noise and weights assumptions are widely used in the linear causal bandits literature [15, 26, 27]. These assumptions imply boundedness of the variables, i.e., there exists constant $m$ such that $\|X\| \leq m$. The recent study in [28] shows that in linear bandits, the regret will scale linearly with this constant. Without loss of generality, we assume $m_\epsilon = m_{\mathbf{B}} = 1$. Finally, we provide the following standard regularity condition on interventions to ensure sufficient distinction between the observational and interventional statistical models [20, 23].

**Assumption 5** (Intervention regularity). *A soft intervention on node $i$ with causal depth $L_i \geq 1$ shifts the expected values of the descendants of $i$ at least $\eta$, i.e., $\left|\mu_{j,\emptyset} - \mu_{j,\{i\}}\right| > \eta$ for all $i \in [N]$ and $j \in \mathsf{de}(i)$.*

## 3 Graph-Agnostic Linear Causal Bandit (GA-LCB) Algorithm

In this section, we introduce our proposed algorithm **G**raph-**A**gnostic **L**inear **C**ausal **B**andit (GA-LCB). We also provide detailed comparisons to the existing algorithms designed for soft interventions. We will provide the regret analysis in Section 4, and defer all the proofs to the appendices.

**Algorithm overview.** Identifying the best intervention $\mathbf{a}^*$ defined in (4) hinges on learning all the possible probability distributions $\mathbb{P}_{\mathbf{a}}$, the number of which grows exponentially with graph size $N$. Learning such an excessive number of distributions can be circumvented by properly leveraging the SEM parameters. Specifically, all the distributions $\{\mathbb{P}_{\mathbf{a}} : \mathbf{a} \in \mathcal{A}\}$ are functions of the observational and interventional weight matrices $\mathbf{B}$ and $\mathbf{B}^*$. Furthermore, we note that each of these matrices consists of at most $Nd$ non-zero entries. Hence, learning the entire set of distributions is equivalent to estimating at most $2Nd$ non-zero parameters of $\mathbf{B}$ and $\mathbf{B}^*$. This problem, however, faces the hard constraint that the estimated matrices $\mathbf{B}$ and $\mathbf{B}^*$ must conform to a valid DAG structure. Not enforcing this constraint gives rise to issues such as the possibility of support structures that include cycles or inconsistent supports for the estimates of $\mathbf{B}$ and $\mathbf{B}^*$.

For this DAG-constrained problem of estimating matrices $\mathbf{B}$ and $\mathbf{B}^*$, we take a two-step approach. The first step aims to resolve the skeleton uncertainty to the extent needed to identify the best intervention, and the second step leverages the skeleton estimates to identify the best intervention design. More specifically, the first step (GA-LCB-SL in Algorithm 1) focuses on estimating the skeleton, which is equivalent to estimating the parent sets $\{\mathsf{pa}(i) : i \in [N]\}$. Forming such estimates based on *soft* interventions is fundamentally different from doing so based on *do* intervention setting [20, 23] since under *do* interventions, identifying $\mathsf{pa}(N)$ suffices to determine the best intervention when there are no confounders. This is because *do* interventions remove all ancestral dependence

---

**Algorithm 1** Graph-Agnostic Linear Causal Bandit: Structure Learning (GA-LCB-SL)

---

1: Inputs: identifiability parameter $\eta$, sufficient exploration conditions $T_1$ and $T_2$
2: $t = 0$
3: **while** $\widehat{\mathcal{G}}$ is not a DAG or $t \leq (N+1)T_1$ **do**
4:     Pull the arm $\mathbf{a}(Nt+1) = \emptyset$ and observe $X(t)$ and set $t = t+1$
5:     **for** $i \in [N-1]$ **do**
6:         Pull the arm $\mathbf{a}(Nt+i+1) = \{i\}$ and observe $X(t)$
7:         Identify the ancestors set $\widehat{\mathsf{an}}(i)$ according to (9) and construct $\widehat{\mathcal{G}}$
8:         $t = t+1$
9: **if** $N_\emptyset < T_2$ **then** Pull the arm $\mathbf{a}(t) = \emptyset$ and observe $X(t)$ until $N_\emptyset = T_2$
10: Calculate the Lasso estimator $[\widehat{\mathbf{B}}]_i^{\mathsf{Lasso}}$ as in (10)
11: Identify the parents: $\widehat{\mathsf{pa}}(i) = \mathrm{supp}\left([\widehat{\mathbf{B}}]_i^{\mathsf{Lasso}}\right)$
12: Return: $\{\widehat{\mathsf{pa}}(i) \mid i \in [N]\}$ and topological ordering $\widehat{\pi}$ based on ancestors sets

---

of $X_N$, and its statistical model can be specified only by the value assigned to its parents under a *do* intervention. Under soft interventions, however, all the causal paths from the youngest nodes to the reward node remain intact. This means that all nodes along the causal paths that end at the reward node contribute to the reward. Therefore, inevitably, we need to estimate the parent sets $\{\mathsf{pa}(i) : i \in [N]\}$. Motivated by these, Algorithm 1 provides estimates $\{\widehat{\mathsf{pa}}(i) : i \in [N]\}$ such that with a high probability: (1) $\mathsf{pa}(i) \subseteq \widehat{\mathsf{pa}}(i)$; and (2) $|\widehat{\mathsf{pa}}(i)| \leq c|\mathsf{pa}(i)|$ for a small constant $c > 1$. We note that our approach is distinct from the conventional approaches to learning causal skeletons, which typically identify only the Markov equivalence class and assume the existence of an oracle rather than focusing on sample complexity and computational efficiency. For linear non-Gaussian data, a DAG can be learned using observational data with a sample complexity of $O(d^4 \log n)$ [29]. In comparison, our CB-based structure learning saves on sample complexity and computation.

The second step is focused on narrowing the search space for the set of candidate interventions among which the optimal one is identified. This will provide significant computational savings as we will discuss. In this step, specifically, based on the estimates $\{\widehat{\mathsf{pa}}(i) : i \in [N]\}$, the GA-LCB-ID algorithm performs a successive refinement of the set $\mathcal{A}$ to identify the intervention set of interest. This process consists of $S = \lceil \log \sqrt{T} \rceil$ refinement stages, where the refined set in stage $s \in [S]$ is denoted by $\hat{\mathcal{A}}_s \subset \mathcal{A}_{s-1}$ with $\hat{\mathcal{A}}_1 \triangleq \mathcal{A}$. To identify the interventions to be eliminated at stage $s$, the GA-LCB-ID algorithm identifies the interventions whose UCB values fall below the maximum in $\hat{\mathcal{A}}_s$ minus a bandwidth $m2^{1-s}$. Such successive refinement allows us to calculate UCBs only for promising interventions, leading to a higher computational efficiency. Furthermore, the refinement rules do not need to calculate the exact UCB values. Instead, they calculate an upper bound for the UCBs, referred to as the UCB *widths*. This circumvents the computational challenge of calculating the exact UCBs.

### 3.1 Step 1: CB-based Structure Learning

Our approach to using a sequence of interventions to learn the unknown graph $\mathcal{G}$ consists of two procedures. It starts by identifying the correct ancestors $\mathsf{an}(i)$ for $i \in [N]$. After $T_1$ rounds of exploration, for all $i, j \in [N]$ we compute the mean estimates as

$$\hat{\mu}_{i,\emptyset}(t) \;=\; \frac{1}{N_\emptyset(t)} \sum_{\tau \in [t], \mathbf{a}(\tau)=\emptyset} X_i(\tau)\,, \quad \text{and} \quad \hat{\mu}_{i,\{j\}}(t) \;=\; \frac{1}{N_{\{j\}}(t)} \sum_{\tau \in [t], \mathbf{a}(\tau)=\{j\}} X_i(\tau)\,, \quad (6)$$

where $N_\mathbf{a}(t)$ denote the number of times the $\mathbf{a} \in \mathcal{A}$ is selected up to time $t$

$$N_\mathbf{a}(t) \;\triangleq\; \sum_{\tau \in [t]} \mathbb{1}\{\mathbf{a}(\tau) = \mathbf{a}\}\,. \tag{7}$$

Subsequently, the algorithm identifies descendants sets $\widehat{\mathsf{de}}(i)$ for $i \in [N]$ according to:

$$\widehat{\mathsf{de}}(i) \;=\; \left\{ j \in [N] \;:\; |\hat{\mu}_{j,\emptyset} - \hat{\mu}_{j,\{i\}}| > \frac{\eta}{2} \right\}\,, \tag{8}$$

**Algorithm 2** Graph-Agnostic Linear Causal Bandit: Intervention Design (GA-LCB-ID)

1: Inputs: Time Horizon $T$, $S = \lceil \log \sqrt{T} \rceil$, exploration parameter $\alpha \in \mathbb{R}^+$, {identifiability parameter $\eta$, sufficient exploration conditions $T_1$ and $T_2$} or {edge set $\mathcal{E}$},
2: **if** $\mathcal{E}$ not given **then**
3:     $\{\widehat{\mathsf{pa}}(i) \mid i \in [N], \widehat{\pi} = \text{GraphLearning}(\eta, T_1, T_2)$
4: set $s = 1$ and $\hat{\mathcal{A}}_1 = \mathcal{A}$
5: **while** $t \leq T$ **do**
6:     **for** $i \in [N]$ **do**
7:         Calculate the ridge regression estimators $[\mathbf{B}(t-1)]_i$ and $[\mathbf{B}^*(t-1)]_i$ as in (13) and (14)
8:     **for** $\mathbf{a} \in \hat{\mathcal{A}}_s$ **do**
9:         **for** $i \in \widehat{\pi}$ **do**
10:             Calculate estimated mean $\hat{\mu}_{i,\mathbf{a}}(t)$ as in (17)
11:             Calculate the width $w_{i,\mathbf{a}}(t-1)$ according to (19)
12:         Calculate $\text{UCB}_{\mathbf{a}}(t-1) = \hat{\mu}_{N,\mathbf{a}}(t-1) + w_{N,\mathbf{a}}(t-1)$
13:     **if** $w_{N,\mathbf{a}}(t-1) \leq m\frac{1}{\sqrt{T}}$ for all $\mathbf{a} \in \hat{\mathcal{A}}_s$ **then**
14:         Choose $\mathbf{a}(t)$ according to (23) until $t = T$
15:         **Break**
16:     **while** $w_{N,\mathbf{a}}(t-1) \leq m2^{-s}$ for all $\mathbf{a} \in \hat{\mathcal{A}}_s$ **do**
17:         Update $\hat{\mathcal{A}}_{s+1}$ as in (22) and set $s = s + 1$
18:     Choose $\mathbf{a}(t) \in \hat{\mathcal{A}}_s$ such that $w_{\mathbf{a}(t)}^s > m2^{-s}$

according to which clearly $i \in \widehat{\mathsf{de}}(i)$. Note that $|\mathsf{de}(i)| = 0$ indicates that node $i$ is a root node (intervention on a root node does not change the conditional distributions). Furthermore, we also estimate the ancestors sets $\{\widehat{\mathsf{an}}(i) : i \in [N]\}$ according to:

$$\widehat{\mathsf{an}}(i) = \left\{ j \in [N] \; : \; i \neq j \text{ and } \{|\widehat{\mathsf{de}}(j)| = 0 \text{ or } i \in \widehat{\mathsf{de}}(j)\} \right\}. \tag{9}$$

The algorithm will check whether the ancestor sets will form a DAG. This is confirmed by verifying that there does not exist $i, j \in [N]$ such that $j \in \widehat{\mathsf{de}}(i)$ and $i \in \widehat{\mathsf{de}}(j)$. To further refine the estimate of the parent set, the algorithm initiates $T_2$ rounds of additional explorations. This ensures the algorithm has gathered sufficient observational data to accurately identify the parent set $\mathsf{pa}(i)$. Subsequently, it uses the Lasso estimator on the ancestors set with $\lambda = m\sqrt{\frac{2\log(4N|\widehat{\mathsf{an}}(i)|/\delta)}{N_\emptyset(t)}}$ for $i \in [N]$ as

$$[\widehat{\mathbf{B}}^{\text{Lasso}}]_i = \operatorname*{argmin}_{\theta \in \mathbb{R}^{|\widehat{\mathsf{an}}(i)|}} \left( \frac{1}{N_\emptyset(t)} \sum_{\tau \in [t], \mathbf{a}(\tau)=\emptyset} \left( X_i(\tau) - \theta^\top X_{\widehat{\mathsf{an}}(i)}(\tau) \right)^2 + \lambda\|\theta\|_1 \right). \tag{10}$$

Based on these steps, Algorithm 1 identifies the parent set of node $i \in [N]$ as

$$\widehat{\mathsf{pa}}(i) = \operatorname{supp}\left( [\widehat{\mathbf{B}}^{\text{Lasso}}]_i \right). \tag{11}$$

Specifically, Algorithm 1 returns the estimates $\{\widehat{\mathsf{pa}}(i) : i \in [N]\}$ and a valid topological order $\hat{\pi}$ based on ancestor information. Given a causal graph $\mathcal{G}$, an ordered permutation of $[N]$, denoted by $\pi$ is said to be a valid topological order if for each edge $(i \to j) \in \mathcal{E}$, we have $\pi_i < \pi_j$. This can be achieved by iteratively adding nodes to $\pi$ such that the parents of that node are already included in $\pi$. Finally, we note that we set the exploration constants $T_1$ and $T_2$ as follows.

$$T_1 = \frac{32m^2}{\eta^2} \log\left(\frac{2N^2}{\delta}\right), \quad \text{and} \quad T_2 = cd\log(N), \tag{12}$$

where $c > 1$ is a constant.

### 3.2 Step 2: Sequential Design of Interventions

Assume at the stage $s \in S \triangleq \lceil \log \sqrt{T} \rceil$, the algorithm maintains a refined set $\hat{\mathcal{A}}_s \subset \mathcal{A}$. It starts with the set of candidates $\hat{\mathcal{A}}_1 = \mathcal{A}$ and successively refines this set by performing elimination on the previous refined set using the UCB *width*. Based on the outputs of Algorithm 1, we first estimate

the weight matrices by ridge regressions. Specifically, based on the choices of interventions and the observed data up to time $t$, we estimate the column vectors $\{[\mathbf{B}]_i, [\mathbf{B}^*]_i : i \in [N]\}$ as follows

$$[\mathbf{B}(t)]_i \triangleq [\mathbf{V}_i(t)]^{-1} \sum_{\tau \in [t]: i \notin \mathbf{a}(\tau)} X_{\widehat{\mathsf{pa}}(i)}(\tau)(X_i(\tau) - \nu_i) , \tag{13}$$

$$\text{and} \qquad [\mathbf{B}^*(t)]_i \triangleq [\mathbf{V}_i^*(t)]^{-1} \sum_{\tau \in [t]: i \in \mathbf{a}(\tau)} X_{\widehat{\mathsf{pa}}(i)}(\tau)(X_i(\tau) - \nu_i) , \tag{14}$$

where we have defined the gram matrices as

$$\mathbf{V}_i(t) \triangleq \sum_{\tau \in [t]: i \notin \mathbf{a}(\tau)} X_{\widehat{\mathsf{pa}}(i)}(\tau) X_{\widehat{\mathsf{pa}}(i)}^\top(\tau) + \mathbf{I}_N , \mathbf{V}_i^*(t) \triangleq \sum_{\tau \in [t]: i \in \mathbf{a}(\tau)} X_{\widehat{\mathsf{pa}}(i)}(\tau) X_{\widehat{\mathsf{pa}}(i)}^\top(\tau) + \mathbf{I}_N , \tag{15}$$

and $\mathbf{I}_N$ is the diagonal matrix with elements 1. Accordingly, we denote our estimate of $\mathbf{B_a}$ for intervention $\mathbf{a} \in \mathcal{A}$ at time $t$ by $\mathbf{B_a}(t)$, which are constructed as follows.

$$[\mathbf{B_a}(t)]_i \triangleq \mathbb{1}\{i \in \mathbf{a}\}[\mathbf{B}^*(t)]_i + \mathbb{1}\{i \notin \mathbf{a}\}[\mathbf{B}(t)]_i . \tag{16}$$

Based on the estimates, we construct the estimator of mean value $\mu_{i,\mathbf{a}}$ as

$$\hat{\mu}_{i,\mathbf{a}}(t) \triangleq \langle f_i(\mathbf{B_a}(t)), \, \boldsymbol{\nu} \rangle , \tag{17}$$

where $f_i(\mathbf{B_a}) \triangleq \sum_{\ell=0}^{L_i} [\mathbf{B_a^\ell}]_i$. Given all the information up to time $t$, the next decision is to identify the intervention $\mathbf{a}(t+1)$. For this purpose, a standard UCB-type approach entails forming a confidence interval for the mean estimates followed by identifying the intervention that achieves the largest UCB. In our algorithm, we avoid such an optimization-based approach and instead compute the UCBs according to

$$\text{UCB}_\mathbf{a}(t) = \hat{\mu}_{N,\mathbf{a}(t)} + w_{N,\mathbf{a}}(t) , \tag{18}$$

in which we refer to $w_{N,\mathbf{a}}(t)$ as the UCB *width* and it is defined recursively as follows.

$$w_{i,\mathbf{a}}(t) \triangleq \sum_{j \in \widehat{\mathsf{pa}}(i)} w_{j,\mathbf{a}} + \alpha\Big(\|\hat{\mu}_{\widehat{\mathsf{pa}}(i)}\|_{[\mathbf{V}_{i,a_i}(t)]^{-1}} + m_{\widehat{\mathsf{pa}}, L_i} \lambda_{\min}^{-1/2}\big(\mathbf{V}_{i,a_i}(t)\big)\Big) , \tag{19}$$

in which $\alpha \in \mathbb{R}^+$ controls the size of width, and $\{m_{\widehat{\mathsf{pa}},\ell} \mid \ell \in [L]\}$ is defined as $m_{\widehat{\mathsf{pa}},\ell} \triangleq \max_{i \in [N], L_i = \ell} \|X_{\widehat{\mathsf{pa}}(i)}\|$. Subsequently, we define the post-intervention gram matrices $\mathbf{V}_{i,a_i}(t)$ as follows.

$$\mathbf{V}_{i,a_i}(t) \triangleq \mathbb{1}\{a_i(s) = 1\}\mathbf{V}_i^*(t) + \mathbb{1}\{a_i(s) = 0\}\mathbf{V}_i(t) , \tag{20}$$

where $a_i(s) = \mathbb{1}\{i \in \mathbf{a}(s)\} \in \{0, 1\}$. A standard UCB-type approach selects the intervention set that maximizes $\text{UCB}_\mathbf{a}$ within $\mathcal{A}$. However, this involves solving optimization problems over a set of cardinality $2^N$, which becomes another computational bottleneck. Phased elimination occurs when the uncertainty of the mean estimator, as captured by the UCB width, is sufficiently small for all interventions in the candidate subset.

$$w_{N,\mathbf{a}}(t-1) \leq m2^{-s} , \qquad \forall \mathbf{a} \in \hat{\mathcal{A}}_s , \tag{21}$$

where $s \in \mathbb{N}$ is the current stage. At this point, only the interventions with UCB values close to the optimistic one are retained.

$$\hat{\mathcal{A}}_{s+1} = \Big\{\mathbf{a} \in \hat{\mathcal{A}}_s \mid \text{UCB}_\mathbf{a}(t) \geq \max_{\mathbf{a} \in \hat{\mathcal{A}}_s} \text{UCB}_\mathbf{a}(t) - m2^{1-s}\Big\} . \tag{22}$$

Alternatively, if (21) does not holds, there is some intervention $\mathbf{a} \in \hat{\mathcal{A}}_s$ such that $w_\mathbf{a}^s(t) > m2^{-s}$ which prevents the perform of elimination. This intervention will then be selected at time $t + 1$ to accelerate the elimination.

This procedure continues until $t = T$ or if the stopping criterion $w_{i,\mathbf{a}}(t) \leq m\frac{1}{\sqrt{T}}$ is met. The stopping criterion indicates that the interventions are approximately equally good given the time horizon $T$. The algorithm will then select the intervention that maximizes $\text{UCB}_\mathbf{a}$ as

$$\mathbf{a}(t+1) = \arg\max_{\mathbf{a} \in \hat{\mathcal{A}}_s} \text{UCB}_\mathbf{a}(t) . \tag{23}$$

Finally, we note that we set $\alpha = \sqrt{\frac{1}{2}\log(\frac{NT}{\delta})} + \sqrt{d}$.

## 3.3 Computational Efficiency

We compare the computational efficiency of our algorithm with those of the existing algorithms for linear SEMs with soft interventions in [15, 18]. The algorithms in these studies adopt similar procedures: they find estimators for observational and interventional weights, form the confidence ellipsoids for the weights, and solve a joint optimization problem to calculate UCBs as

$$\text{UCB}_{\mathbf{a}} = \max_{\widetilde{\mathbf{B}}_{\mathbf{a}} \in \mathcal{C}_{\mathbf{a}}} \langle f_N(\widetilde{\mathbf{B}}_{\mathbf{a}}), \boldsymbol{\nu} \rangle , \tag{24}$$

where $\mathcal{C}_{\mathbf{a}} = \prod_{i \in [N]} \mathcal{C}_{i,\mathbf{a}}$ and $\mathcal{C}_{i,\mathbf{a}} = \mathbb{1}\{i \in \mathbf{a}\}\mathcal{C}_i^* + \mathbb{1}\{i \notin \mathbf{a}\}\mathcal{C}_i$ are confidence regions, $f_N$ is compounding function (see [15, Lemma 1] or Appendix C). Subsequently, the interventions are chosen as those that maximize $\text{UCB}_{\mathbf{a}}$

$$\mathbf{a}(t+1) = \arg\max_{\mathbf{a} \in \mathcal{A}} \text{UCB}_{\mathbf{a}} . \tag{25}$$

All the algorithms estimate the observational and interventional weights by solving $2N$ ridge regressions, which is not the computation bottleneck. The two bottlenecks in this standard UCB-type approach lie in solving (24) and (25).

First, different from the case in linear bandits [30], the optimization problem in (24) for CBs is neither convex nor concave. This is due to the highly non-linear reward function. The nonlinearity arises from the compounding effects of the causal influences along different paths leading to the reward node. The contribution of any given node $X_i$ to the reward value will be multiplied by all the coefficients along the path connecting $X_i$ to the reward node (see Appendix C for more details). When there are multiple such paths, the aggregate weight products of all paths carry the contribution of $X_i$ to the reward node. Therefore, the reward becomes a function of the product of causal weights (i.e., elements of $\mathbf{B}$ and $\mathbf{B}^*$). This non-linearity in weights makes the optimization problem in (24) becomes computationally impossible for larger graphs. In contrast, GA-LCB addresses this issue by computing the upper confidence bounds iteratively through causal depth, which can be done in polynomial time.

Secondly, solving (25) involves an optimization problem over a discrete set of size $2^N$, the computational complexity of which grows exponentially with $N$. To circumvent this, GA-LCB randomly chooses the under-explored intervention (line 18). UCB optimization specified in (23) is performed only when the refinement process is completed, indicating that (23) will be solved over a small subset of sufficiently good interventions.

## 4 Regret Analysis

We show that the GA-LCB is almost minimax optimal by characterizing the achievable regret of the GA-LCB algorithm in the graph-independent setting and establishing that it matches a minimax regret lower bound. We provide additional discussions to interpret the dependence of the regret terms on various graph parameters and the relationship of these results vis-á-vis the existing results in the literature. We also present an improved graph-dependent bound, when additional information about the graph is available.

### 4.1 Graph-independent bounds

We first show the graph-independent bounds that hold for a class of bandits with a maximum in-degree $d$ and maximum causal length $L$. The key steps in these analyses involve determining the exploration time that ensures the identification of the parent sets with high probability and bounding the time instances that the refinement process is conducted. To delineate a regret upper bound, we start by establishing the performance guarantee for the GA-LCB-SL algorithm. In the following theorem, we demonstrate that with high probability, this algorithm correctly identifies the topological ordering and the parent sets. For this purpose, we define $\kappa_{\max}$ and $\kappa_{\min}$ as the maximum and minimum eigenvalue of the following second moment with null intervention:

$$\kappa_{\max} \triangleq \lambda_{\max}\left(\mathbb{E}_{\emptyset}\left[XX^\top\right]\right) , \quad \kappa_{\min} \triangleq \lambda_{\min}\left(\mathbb{E}_{\emptyset}\left[XX^\top\right]\right) . \tag{26}$$

**Theorem 1** (Achievable Graph Skeleton Learning). *Under Assumptions 1–4, the GA-LCB-SL algorithm ensures that*

1. *with probability $1 - \delta$ we have a valid topological ordering $\widehat{\pi}$; and*

2. *with probability at least $1 - 2\delta$, for all $i \in [N]$ we have $\mathsf{pa}(i) \subseteq \widehat{\mathsf{pa}}(i)$ and $|\widehat{\mathsf{pa}}(i)| \leq \kappa|\mathsf{pa}(i)|$,*

*where $\kappa$ is defined as*

$$\kappa = \frac{9 \min\left\{ m^2, \kappa_{\max} + m^2 \sqrt{\frac{16}{3T_2} \log\left(\frac{2dN}{\delta}\right)} \right\}}{\kappa_{\min}} . \tag{27}$$

Leveraging the result of Theorem 1, we characterize the achievable regret of GA-LCB.

**Theorem 2** (Achievable Regret)**.** *Under Assumptions 1–5, the GA-LCB-ID algorithm ensures that with probability $1 - 3\delta$*

$$\mathbb{E}[\mathcal{R}(T)] \leq \tilde{\mathcal{O}}\big((\kappa d)^{L - \frac{1}{2}} \sqrt{T} + d + RN\big) , \tag{28}$$

*where we have defined $R \triangleq \frac{m^2}{\eta^2}$.*

We note that $R$ represents the guaranteed maximum signal-to-intervention-power ratio. This ratio measures the difficulty in distinguishing between observational and interventional distributions (the higher $R$, the harder to distinguish).

To emphasize the cost of learning the skeleton, in the next corollary, we provide the achievable regret of the GA-LCB algorithm when the graph skeleton is known.

**Corollary 1** (Achievable Regret – Known Skeleton)**.** *When the graph skeleton is known, under the same setting as in Theorem 2, with probability $1 - \delta$, GA-LCB-ID ensures that*

$$\mathbb{E}[\mathcal{R}(T)] \leq \tilde{\mathcal{O}}\big(d^{L - \frac{1}{2}} \sqrt{T}\big) . \tag{29}$$

Comparing Corollary 1 with Theorem 1, we observe that the term $\tilde{\mathcal{O}}(d + RN)$ represents the cost to do structure learning, while $\kappa^{L - \frac{1}{2}}$ reflects the cost resulting from imperfect graph learning.

Next, we establish a lower bound on the regret. Any lower bound on the regret of the setting in which the graph's skeleton is *known* will immediately serve as a lower bound for our setting with an *unknown* graph. We will present one such lower bound and show that even though it is expectedly looser than a lower bound for our setting, it still almost matches the achievable regret characterized in Theorem 2. We also emphasize that our result improves the known minimax lower bound when the graph skeleton is known (c.f. [15, 18]).

**Theorem 3** (Regret Lower Bound)**.** *For any given skeleton with parameters $d$ and $L$, there exists a causal bandit instance such that the expected regret of any algorithm is at least*

$$\mathbb{E}[\mathcal{R}(T)] \geq \Omega\big(d^{L - \frac{3}{2}} \sqrt{T}\big) . \tag{30}$$

When comparing the upper bound in Theorem 2 and the lower bound in Theorem 3, we observe that the regret upper bound and lower bound show similar behavior with respect to graph parameter $d$, $L$, and the time horizon $T$. Given these results, we provide some observations.

- **Dependence on $N$.** We first note that the achievable regret has a diminishing dependence on the graph size $N$ as $T$ grows. This is especially important since the number of interventions grows exponentially with $N$. This result indicates that the achievable regret has a diminishing effect not only on the graph size but also on the cardinality of the intervention space.

- **Unknown Skeleton.** Comparing Theorem 2 and Corollary 1 indicates that the impact of an unknown graph has two parts. First, sufficient exploration is required to determine the correct topological ordering and parent sets, which adds a $\tilde{\mathcal{O}}(d + RN)$ term to the regret bound. Secondly, the imperfect identification of the parent set by the Lasso estimator leads to an estimated graph with a maximum in-degree of $cd$ instead of $d$, which is propagated through the network layers.

- **Graph topology.** The regret bounds depend on the graph through its connectivity parameters $d$ and $L$. Unlike the observations in [15, 18], we have almost-matching upper and lower bounds up to a $d$ factor. This significantly improves from the previously-known gap of $d^L$.

- **Dependence on** $T$**.** Regret upper and lower bound both scale with $T$ at the rate $\tilde{\mathcal{O}}(\sqrt{T})$.

- **Linear bandits.** Finally, we note that despite some similarities, our problem is significantly different from linear bandits since, as shown in Appendix C, in linear causal bandits, the reward is non-linear with respect to the parameters or the interventions. This is the case for even $L = 1$. We note that our regret's dependence on $d$ differs from that in the linear bandit setting. In linear bandits, the regret scales linearly with $d$ as shown in [31, 32]. When $L = 1$, the regret upper bound in our case scales as $\tilde{\mathcal{O}}(\sqrt{dT})$, and the regret lower bound scales as $\Omega(\sqrt{T})$. We conjecture that the regret upper bound is tighter because it more accurately captures the uncertainty of parameter estimation.

- **Regret bounds comparisons.** GA-LCB provides significanty improved regret bounds compared to LinSEM-UCB [15] and GCB-UCB [18]. Specifically, under the known graph skeleton setting, GA-LCB achieves a $d^L \sqrt{N}$ factor improvement in the regret bounds compared to LinSEM-UCB. While GCB-UCB removes the $\sqrt{N}$ factor, it underperforms compared to LinSEM-UCB. Furthermore, our regret upper bound has only a $d$ factor more than the lower bound.

### 4.2 Graph-dependent Regret Bound

The graph-independent bounds can be further refined to recover graph-dependent regret bounds that use the instance-level information. To account for the actual influence of the graph parameters on the reward node, we define the *effective maximum in-degree* as $d_e = \max_{i \in \mathsf{an}(N)} d_i$ and the *effective maximum causal depth* as $L_e = \max_{i \in \mathsf{an}(N)} L_i$. We have the natural inequalities $d_e \leq d$ and $L_e \leq L$. To characterize the graph-dependent bound, we need to slightly modify the GA-LCB-ID algorithm. Specifically, we only need to identify the optimal intervention within $\mathcal{A}' = 2^{[\widehat{\mathsf{an}}(N)]}$ and estimate the column vectors $\{[\mathbf{B}]_i, [\mathbf{B}^*]_i : i \in \widehat{\mathsf{an}}(N)\}$. By incorporating instance-specific information about the graph structure, the regret upper bound can be further refined as follows.

**Corollary 2** (Achievable Regret - Graph-Dependent)**.** *When* $L_e$ *and* $d_e$ *are known, the modified GA-LCB-ID algorithm ensures that with probability* $1 - 3\delta$

$$\mathbb{E}[\mathcal{R}(T)] \leq \tilde{\mathcal{O}}\big((\kappa d_e)^{L_e - \frac{1}{2}} \sqrt{T} + d + RN\big) . \tag{31}$$

By comparing the upper bound in Theorem 2 and Corollary 2, we observe that the cost of learning the graph remains intact. The reason is that we must explore interventions on every node $i \in [N]$ to identify the ancestor relationships, even when graph-dependent information is known. Hence, all the regret improvements are due to the part of learning the best intervention, particularly in relation to the graph topology. The term $(\kappa d)^{L - \frac{1}{2}}$ in Theorem 2 is replaced with $(\kappa d_e)^{L_e - \frac{1}{2}}$. The change is due to the fact we do not need to learn optimal intervention in the whole graph $\mathcal{G}$ as the interventions on non-ancestor nodes will not affect the reward. Instead, it suffices to learn only the optimal intervention on the subgraph $\tilde{\mathcal{G}}$ formed by $\widehat{\mathsf{an}}(N)$ and the parameters of $\tilde{\mathcal{G}}$.

## 5  Conclusions

In this paper, we have solved the causal bandit problem with unknown graph skeletons under general stochastic interventions. We have proposed an implementable algorithm and provided regret analysis for both unknown and known graph skeletons. The unknown skeleton affects the achievable regret bounds in two ways: a term that is linear in $d + N$ but is independent of $T$ and a $cd$ factor due to the imperfect identification of the parents. When the graph skeleton is unknown, the achievable regret bounds and the minimax regret lower bound are shown to match up to a $d$ factor. Compared to the existing algorithms, the proposed algorithm is more amenable to scalable implementation.

## Acknowledgments

This work was supported in part by the U.S. National Science Foundation under Grant DMS-2319996, and in part by the Rensselaer-IBM Future of Computing Research Collaboration (FCRC).

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

# Linear Causal Bandits: Unknown Graph and Soft Interventions Supplementary Materials

## Table of Contents

## A  Empirical Evaluations

In this section, we assess the regret performance of GA-LCB. As the most relevant existing approaches, we compare the regret of our algorithm to those of LinSEM-UCB [15] and GCB-UCB [18], which are designed for causal bandits with soft interventions.

**Causal graph.** We consider the hierarchical graph illustrated in Figure 1. This graph consists of $(L + 1)$ layers, with the first $L$ layers having $d$ nodes. The nodes between two adjacent layers are fully connected. The last layer consists of one node (reward node) that is fully connected to nodes in layer $L$. The number of nodes in this graph is $N = dL + 1$.

**Parameter setting.** The noise terms $\{\epsilon_i : i \in [N]\}$ are set to be drawn from the uniform distribution $\mathsf{Unif}(0, 1)$. We set the non-zero elements in the observational and interventional weights matrix to 1 and 0.5, respectively. We evaluate $L \in \{2, 4, 6\}$. The experiment was conducted using 2 CPUs from Mac Mini 2023. We set $T_1 = T_2 = 500$ for experiments with $L = 2$, $T_1 = T_2 = 1000$ for experiments with $L = 4$ and $L = 6$.

**Algorithm settings.** The theoretical guarantees relied on specific technical conditions on parameters. We observe that the algorithms designed can provide better-than-foreseen empirical performance by tuning the parameters involved. Specifically, we adjusted the parameters $\lambda$, $\alpha$, and the sufficient exploration times $T_1$ and $T_2$. We observe that setting $\lambda = \alpha = 0.1$ yields reasonable performance. The experiments are repeated 100 times, and the average cumulative regret is reported.

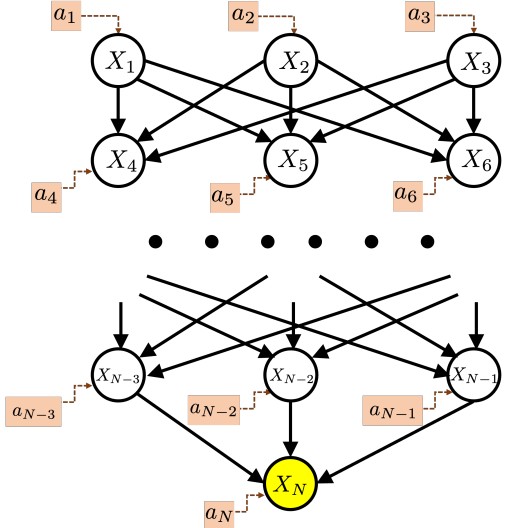

Figure 1: Example of hierarchical graph.

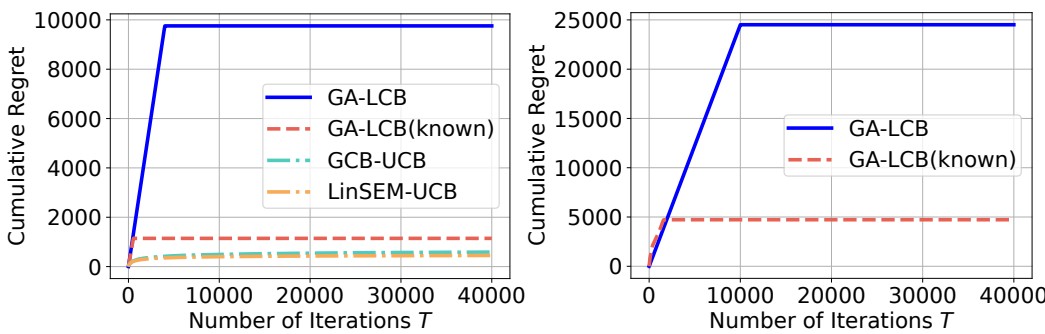

Figure 2: Cumulative regret with $L = 2$.  Figure 3: Cumulative regret with $L = 4$.

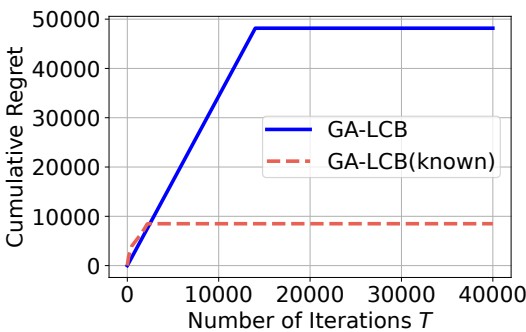

Figure 4: Cumulative regret with $L = 6$.

**Regret performance.** In Figure 2, we present the cumulative regret of GA-LCB algorithm and other algorithms with the hierarchical graph with $d = 3$ and $L = 2$. LinSEM-UCB and GCB-UCB exhibit lower regret within the shown time horizon, while our algorithm (GA-LCB with known graph) shows a slightly higher regret. This difference is due to balancing a more precise confidence radius with enhanced scalability. Besides, we observe that GA-LCB incurs higher regret due to the structure learning phase. Since both LinSEM-UCB and GCB-UCB face computational challenges for scaling up to larger graphs, we evaluate the cumulative regret of GA-LCB under known and unknown graph

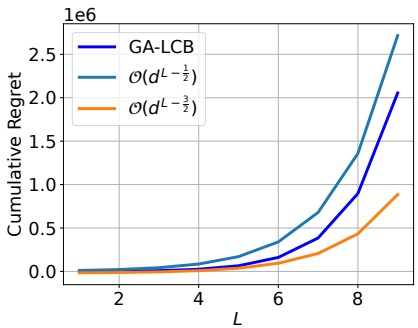

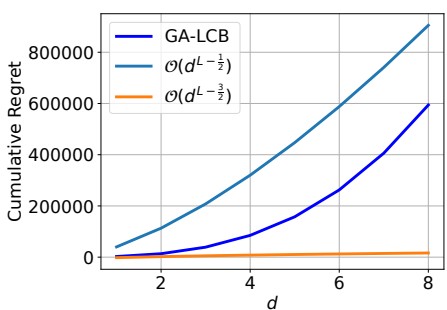

Figure 5: Cumulative regret with different length $L$ under hierarchical graph with $d = 2$.

Figure 6: Cumulative regret with different degree $d$ under hierarchical graph with $L = 2$.

settings when the maximum causal depth is $L = 4$ in Figure 3 and $L = 6$ in Figure 4. We show the scalability of the GA-LCB algorithm. Besides, comparing the regret of known and unknown settings, we see the additional cost of structured learning is diminishing, which is desirable. The fluctuation of GA-LCB under the known graph setting is due to imperfect phased eliminations which remove the interventions with larger expected values first.

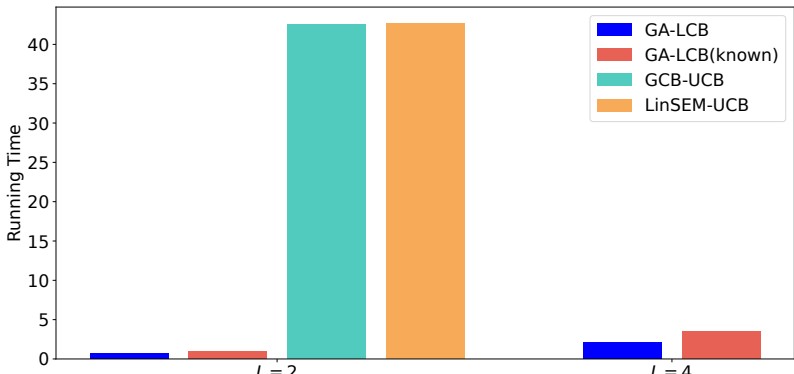

Figure 7: Computational time of different algorithms.

**Scaling with $L$:** Scaling of the regret with respect to the causal depth $L$ is depicted in Figure 5 for the setting of a hierarchical graph with $d = 2$ and $L$ varying in the range $\{1, \cdots, 9\}$. The theoretical results (regret upper and lower bounds) predict that the regret grows at the rate $(2\kappa)^L$ (i.e., exponential in $L$). The empirical results in Figure 5 corroborate that the cumulative regret scales exponentially with length $L$, and the actual regret closely tracks the upper bound's trend.

**Scaling with $d$:** Scaling of the regret with respect to the maximum in-degree $d$ is depicted in Figure 6 for a hierarchical graph with $L = 2$. We increase the number of sufficient exploration parameters $T_1$ and $T_2$ to ensure we accommodate all settings with different degrees. The theoretical predictions suggest that our algorithm's regret scales as $d^{3/2}$ (i.e., polynomial $d$). Figure 6 demonstrates that our regret is super-linear and tracks the polynomial trend of the regret upper bound (i.e., the achievable regret).

**Computational time.** In Figure 7, we compare the running times of different algorithms under the two settings mentioned above. The figure indicates that our proposed algorithm significantly reduces computational time in the hierarchical graph with $L = 2$ when compared with LinSEM-UCB and GCB-UCB. In the hierarchical graph with $L = 4$, which is considered sufficiently large in the related studies, the GA-LCB still demonstrates a notable stronger running time.

## B  Additional Notations

In this section, we present the notations that will be useful in our analyses. We denote the singular values of a matrix $\mathbf{A} \in \mathbb{R}^{M \times N}$ with $M \geq N$ in the descending order by

$$\sigma_1(\mathbf{A}) \geq \sigma_2(\mathbf{A}) \geq \cdots \geq \sigma_N(\mathbf{A}) . \tag{32}$$

In the proofs, the analyses often involve with zero-padded vectors and their corresponding matrices (e.g., $X_{\mathsf{pa}(i)}$ and $\mathbf{V}_{i,a_i(t)}$). Consequently, these matrices have non-trivial *null spaces*, resulting in zero singular values. In such instances, we are interested in the *effective* smallest singular values that are non-zero. We denote the *effective* largest and smallest eigenvalues, which correspond to the effective dimensions of a positive semidefinite matrix $\mathbf{A}$ with rank $k$, by

$$\sigma_{\max}(\mathbf{A}) \triangleq \sigma_1(\mathbf{A}) , \quad \text{and} \quad \sigma_{\min}(\mathbf{A}) \triangleq \sigma_k(\mathbf{A}) . \tag{33}$$

For a square matrix $\mathbf{U} = \mathbf{A}\mathbf{A}^\top \in \mathbb{R}^{N \times N}$, we denote the *effective* largest and smallest eigenvalues by[2]

$$\lambda_{\max}(\mathbf{U}) \triangleq \sigma_{\max}^2(\mathbf{A}) , \quad \text{and} \quad \lambda_{\min}(\mathbf{U}) \triangleq \sigma_{\min}^2(\mathbf{A}) . \tag{34}$$

Then we construct data matrices that are related to gram matrices. At time $t \in \mathbb{N}$ and for any node $i \in [N]$, the data matrices $\mathbf{U}_i(t) \in \mathbb{R}^{t \times N}$ and $\mathbf{U}_i^*(t) \in \mathbb{R}^{t \times N}$ consist of the weighted observational and interventional data, respectively. Specifically, for any $\tau \in [t]$ and $i \in [N]$, we define

$$\left[\mathbf{U}_i^\top(t)\right]_\tau \triangleq \mathbb{1}\{a_i(\tau) = 0\} X_{\widehat{\mathsf{pa}}(i)}^\top(\tau) , \tag{35}$$

$$\text{and} \quad \left[\mathbf{U}_i^{*\top}(t)\right]_\tau \triangleq \mathbb{1}\{a_i(\tau) = 1\} X_{\widehat{\mathsf{pa}}(i)}^\top(\tau) . \tag{36}$$

We also define the observation matrix that is used for Lasso estimator $\mathbf{U}_{i,\widehat{\mathsf{an}}(i)}(t) \in \mathbb{R}^{t \times N}$ that stores the observational data on the ancestor's set.

$$[\mathbf{U}_{i,\widehat{\mathsf{an}}(i)}^\top(t)]_\tau = \mathbb{1}\{\mathbf{a}(\tau) = \emptyset\} X_{\widehat{\mathsf{an}}(i)}^\top(\tau) . \tag{37}$$

We also define the data vector $\mathbf{D}_i(t) \in \mathbb{R}^t$ and $\mathbf{D}_i^*(t) \in \mathbb{R}^t$ as

$$\left[\mathbf{D}_i(t)\right]_\tau \triangleq \mathbb{1}\{a_i(\tau) = 0\}(X_i(\tau) - \nu_i) , \tag{38}$$

$$\text{and} \quad \left[\mathbf{D}_i^*(t)\right]_\tau \triangleq \mathbb{1}\{a_i(\tau) = 1\}(X_i(\tau) - \nu_i) . \tag{39}$$

Similarly to (2), we denote the relevant data matrices for node $i \in [N]$ under intervention $\mathbf{a} \in \mathcal{A}$ by

$$\mathbf{U}_{i,a_i}(t) \triangleq \mathbb{1}\{a_i(t) = 1\}\mathbf{U}_i^*(t) + \mathbb{1}\{a_i(t) = 0\}\mathbf{U}_i(t) , \tag{40}$$

$$\mathbf{V}_{i,a_i}(t) \triangleq \mathbb{1}\{a_i(t) = 1\}\mathbf{V}_i^*(t) + \mathbb{1}\{a_i(t) = 0\}\mathbf{V}_i(t) . \tag{41}$$

$$\mathbf{D}_{i,a_i}(t) \triangleq \mathbb{1}\{a_i(t) = 1\}\mathbf{D}_i^*(t) + \mathbb{1}\{a_i(t) = 0\}\mathbf{D}_i(t) . \tag{42}$$

Define $N_i^*(t)$ as the number of times that node $i \in [N]$ is intervened, and $N_i(t)$ as its complement

$$N_i^*(t) \triangleq \sum_{\tau=1}^{t} \mathbb{1}\{a_i(\tau) = 1\} , \tag{43}$$

$$\text{and} \quad N_i(t) \triangleq t - N_i^*(t) . \tag{44}$$

Accordingly, for any $i \in [N]$ and $t \in \mathbb{N}$, define

$$N_{i,a_i}(t) \triangleq \mathbb{1}\{a_i(t) = 1\}N_i^*(t) + \mathbb{1}\{a_i(t) = 0\}N_i(t) , \tag{45}$$

For $\mathbf{a} \in \mathcal{A}$, we define the pseudo estimated variables $\widehat{X}_{\mathbf{a}}(t)$ and pseudo underground variables $X_{\mathbf{a}}(t)$ is the random variable at time $t$ generated according to the following linear SEMs

$$\widehat{X}_{\mathbf{a}}(t) = \mathbf{B}_{\mathbf{a}}^\top(t)\widehat{X}_{\mathbf{a}}(t) + \epsilon(t) , \tag{46}$$

$$\text{and} \quad X_{\mathbf{a}}(t) = \mathbf{B}_{\mathbf{a}}^\top X_{\mathbf{a}}(t) + \epsilon(t) , \tag{47}$$

---

[2]For matrix $\mathbf{V} = \mathbf{U} + \mathbf{I}$, we denote the *effective* smallest eigenvalues by $\lambda_{\min}(\mathbf{V}) \triangleq \sigma_{\min}^2(\mathbf{A}) + 1$.

which share the same $\epsilon(t)$ acorss the different interventions.

Finally, let us denote the second moment of the parents of a node $i$ under intervention $\mathbf{a} \in \mathcal{A}$ with unknown weight matrices $\mathbf{B}$ and $\mathbf{B}^*$ by

$$\widetilde{\Sigma}_{i,\mathbf{a}} \triangleq \mathbb{E}_{\mathbf{a}} \left[ X_{\widehat{\mathsf{an}}(i)} X_{\widehat{\mathsf{an}}(i)}^\top \right] . \tag{48}$$

Accordingly, we denote the lower and upper bounds on the minimum and maximum singular values of these moments by

$$\tilde{\kappa}_{i,\mathbf{a},\min} \triangleq \sigma_{\min} \left( \widetilde{\Sigma}_{i,\mathbf{a}} \right), \quad \tilde{\kappa}_{\min} \triangleq \min_{i \in [N]} \tilde{\kappa}_{i,\emptyset,\min}, \tag{49}$$

$$\tilde{\kappa}_{i,\mathbf{a},\max} \triangleq \sigma_{\max} \left( \widehat{\Sigma}_{i,\mathbf{a}} \right), \quad \tilde{\kappa}_{\max} \triangleq \min_{i \in [N]} \tilde{\kappa}_{i,\emptyset,\max}. \tag{50}$$

We note due to the Cauchy interlacing theorem, we have

$$\kappa_{\min} \leq \tilde{\kappa}_{\min} \leq \tilde{\kappa}_{\max} \leq \kappa_{\max} . \tag{51}$$

Finally, we use $\Delta(\mathbf{x})$ to represent the diagonal matrix with elements in $\mathbf{x}$.

## C  Decomposition of Node-level Rewards

Similar to [15, Lemma 1], we present the following decomposition for the expected value of the variable $X_i$ for $i \in [N]$. We note our design of the mean value estimator in (17) based on this lemma.

**Lemma 1.** *Given intervention $\mathbf{a} \in \mathcal{A}$, the value of node $X_i$ is related to the noise vector $\epsilon$ via $X_i = \langle f_i(\mathbf{B_a}), \epsilon \rangle$, where $f_i(\mathbf{B_a}) \triangleq \sum_{\ell=0}^{L_i} \left[ \mathbf{B_a^\ell} \right]_i$. Consequently, the expected value of $X_i$ under intervention $\mathbf{a}$ is*

$$\mu_{i,\mathbf{a}} = \langle f_i(\mathbf{B_a}), \boldsymbol{\nu} \rangle . \tag{52}$$

*Proof.* To capture the contribution of $\epsilon_j$ on node $i$, we use the fact that the entry at row $j$ and column $i$ of $\mathbf{B_a^\ell}$ is the sum aggregate of the weight products along all paths from node $j$ to node $i$ that have the exact length $\ell$. Since the longest length will be $L_i$, the term $\sum_{\ell=1}^{L_i} [\mathbf{B_a}]_{j,i}$ becomes the aggregated sum of weight products along all paths from node $j$ to node $i$ regardless of path length. We denote this sum by

$$f_i (\mathbf{B_a}) \triangleq \sum_{\ell=0}^{L_i} \left[ \mathbf{B_a^\ell} \right]_i . \tag{53}$$

We note that the noise $\epsilon$ at any time $t$ is independent of the process that decides $\mathbf{B_a}$. Therefore, the expected value of $X_i$ under intervention $\mathbf{a}$ is

$$\mu_{i,\mathbf{a}} = \mathbb{E}_{\mathbf{a}} [X_i] = \mathbb{E} \left[ \sum_{\ell=0}^{L_i} \left\langle \left[ \mathbf{B_a^\ell} \right]_i, \epsilon \right\rangle \right] = \langle f_i (\mathbf{B_a}), \boldsymbol{\nu} \rangle . \tag{54}$$

∎

## D  Proof of Theorem 1 (Structure Learning)

The proof is divided into two parts. First, we show that $T_1 = \frac{32m^2}{\eta^2} \log\left(\frac{2N^2}{\delta}\right)$ is sufficient to identify the ancestors sets and a valid topological ordering with probability at least $1 - \delta$. Subsequently, we show that with a probability of at least $1 - \delta$, Lasso regression yields the desired estimates for the parent sets.

**Part 1: topological ordering.** In this part, we show that $\mathsf{de}(i) \subseteq \widehat{\mathsf{de}}(i)$. Based on this, we will see an efficient topological ordering would be natural. Recall the definition of $\widehat{\mathsf{de}}(i)$ and $\widehat{\mathsf{an}}(i)$ in (8) and (9), respectively, it is equivalent to show that for $i \in [N]$ with $L_i \geq 1$ all $j \in \mathsf{de}(i)$, we have $|\hat{\mu}_{j,\emptyset} - \hat{\mu}_{j,\{i\}}| > \frac{\eta}{2}$ and for all $j \in \mathsf{an}(i)$, we have $|\hat{\mu}_{j,\emptyset} - \hat{\mu}_{j,\{i\}}| \leq \frac{\eta}{2}$.

We note that the bounded noises that satisfy Assumption 4 are 1-sub-Gaussian. After sufficient exploration, the mean estimators $\hat{\mu}_{j,\emptyset}$ and $\hat{\mu}_{j,\emptyset}$ defined in (6) will be close enough to the true means

$\mu_{j,\emptyset}$ and $\mu_{j,\emptyset}$ with high probability, respectively. When the mean estimates are accurate enough, based on Assumption 5, we show that $\frac{\eta}{2}$ can be used to separate descendants from non-descendants based on the value of $|\hat{\mu}_{j,\emptyset} - \hat{\mu}_{j,i}|$. Based on the definition of $\hat{\mu}_{j,\emptyset}$ and $\hat{\mu}_{j,\emptyset}$ in (6), we have

$$\mathbb{P}\left(|\hat{\mu}_{j,\emptyset} - \mu_{j,\emptyset}| \geq \frac{\eta}{4}\right) = \mathbb{P}\left(|\hat{\mu}_{j,\emptyset} - \mu_{j,\emptyset}| \geq \frac{\eta}{4}\right) \tag{55}$$

$$\leq 2\exp\left(-\frac{2T_1 \frac{\eta^2}{16}}{4m^2}\right) \tag{56}$$

$$= \frac{\delta}{N^2} \, , \tag{57}$$

where (55) holds because Hoeffding's inequality and (57) holds since the definition of $T_1$ in (12).

Similarly, we have

$$\mathbb{P}\left(|\hat{\mu}_{j,\{i\}} - \mu_{j,\{i\}}| \geq \frac{\eta}{4}\right) \leq \frac{\delta}{2N^2} \, . \tag{58}$$

We define error events in which the mean estimates have a large error as follows.

$$\mathcal{E}_{\mathrm{TO}} \triangleq \left\{|\hat{\mu}_{j,\emptyset} - \mu_{j,\emptyset}| \leq \frac{\eta}{4} \text{ and } |\hat{\mu}_{j,\{i\}} - \mu_{j,\{i\}}| \leq \frac{\eta}{4}, \quad \forall j \in [N], i \in [N-1]\right\} . \tag{59}$$

By taking a union bound and leveraging (57)-(58), we obtain

$$\mathbb{P}(\mathcal{E}_{\mathrm{TO}}) \leq N^2 \times \frac{\delta}{N^2} = \delta \, . \tag{60}$$

Next, we prove that under event $\mathcal{E}_{\mathrm{TO}}^{\mathrm{c}}$, we can correctly identify $\mathsf{de}(i)$ for all node $i \in [N]$. If $j \in \mathsf{de}(i)$, to evaluate the gap $|\hat{\mu}_{j,\emptyset} - \hat{\mu}_{j,\{i\}}|$, we leverage the following relationship:

$$|\mu_{j,\emptyset} - \mu_{j,\{i\}}| = |(\hat{\mu}_{j,\emptyset} - \hat{\mu}_{j,\{i\}}) + (\mu_{j,\emptyset} - \hat{\mu}_{j,\emptyset}) + (\mu_{j,\{i\}} - \hat{\mu}_{j,\{i\}})| \tag{61}$$

$$\leq |\hat{\mu}_{j,\emptyset} - \hat{\mu}_{j,\{i\}})| + |\mu_{j,\emptyset} - \hat{\mu}_{j,\emptyset}| + |\mu_{j,\{i\}} - \hat{\mu}_{j,\{i\}}| \, , \tag{62}$$

where (62) is due to the triangle inequality. Thus, for all $i \in [N-1]$ and $j \in \mathsf{de}(i)$ we have

$$|\hat{\mu}_{j,\emptyset} - \hat{\mu}_{j,\{i\}}| \geq |\mu_{j,\emptyset} - \mu_{j,\{i\}}| - |\hat{\mu}_{j,\emptyset} - \mu_{j,\emptyset}| - |\hat{\mu}_{j,\{i\}} - \mu_{j,\{i\}}| \tag{63}$$

$$\overset{(59)}{>} \eta - \frac{\eta}{4} - \frac{\eta}{4} \tag{64}$$

$$= \frac{\eta}{2} \, . \tag{65}$$

On the other hand, when $j \notin \mathsf{de}(i)$, we have $\mu_{j,\emptyset} = \mu_{j,\{i\}}$, based on which we obtain

$$|\hat{\mu}_{j,\emptyset} - \hat{\mu}_{j,\{i\}}| = |\hat{\mu}_{j,\emptyset} - \mu_{j,\emptyset} + \hat{\mu}_{j,\{i\}} - \mu_{j,\{i\}}| \tag{66}$$

$$\leq |\hat{\mu}_{j,\emptyset} - \mu_{j,\emptyset}| + |\hat{\mu}_{j,\{i\}} - \mu_{j,\{i\}}| \tag{67}$$

$$\overset{(59)}{\leq} \eta/2 \, . \tag{68}$$

In conclusion, with probability $1 - \delta$, the estimates of descendants sets $\widehat{\mathsf{de}}(i) = \mathsf{de}(i)$ for node $i \in [N]$ with $L_i > 0$ with probability at least $1 - \delta$. Hence, with probability at least $1 - \delta$ the $\widehat{\mathsf{an}}(i)$ defined in (9) will be the best set we can find, that is $\widehat{\mathsf{an}}(i) \subseteq \mathsf{an}(i)$ for $i \in [N]$ and for $i \in \widehat{\mathsf{an}}(i) \setminus \mathsf{an}(i)$, we can infer that $i$ is a root node. Besides, the topological ordering $\hat{\pi}$ is valid.

**Part 2: Lasso regression.** In this part, we establish a sparsity property of the Lasso estimator in linear causal bandit, which is inspired by [33] and [34]. We consider the case when $T_2 \geq T_1$. The case for $T_2 < T_1$ can be analyzed similarly. We prove for fixed $i \in [N]$ as it is the same for all $i \in [N]$. When the ancestors sets and topological ordering are correct with probability at least $1 - \delta$ from Part 1. We define the time instance $T_3 = NT_1 + T_2 - T_1$ be the time that the Lasso estimators are calculated.

We show that when $T_2 \gtrsim d\log(N)$ and when ancestors sets and topological ordering are correct, with probability at least $1 - \delta$, the Lasso estimates satisfy the desired property. The proof consists of three steps. In the first step, we show that the Lasso estimates provide a bounded cardinality,

that is for $i \in [N]$ we have $|\widehat{\mathsf{pa}}(i)| \leq \kappa |\mathsf{pa}(i)|$. his step involves first bounding the support of the Lasso estimator by the empirical error, followed by bounding the empirical error itself. In the second step, we prove by contradiction that for $i \in [N]$ we prove $|\mathsf{pa}(i) \subseteq |\widehat{\mathsf{pa}}(i)|$. Finally, we take a union bounds on all nodes $i \in [N]$ to get the desired result.

*Step 1: Bounded cardinality.* In this step we show $|\widehat{\mathsf{pa}}(i)| \leq \kappa |\mathsf{pa}(i)|$. Recall that the Lasso estimator in the feature selection stage is defined as

$$[\widehat{\mathbf{B}}^{\text{Lasso}}]_i \triangleq \underset{\theta \in \mathbb{R}^{|\widehat{\mathsf{an}}(i)|}}{\text{argmin}} \left( \frac{1}{N_\emptyset(t)} \sum_{\tau \in [t], \mathbf{a}(\tau) = \emptyset} \left( X_i(\tau) - \theta^\top X_{\widehat{\mathsf{an}}(i)}(\tau) \right)^2 + \lambda_i \|\theta\|_1 \right). \tag{69}$$

We state some preliminary properties of the Lasso estimator and later show that our estimator satisfies the conditions for these properties.

**Property 1** (Restricted eigenvalues). *Let $\mathcal{S} \triangleq \text{supp}([\mathbf{B}]_i)$, define the cone*

$$\mathcal{C}(S) \triangleq \left\{ \theta \in \mathbb{R}^N \mid \text{supp}(\theta) = \widehat{\mathsf{an}}(i), \|\theta_{\mathcal{S}^c}\|_1 \leq 3 \|\theta_{\mathcal{S}}\|_1 \right\} . \tag{70}$$

*Then for all $\theta \in \mathbb{C}(S)$, there exists some positive constant $\kappa'$ such that the observation matrix $\mathbf{U}_{i,\widehat{\mathsf{an}}(i)}(t) \in \mathbb{R}^{t \times |\widehat{\mathsf{an}}(i)|}$ satisfies the condition*

$$\frac{\|\mathbf{U}_{i,\widehat{\mathsf{an}}(i)}(t)\theta\|_2^2}{t} \geq \kappa' \|\theta\|_2^2 . \tag{71}$$

**Property 2** (Column normalized). *We say that $\mathbf{U}_{i,\widehat{\mathsf{an}}(i)}(t)$ is column-normalized if*

$$\frac{\left\| [\mathbf{U}_{i,\widehat{\mathsf{an}}(i)}(t)]_j \right\|_2}{\sqrt{t}} \leq m , \quad \forall j \in \widehat{\mathsf{pa}}(i) . \tag{72}$$

**Lemma 2.** *Consider a $d$-sparse linear regression and assume that design matrix $\mathbf{U}_{i,0}(t) \in \mathbb{R}^{t \times |\widehat{\mathsf{an}}(i)|}$ satisfies Properties 1–2. Given the Lasso estimator with regularization parameter $\lambda = 4m\sqrt{\log(|\widehat{\mathsf{an}}(i)|)/t}$, then the following properties hold with probability at least $1 - \delta$.*

- *The estimation error under $\ell_1$-norm of any optimal solution $[\widehat{\mathbf{B}}^{\text{Lasso}}]_i$ satisfies [35, Theorem 7.13]:*

$$\left\| [\widehat{\mathbf{B}}^{\text{Lasso}}]_i - [\mathbf{B}]_i \right\|_1 \leq \frac{d}{\kappa'} \sqrt{\frac{2 \log(2|\widehat{\mathsf{an}}(i)|/\delta)}{t}} . \tag{73}$$

- *The mean square prediction error of any optimal solution $[\widehat{\mathbf{B}}^{\text{Lasso}}]_i$ satisfies [35, Theorem 7.20]:*

$$\frac{1}{t} \sum_{s=1}^{t} \left( X_{\widehat{\mathsf{an}}(i)}^\top(s) \left( [\widehat{\mathbf{B}}^{\text{Lasso}}]_i - [\mathbf{B}]_i \right) \right)^2 \leq \frac{9}{\kappa'} \cdot \frac{d \log(|\widehat{\mathsf{an}}(i)|/\delta)}{t} . \tag{74}$$

For $j \in \widehat{\mathsf{an}}(i)$ define the random variables

$$b_{i,j} \triangleq \frac{1}{N_\emptyset(t)} \sum_{\tau \in [T_3], \mathbf{a}(\tau) = \emptyset} X_j(t) \left( \epsilon_i(t) - \nu_i \right) . \tag{75}$$

Since $\left\| X_{\widehat{\mathsf{an}}(i)}(t) \right\|_\infty \leq m$, the Hoeffding's inequality for sub-Gaussian random variables implies

$$\mathbb{P}\left( \left| \sum_{\tau \in [T_3], \mathbf{a}(\tau) = \emptyset} X_j(t) \left( \epsilon_i(t) - \nu_i \right) \right| \geq \zeta \right) \leq 2 \exp\left( -\frac{\zeta^2}{2 N_\emptyset(t) m^2} \right). \tag{76}$$

We note that content we have $N_\emptyset(t) = T_2$. For $j \in [N]$, define $\mathcal{E}_{g_j}$ as the event that $g_j$ is contained in the interval close to mean value, i.e.,

$$\mathcal{E}_{b_{i,j}} = \left\{ |b_{i,j}| \leq \sqrt{\frac{2m^2 \log\left( \frac{4N|\widehat{\mathsf{an}}(i)|}{\delta} \right)}{T_2}} \right\} . \tag{77}$$

Based on the probability bounds in (76), we have

$$\mathbb{P}(\mathcal{E}_{b_{i,j}}) \leq \frac{\delta}{2N|\widehat{\mathsf{an}}(i)|} \ . \tag{78}$$

Accordingly, define

$$\mathcal{E}_{b_i} = \bigcup_{j \in \widehat{\mathsf{an}}(i)} \mathcal{E}_{b_{i,j}} \ . \tag{79}$$

By taking a union bound and leveraging (78), we obtain

$$\mathbb{P}\left(\mathcal{E}_{b_i}^{\mathsf{c}}\right) \leq \sum_{j \in \widehat{\mathsf{an}}(i)} \mathbb{P}\left(\mathcal{E}_{b_{i,j}}^{\mathsf{c}}\right) \leq |\widehat{\mathsf{an}}(i)| \times \frac{\delta}{2N|\widehat{\mathsf{an}}(i)|} \leq \frac{\delta}{2N} \ . \tag{80}$$

From the Karush-Kuhn-Tucker (KKT) condition of Lasso regression, the solution $[\widehat{\mathbf{B}}^{\mathrm{Lasso}}]_i$ satisfies

$$\frac{1}{T_2} \sum_{\tau \in [T_3], \mathbf{a}(\tau) = \emptyset} X_j(t) \left(X_i(t) - [\widehat{\mathbf{B}}^{\mathrm{Lasso}}]_i^\top X_{\widehat{\mathsf{an}}(i)}(t)\right) = \lambda \operatorname{sign}\left([\widehat{\mathbf{B}}^{\mathrm{Lasso}}]_i\right), \quad \text{if } [\widehat{\mathbf{B}}^{\mathrm{Lasso}}]_{i,j} \neq 0 \ , \tag{81}$$

$$\left| \frac{1}{T_2} \sum_{\tau \in [t], \mathbf{a}(\tau) = \emptyset} X_j(t) \left(X_i(t) - [\widehat{\mathbf{B}}^{\mathrm{Lasso}}]_i^\top X_{\widehat{\mathsf{an}}(i)}(t)\right) \right| \leq \lambda, \quad \text{if } [\widehat{\mathbf{B}}^{\mathrm{Lasso}}]_{i,j} = 0 \ . \tag{82}$$

Therefore, we have

$$\frac{1}{T_2} \sum_{\tau \in [t], \mathbf{a}(\tau) = \emptyset} X_j(t) \left([\mathbf{B}]_i^\top X_{\mathsf{pa}(i)}(t) - [\widehat{\mathbf{B}}^{\mathrm{Lasso}}]_i^\top X_{\mathsf{pa}(i)}(t)\right)$$

$$= \frac{1}{T_2} \sum_{\tau \in [t], \mathbf{a}(\tau) = \emptyset} X_j(t) \left(X_i(t) - [\widehat{\mathbf{B}}^{\mathrm{Lasso}}]_i^\top X_{\mathsf{pa}(i)}(t)\right) - \frac{1}{T_2} \sum_{\tau \in [t], \mathbf{a}(\tau) = \emptyset} X_j(t) \epsilon_i(t) \ . \tag{83}$$

Since $\lambda = m \sqrt{\frac{2 \log\left(\frac{4N|\widehat{\mathsf{an}}(i)|}{\delta}\right)}{T_2}}$, under event $\mathcal{E}_{g_i}$, we have

$$\left| \frac{1}{T_2} \sum_{\tau \in [T_3], \mathbf{a}(\tau) = \emptyset} X_j(t) \left([\mathbf{B}]_i^\top X_{\mathsf{pa}(i)}(t) - [\widehat{\mathbf{B}}^{\mathrm{Lasso}}]_i^\top X_{\mathsf{pa}(i)}(t)\right) \right| \geq \lambda/2, \text{ if } [\widehat{\mathbf{B}}^{\mathrm{Lasso}}]_{j,i} \neq 0 \ . \tag{84}$$

Based on (84), we can have the following lower bound

$$\frac{1}{T_2^2} \sum_{j \in \widehat{\mathsf{an}}(i)} \left( \sum_{\tau \in [T_3], \mathbf{a}(\tau) = \emptyset} X_j(t) \left([\mathbf{B}]_i^\top X_{\mathsf{pa}(i)}(t) - [\widehat{\mathbf{B}}^{\mathrm{Lasso}}]_i^\top X_{\mathsf{pa}(i)}(t)\right) \right)^2$$

$$\geq \sum_{j \in \operatorname{supp}([\mathbf{B}^{\mathrm{Lasso}}]_i)} \left( \frac{1}{T_2} \sum_{\tau \in [T_3], \mathbf{a}(\tau) = \emptyset} X_j(t) \left([\mathbf{B}]_i^\top X_{\mathsf{pa}(i)}(t) - [\widehat{\mathbf{B}}^{\mathrm{Lasso}}]_i^\top X_{\mathsf{pa}(i)}(t)\right) \right)^2 \tag{85}$$

$$\geq \frac{\lambda^2}{4} \left| \operatorname{supp}\left([\widehat{\mathbf{B}}^{\mathrm{Lasso}}]_i\right) \right| \ , \tag{86}$$

where (85) holds since $\widehat{\mathsf{an}}(i) \subseteq \operatorname{supp}([\mathbf{B}^{\mathrm{Lasso}}]_i)$ and (86) holds due to (84). On the other hand, define the uncentered empirical covariance matrix as

$$\widehat{\Sigma}_i = \frac{1}{T_2} \mathbf{U}_{i,\widehat{\mathsf{an}}(i)}^\top(T_3) \mathbf{U}_{i,\widehat{\mathsf{an}}(i)}(T_3) \ . \tag{87}$$

Let $\hat{\kappa}_{i,\max} = \sigma_{\max}\left(\widehat{\Sigma}_i\right)$. Then we have

$$\hat{\kappa}_{i,\max} = \sigma_{\max}\left(\mathbf{U}_{i,\widehat{\mathsf{an}}(i)}^\top(T_3) \mathbf{U}_{i,\widehat{\mathsf{an}}(i)}(T_3)/T_2\right) = \frac{1}{T_2} \sigma_{\max}\left(\mathbf{U}_{i,\widehat{\mathsf{an}}(i)}^\top(T_3) \mathbf{U}_{i,\widehat{\mathsf{an}}(i)}(T_3)\right) \ . \tag{88}$$

Now we need a high probability bound for $\phi_{\max}$. In order to proceed, we need upper and lower bounds for the maximum and minimum singular values of $\mathbf{U}_{i,\widehat{\mathsf{an}}(i)}(t)$. However, these bounds depend

on the number of non-zero rows of $\mathbf{U}_{i,\widehat{\mathsf{an}}(i)}(t)$ matrices, which equals to value of the random variable $N_{i,a(t)}(t)$. Let us define the weighted constant

$$\gamma_n \triangleq \max\left\{\eta m^2 \sqrt{T_2}, \eta^2 m^2\right\} . \tag{89}$$

We define the error events corresponding to the maximum and minimum singular values of $\mathbf{U}_{i,\widehat{\mathsf{pa}}(i)}(t)(T_3)$ as follows.

$$\mathcal{E}_i \triangleq \left\{ \sigma_{\min}\left(\mathbf{U}_{i,\widehat{\mathsf{an}}(i)}(T_3)\right) \leq \sqrt{\max\left\{0, T_2\tilde{\kappa}_{i,\emptyset,\min} - \gamma_n\right\}}. \right.$$

$$\left. \text{or } \sigma_{\max}\left(\mathbf{U}_{i,\widehat{\mathsf{an}}(i)}(t)\right) \geq \sqrt{T_2\tilde{\kappa}_{i,\emptyset,\max} + \gamma_n} \right\}, \tag{90}$$

**Lemma 3.** *[15, Lemma 8] The probability of the error events $\mathcal{E}_i(t)$ are upper bounded as*

$$\mathbb{P}(\mathcal{E}_i(t)) \leq d\exp\left(-\frac{3\eta^2}{16}\right) . \tag{91}$$

Thus, by setting $\eta = \sqrt{\frac{16}{3}\log\left(\frac{2dN}{\delta}\right)}$, we have with probability $1 - \frac{\delta}{2N}$ we have

$$\sigma_{\max}\left(\mathbf{U}_{i,\widehat{\mathsf{an}}(i)}(t)\right) < \sqrt{T_2\tilde{\kappa}_{i,\emptyset,\max} + \gamma_n} . \tag{92}$$

Hence, with probability $1 - \delta$ we have

$$\hat{\kappa}_{i,\max} = \sigma_{\max}\left(\mathbf{U}_{i,\widehat{\mathsf{an}}(i)}^\top \mathbf{U}_{i,\widehat{\mathsf{an}}(i)}/T_2\right) \tag{93}$$

$$= \frac{1}{T_2}\sigma_{\max}\left(\mathbf{U}_{i,\widehat{\mathsf{an}}(i)}^\top \mathbf{U}_{i,\widehat{\mathsf{an}}(i)}\right) \tag{94}$$

$$< \tilde{\kappa}_{i,\emptyset,\max} + \frac{\gamma_n}{T_2} \tag{95}$$

$$< \kappa_{\max} + \frac{\max\left\{\alpha m^2\sqrt{T_2}, \alpha^2 m^2\right\}}{T_2} \tag{96}$$

$$= \kappa_{\max} + m^2\max\left\{\sqrt{\frac{16}{3T_2}\log\left(\frac{2dN}{\delta}\right)}, \frac{16}{3T_2}\log\left(\frac{2dN}{\delta}\right)\right\} \tag{97}$$

$$= \kappa_{\max} + m^2\sqrt{\frac{16}{3T_2}\log\left(\frac{2dN}{\delta}\right)} , \tag{98}$$

where we use (92) in (95) and the last inequality is due to $T_2 \gtrsim d\log(N)$. Since $\kappa_{i,\max}$ has the natural upper bound $m^2$, we define

$$\kappa_0 \triangleq \min\left\{m^2, \kappa_{\max} + m^2\sqrt{\frac{16}{3T_2}\log\left(\frac{2dN}{\delta}\right)}\right\} . \tag{99}$$

Combined with (98) we know that

$$\hat{\kappa}_{i,\max} \leq \kappa_0 , \tag{100}$$

based on which we have

$$\frac{1}{T_2^2}\sum_{j\in\widehat{\mathsf{pa}}(i)}\left(\sum_{\tau\in[T_3],\mathbf{a}(\tau)=\emptyset} X_j(t)\left([\mathbf{B}]_i^\top X_{\mathsf{pa}(i)}(t) - [\widehat{\mathbf{B}}^{\mathsf{Lasso}}]_i^\top X_{\mathsf{pa}(i)}(t)\right)\right)^2 \tag{101}$$

$$= \frac{1}{T_2^2}\left(\mathbf{U}_{i,\widehat{\mathsf{an}}(i)}(T_3)[\mathbf{B}]_i - \mathbf{U}_{i,\widehat{\mathsf{an}}(i)}(T_3)[\widehat{\mathbf{B}}^{\mathsf{Lasso}}]_i\right)^\top \mathbf{U}_{i,\widehat{\mathsf{an}}(i)}(T_3)\mathbf{U}_{i,\widehat{\mathsf{an}}(i)}^\top(T_3)$$

$$\times \left(\mathbf{U}_{i,\widehat{\mathsf{an}}(i)}(T_3)[\mathbf{B}]_i - \mathbf{U}_{i,\widehat{\mathsf{an}}(i)}(T_3)[\widehat{\mathbf{B}}^{\mathsf{Lasso}}]_i\right) \tag{102}$$

$$\leq \kappa_0 \frac{1}{T_2}\|\mathbf{U}_{i,\widehat{\mathsf{an}}(i)}(T_3)[\widehat{\mathbf{B}}^{\mathsf{Lasso}}]_i - \mathbf{U}_{i,\widehat{\mathsf{an}}(i)}(T_3)[\mathbf{B}]_i\|_2^2 , \tag{103}$$

where (102) holds due to the matrix formulation of the equation and the definition of $\mathbf{U}_{i,\widehat{\mathsf{an}}(i)}(T_3)$ in (37), and (103) holds due to (100).

Combining all the results in (86) and (103), we find that with probability at least $1 - \frac{1}{N\delta}$,

$$\left|\operatorname{supp}\left([\mathbf{B}^{\text{Lasso}}]_i\right)\right| \leq \frac{4\kappa_0}{\lambda^2 T_2} \|\mathbf{U}_{i,\widehat{\mathsf{an}}(i)}[\mathbf{B}^{\text{Lasso}}]_i - \mathbf{U}_{i,\widehat{\mathsf{an}}(i)}[\mathbf{B}]_i\|_2^2 . \tag{104}$$

The speed of convergence of Lasso estimators depends on how rapidly the term $\|\mathbf{U}_{i,\widehat{\mathsf{an}}(i)}[\mathbf{B}^{\text{Lasso}}]_i - \mathbf{U}_{i,\widehat{\mathsf{an}}(i)}[\mathbf{B}]_i\|_2^2$ decreases. We now ensure $\mathbf{D}$ satisfies Property 1 with $\kappa = \kappa_{\min,\emptyset}/2$ when $T_2 \gtrsim d\log(|\widehat{\mathsf{an}}(i)|)$. The uncentered empirical covariance matrix defined in (87) satisfies

$$\mathbb{E}(\widehat{\Sigma}_i) = \operatorname{Cov}(X_{\widehat{\mathsf{an}}(i)}) = \Sigma_{i,\emptyset} . \tag{105}$$

We need the notion of restricted eigenvalue defined as follows.

**Definition 1.** *Given a symmetric matrix $\mathbf{H} \in \mathbb{R}^{d\times d}$, positive integer $k$, and $L > 0$, the restricted eigenvalue of $H$ is defined as*

$$\phi^2(H, k, L) \triangleq \min_{\mathcal{S}\subset[d], |\mathcal{S}|\leq k} \min_{\theta\in\mathbb{R}^d} \left\{ \frac{\langle\theta, \mathbf{H}\theta\rangle}{\|\theta_{\mathcal{S}}\|_1^2} : \theta \in \mathbb{R}^d, \|\theta_{\mathcal{S}^c}\|_1 \leq L \|\theta_{\mathcal{S}}\|_1 \right\} . \tag{106}$$

It is easy to see $\mathbf{U}_{i,\widehat{\mathsf{an}}(i)}(T_3)\Sigma_{i,\emptyset}^{-1/2}$ has independent sub-Gaussian rows with sub-Gaussian norm $\left\|\Sigma_{i,\emptyset}^{-1/2}X_{\widehat{\mathsf{an}}(i)}\right\|_{\psi_2} = \tilde{\kappa}_{i,\emptyset,\min}^{-1/2}$. If the population covariance matrix meets the restricted eigenvalue condition, then the empirical covariance matrix also satisfies this condition with high probability [36, Theorem 10]. Specifically, suppose the number of rounds in the exploration phase satisfies

$$T_2 \geq 4c_*c'\tilde{\kappa}_{i,\emptyset,\min}^{-2} \log(e|\widehat{\mathsf{an}}(i)|/c') , \tag{107}$$

for some $c_* \leq 2000$ and $c' = 10^4 d\tilde{\kappa}_{i,\emptyset,\max}^2/\phi^2(\Sigma, k, 9)$. Then the following condition holds

$$\mathbb{P}\left(\phi(\widehat{\Sigma}, k, 3) \geq \frac{1}{2}\phi(\Sigma, k, 9)\right) \geq 1 - 2\exp\left(-\frac{T_2}{4c_*\tilde{\kappa}_{i,\emptyset,\min}^{-1/2}}\right) . \tag{108}$$

Noting $\phi(\Sigma, k, 9) \geq \tilde{\kappa}_{i,\emptyset,\min}^{1/2}$, we subsequently get

$$\mathbb{P}\left(\phi^2(\widehat{\Sigma}, k, 3) \geq \frac{\tilde{\kappa}_{i,\emptyset,\min}}{2}\right) \geq 1 - 2\exp\left(-c_1 T_2\right) , \tag{109}$$

where $c_1 = \frac{1}{4c^*\tilde{\kappa}_{i,\min}^{-1/2}}$. This guarantees $\widehat{\Sigma}$ satisfies Property 1 in the appendix with $\kappa_0 = \frac{\tilde{\kappa}_{i,\emptyset,\min}}{2}$. It can be readily verified that Property 2 holds. Applying the in-sample prediction error bound in Lemma 2, we have with probability at least $1 - \frac{\delta}{N}$,

$$\frac{1}{T_3}\|\mathbf{U}_{i,\widehat{\mathsf{an}}(i)}(t)[\widehat{\mathbf{B}}^{\text{Lasso}}]_i - \mathbf{U}_{i,\widehat{\mathsf{an}}(i)}(t)[\mathbf{B}]_i\|_2^2 \leq \frac{18}{\tilde{\kappa}_{\min}} \cdot \frac{d\log(\frac{N|\widehat{\mathsf{an}}(i)|}{\delta})}{T_2} \leq \frac{18}{\kappa_{\min}} \cdot \frac{d\log(\frac{N|\widehat{\mathsf{an}}(i)|}{\delta})}{T_2} . \tag{110}$$

Putting (104) and (110) together, with probability at least $1 - \frac{2\delta}{N}$, we have

$$|\widehat{\mathsf{pa}}(i)| \leq |\operatorname{supp}([\widehat{\mathbf{B}}^{\text{Lasso}}]_i)| \leq \frac{9\hat{\kappa}|\mathsf{pa}(i)|}{\kappa_{\min}} = \kappa|\mathsf{pa}(i)| . \tag{111}$$

*Step 2: Containing* $\mathsf{pa}(i)$ *set.* This step is to verify the variable screening property of the Lasso estimator, that is $\operatorname{supp}([\widehat{\mathbf{B}}^{\text{Lasso}}]_i) \supseteq \operatorname{supp}([\mathbf{B}]_i)$. Since we set

$$T_2 > \frac{4d\log(N)}{\kappa_{\min}^2 \min_{j\in\operatorname{supp}([\mathbf{B}]_i)}|[\mathbf{B}]_{j,i}|^2} , \tag{112}$$

by using Lemma 2, it holds that with probability at least $1 - \frac{\delta}{2N}$,

$$\min_{j\in\operatorname{supp}([\mathbf{B}]_i)}|[\mathbf{B}]_{j,i}| > \|[\widehat{\mathbf{B}}^{\text{Lasso}}]_i - [\mathbf{B}]_i\|_2 \geq \|[\widehat{\mathbf{B}}^{\text{Lasso}}]_i - [\mathbf{B}]_i\|_\infty , \tag{113}$$

where in the last inequality we use the fact that $\|\cdot\|_2 \geq \|\cdot\|_\infty$.

Now we prove the variable screening property by contradiction. If there exists a $j$ such that $j \in \text{supp}([\mathbf{B}]_i)$ but $j \notin \text{supp}([\widehat{\mathbf{B}}^{\text{Lasso}}]_i)$, we have

$$\left| [\widehat{\mathbf{B}}^{\text{Lasso}}]_{j,i} - [\mathbf{B}]_{j,i} \right| = |[\mathbf{B}]_{j,i}| > \|[\widehat{\mathbf{B}}^{\text{Lasso}}]_i - [\mathbf{B}]_i\|_\infty . \tag{114}$$

On the other hand,

$$\left| [\widehat{\mathbf{B}}^{\text{Lasso}}]_{j,i} - \mathbf{B}_{j,i} \right| \leq \|[\widehat{\mathbf{B}}^{\text{Lasso}}]_i - [\mathbf{B}]_i\|_\infty , \tag{115}$$

which leads to a contradiction. Hence, we conclude that $\text{supp}([\widehat{\mathbf{B}}^{\text{Lasso}}]_i) \supseteq \text{supp}([\mathbf{B}]_i)$.

*Step 3: Union bounds.* By taking a Union bounds on $N$ nodes with probability at least $1 - 2\delta$, for all $i \in [N]$, we have

$$\mathsf{pa}(i) \subseteq \widehat{\mathsf{pa}}(i) , \quad \text{and} \quad |\widehat{\mathsf{pa}}(i)| \leq \kappa |\mathsf{pa}(i)| . \tag{116}$$

$\blacksquare$

# E  Proof of Theorem 2 (Intervention Design)

This proof leverages a technique initially introduced by [37] and further developed by [38] and [32] for contextual linear bandit setting. However, unlike their settings, we face stochastic environments and compound effects due to the causal structures. In the proof, we address the added uncertainty introduced by noise from the parent variables, which requires careful handling. In the proof of Theorem 2, we work with the estimate of the maximum in-degree, denoted by $\widehat{d} \triangleq \max_{i \in [N]} |\widehat{\mathsf{pa}}(i)|$ which, with probability at least $1 - 2\delta$, satisfies $\widehat{d} \leq \kappa d$, as proved in Theorem 1. However, to cover the proof for Corollary 1 and Corollary 2 where the proof needs to deal with $d$ and $d_e$, respectively, we use the notation $d$ to represent $\widehat{d}$, and $\widehat{\mathsf{pa}}(i)$ to represent $\mathsf{pa}(i)$ throughout. Besides, we work on the time instances from $T_3$ to $T$ for GA-LCB , to accommodate the known graph setting, we extend the proof across the entire time horizon, and prove the theorem for the entire time horizon $T$. The proof consists of four main parts: first, we demonstrate that the UCB width holds in Section E.1, second, we bound the cumulative width in Section E.2, then bound the time required for elimination in Section E.3, and finally, we bound the regret in Section E.4.

## E.1  Bounding UCB width

In this section, we begin by proving the following estimation error lemma, that is for node $i \in \widehat{\pi}$, if we can observe the contextual observation $X_{\mathsf{pa}(i),\mathbf{a}}$, a tighter contextual UCB width can be achieved, as shown in the following lemma.

**Lemma 4.** *For $i \in \widehat{\pi}$ and $\mathbf{a} \in \mathcal{A}$, define the true UCB width as*

$$\widehat{w}_{i,\mathbf{a}}(t) \triangleq \sum_{j \in \widehat{\mathsf{pa}}(i)} w_{j,\mathbf{a}} + \alpha \max_{\mathbf{a} \in \mathcal{A}} \|X_{\mathsf{pa}(i),\mathbf{a}}(t)\|_{[\mathbf{V}_{i,a_i}(t)]^{-1}} , \tag{117}$$

*where $\alpha = \sqrt{1/2 \log(2NT)} + \sqrt{d}$. With probability at least $1 - \frac{\delta}{T}$ for all $\mathbf{a} \in \mathcal{A}$ and $j \in \mathsf{an}(i)$ we have*

$$\left| \widehat{X}_{j,\mathbf{a}}(t) - X_{j,\mathbf{a}}(t) \right| \leq \widehat{w}_{j,\mathbf{a}}(t) . \tag{118}$$

*Proof:* We first provide a high probability bound on the exploration bonus $\|X_{\mathsf{pa}(i),\mathbf{a}}(t)\|^2_{[\mathbf{V}_{i,a_i}(t)]^{-1}}$ via Azuma's inequality and then prove this lemma via induction on the causal depth $L_i$.

**High probability bound.** Due to the statistical independence of the observation samples $\mathbf{U}_{i,a_i}(t)$ and $X_{\mathsf{pa}(i),\mathbf{a}}(t)$ for all $\mathbf{a} \in \mathcal{A}$, we have $\mathbb{E}[\boldsymbol{\epsilon}_i(t)] = \mathbf{0}$, where $\boldsymbol{\epsilon}_i(t) \triangleq (\epsilon_i(1), \cdots, \epsilon_i(t))^\top$. Hence, for

all $\mathbf{a} \in \mathcal{A}$ we have

$$\mathbb{P}\left(\left|\epsilon_i^\top \mathbf{U}_{i,a_i}(t)[\mathbf{V}_{i,a_i}(t)]^{-1} X_{\mathsf{pa}(i),\mathbf{a}}(t)\right| > \beta \max_{\mathbf{a} \in \mathcal{A}} \|X_{\mathsf{pa}(i),\mathbf{a}}(t)\|_{[\mathbf{V}_{i,a_i}(t)]^{-1}}\right) \tag{119}$$

$$\leq 2 \exp\left(-\frac{2\beta^2 \max_{\mathbf{a} \in \mathcal{A}} \|X_{\mathsf{pa}(i),\mathbf{a}}(t)\|^2_{[\mathbf{V}_{i,a_i}(t)]^{-1}}}{\left\|\mathbf{U}_{i,a_i}(t)[\mathbf{V}_{i,a_i}(t)]^{-1} X_{\mathsf{pa}(i),\mathbf{a}}(t)\right\|^2}\right) \tag{120}$$

$$\leq 2 \exp\left(-2\beta^2\right) \tag{121}$$

$$= \frac{\delta}{TN} , \tag{122}$$

where (120) holds since Azuma's inequality implies, and (121) is due to the following fact

$$\max_{\mathbf{a} \in \mathcal{A}} \|X_{\mathsf{pa}(i),\mathbf{a}}^\top(t)\|^2_{[\mathbf{V}_{i,a_i}(t)]^{-1}} \tag{123}$$

$$\geq \|X_{\mathsf{pa}(i),\mathbf{a}}^\top(t)\|^2_{[\mathbf{V}_{i,a_i}(t)]^{-1}} \tag{124}$$

$$= X_{\mathsf{pa}(i),\mathbf{a}}^\top(t)[\mathbf{V}_{i,a_i}(t)]^{-1}\left(\mathbf{I}_N + \mathbf{U}_{i,a_i}(t)^\top \mathbf{U}_{i,a_i}(t)\right)[\mathbf{V}_{i,a_i}(t)]^{-1} X_{\mathsf{pa}(i),\mathbf{a}}(t) \tag{125}$$

$$\geq X_{\mathsf{pa}(i),\mathbf{a}}^\top(t)[\mathbf{V}_{i,a_i}(t)]^{-1}\mathbf{U}_{i,a_i}(t)^\top \mathbf{U}_{i,a_i}(t)[\mathbf{V}_{i,a_i}(t)]^{-1} X_{\mathsf{pa}(i),\mathbf{a}}(t) \tag{126}$$

$$= \left\|\mathbf{U}_{i,a_i}(t)[\mathbf{V}_{i,a_i}(t)]^{-1} X_{\mathsf{pa}(i),\mathbf{a}}(t)\right\|^2 . \tag{127}$$

and we use $\beta = \sqrt{\frac{1}{2}\log\frac{NT}{\delta}}$ in (122), which corresponding to the first term in $\alpha$ we used for the UCB *width* in Theorem 2.

Next, we prove the Lemma 4 by induction.

**Base step:** $L_i = 1$. For node $i \in [N]$ with causal depth $L_i = 1$, we show that for all $\mathbf{a} \in \mathcal{A}$ with probability $1 - \frac{\delta}{TN}$ we have

$$\left|\widehat{X}_{i,\mathbf{a}}(t) - X_{i,\mathbf{a}}(t)\right| \leq \alpha \max_{\mathbf{a} \in \mathcal{A}} \|X_{\mathsf{pa}(i)}\|_{[\mathbf{V}_{i,a_i}(t)]^{-1}} . \tag{128}$$

We start by decomposing the left-hand side in (128) as follows.

$$\widehat{X}_{i,\mathbf{a}}(t) - X_{i,\mathbf{a}}(t) = [\mathbf{B_a}(t)]_i^\top \widehat{X}_{\mathsf{pa}(i),\mathbf{a}}(t) - [\mathbf{B_a}]_i^\top X_{\mathsf{pa}(i),\mathbf{a}}(t) \tag{129}$$

$$= [\mathbf{B_a}(t)]_i^\top \left(\widehat{X}_{\mathsf{pa}(i),\mathbf{a}}(t) - X_{\mathsf{pa}(i),\mathbf{a}}(t)\right)$$

$$+ \mathbf{D}_{i,a_i}^\top(t)\mathbf{U}_{i,a_i}(t)[\mathbf{V}_{i,a_i}(t)]^{-1} X_{\mathsf{pa}(i),\mathbf{a}}(t)$$

$$- [\mathbf{B_a}]_i^\top \left(\mathbf{I}_N + \mathbf{U}_{i,a_i}(t)^\top \mathbf{U}_{i,a_i}(t)\right)[\mathbf{V}_{i,a_i}(t)]^{-1} X_{\mathsf{pa}(i),\mathbf{a}}(t) \tag{130}$$

$$= \mathbf{D}_{i,a_i}^\top(t)\mathbf{U}_{i,a_i}(t)[\mathbf{V}_{i,a_i}(t)]^{-1} X_{\mathsf{pa}(i),\mathbf{a}}(t)$$

$$- [\mathbf{B_a}]_i^\top \left(\mathbf{I}_N + \mathbf{U}_{i,a_i}(t)^\top \mathbf{U}_{i,a_i}(t)\right)[\mathbf{V}_{i,a_i}(t)]^{-1} X_{\mathsf{pa}(i),\mathbf{a}}(t) \tag{131}$$

$$= \mathbf{D}_{i,a_i}^\top(t)\mathbf{U}_{i,a_i}(t)[\mathbf{V}_{i,a_i}(t)]^{-1} X_{\mathsf{pa}(i),\mathbf{a}}(t)$$

$$- \left([\mathbf{B_a}]_i^\top + [\mathbf{B_a}]_i^\top \mathbf{U}_{i,\mathbf{a}}(t)^\top \mathbf{U}_{i,a_i}(t)\right)[\mathbf{V}_{i,\mathbf{a}}(t)]^{-1} X_{\mathsf{pa}(i),\mathbf{a}}(t) \tag{132}$$

$$= \left(\mathbf{D}_{i,a_i}^\top(t) - [\mathbf{B_a}]_i^\top \mathbf{U}_{i,\mathbf{a}}(t)^\top\right)\mathbf{U}_{i,a_i}(t)[\mathbf{V}_{i,a_i}(t)]^{-1} X_{\mathsf{pa}(i),\mathbf{a}}(t)$$

$$- [\mathbf{B_a}]_i^\top [\mathbf{V}_{i,\mathbf{a}}(t)]^{-1} X_{\mathsf{pa}(i),\mathbf{a}}(t) \tag{133}$$

$$= \epsilon_i^\top(t)\mathbf{U}_{i,a_i}(t)[\mathbf{V}_{i,a_i}(t)]^{-1} X_{\mathsf{pa}(i),\mathbf{a}}(t)$$

$$- [\mathbf{B_a}]_i^\top [\mathbf{V}_{i,\mathbf{a}}(t)]^{-1} X_{\mathsf{pa}(i),\mathbf{a}}(t) , \tag{134}$$

where (131) is due to when $L_i = 1$ we have $\widehat{X}_{\mathsf{pa}(i),\mathbf{a}}(t) = X_{\mathsf{pa}(i),\mathbf{a}}(t) = \epsilon_{\mathsf{pa}(i)}$. Since $\|[\mathbf{B_a}]_i\| \leq \sqrt{d}$, we obtain

$$\left|\widehat{X}_{i,\mathbf{a}}(t) - X_{i,\mathbf{a}}(t)\right| \leq \left|\epsilon_i^\top(t)\mathbf{U}_{i,a_i}(t)[\mathbf{V}_{i,a_i}(t)]^{-1} X_{\mathsf{pa}(i),\mathbf{a}}\right|$$

$$+ \sqrt{d}\left\|[\mathbf{V}_{i,a_i}(t)]^{-1} X_{\mathsf{pa}(i),\mathbf{a}}(t)\right\| . \tag{135}$$

The right-hand side above decomposes the prediction error into a variance term (first term) and a bias term (second term). Next, we bound the second term in (135) as follows

$$\left\|[\mathbf{V}_{i,a_i}(t)]^{-1}X_{\mathsf{pa}(i),\mathbf{a}}(t)\right\| \tag{136}$$

$$= \sqrt{X_{\mathsf{pa}(i),\mathbf{a}}^{\top}(t)[\mathbf{V}_{i,a_i}(t)]^{-1}\mathbf{I}_N[\mathbf{V}_{i,a_i}(t)]^{-1}X_{\mathsf{pa}(i),\mathbf{a}}(t)} \tag{137}$$

$$\leq \sqrt{X_{\mathsf{pa}(i),\mathbf{a}}^{\top}(t)[\mathbf{V}_{i,a_i}(t)]^{-1}\left(\mathbf{I}_N + \mathbf{U}_{i,a_i}(t)^{\top}\mathbf{U}_{i,a_i}(t)\right)[\mathbf{V}_{i,a_i}(t)]^{-1}X_{\mathsf{pa}(i),\mathbf{a}}(t)} \tag{138}$$

$$= \|X_{\mathsf{pa}(i),\mathbf{a}}(t)\|_{[\mathbf{V}_{i,a_i}(t)]^{-1}}, \tag{139}$$

where (138) holds since the matrix $\mathbf{U}_{i,a_i}(t)^{\top}\mathbf{U}_{i,a_i}(t)$ is positive semidefinite.

Combining the bounds in (135), (122) and (139) completes proof for $L_i = 1$.

**Induction Step:** Assume that the property holds true for causal depths up to $L_i = k$. We show that it will also hold for $L_i = k + 1$. For this purpose, we start with the following expansion and apply the triangular inequality to find an upper bound for it. Similar to (135), we have

$$\widehat{X}_{i,\mathbf{a}}(t) - X_{i,\mathbf{a}}(t) \tag{140}$$

$$= [\mathbf{B}_{\mathbf{a}}(t)]_i^{\top}\left(\widehat{X}_{\mathsf{pa}(i),\mathbf{a}}(t) - X_{\mathsf{pa}(i),\mathbf{a}}(t)\right)$$

$$+ \epsilon_i^{\top}(t)\mathbf{U}_{i,a_i}(t)[\mathbf{V}_{i,a_i}(t)]^{-1}X_{\mathsf{pa}(i),\mathbf{a}}(t)$$

$$- [\mathbf{B}_{\mathbf{a}}]_i^{\top}[\mathbf{V}_{i,\mathbf{a}}(t)]^{-1}X_{\mathsf{pa}(i),\mathbf{a}}(t). \tag{141}$$

Using $\|[\mathbf{B}_{\mathbf{a}}]_i]\| \leq \sqrt{d}$ and triangle inequality, we obtain

$$\left|\widehat{X}_{i,\mathbf{a}}(t) - X_{i,\mathbf{a}}(t)\right| \leq \left|[\mathbf{B}_{\mathbf{a}}(t)]_i^{\top}\left(\widehat{X}_{\mathsf{pa}(i),\mathbf{a}}(t) - X_{\mathsf{pa}(i),\mathbf{a}}(t)\right)\right|$$

$$+ \left|\left(\mathbf{D}_{i,a_i}(t)^{\top} - [\mathbf{B}_{\mathbf{a}}]_i^{\top}\mathbf{U}_{i,a_i}(t)^{\top}\right)\mathbf{U}_{i,a_i}(t)[\mathbf{V}_{i,a_i}(t)]^{-1}X_{\mathsf{pa}(i),\mathbf{a}}(t)\right|$$

$$+ \sqrt{d}\left\|[\mathbf{V}_{i,a_i}(t)]^{-1}X_{\mathsf{pa}(i),\mathbf{a}}(t)\right\|, \tag{142}$$

where the last two terms can be bounded similarly as in the Base Step. It remains to bound the term

$$\left|[\mathbf{B}_{\mathbf{a}}(t)]_i^{\top}\left(\widehat{X}_{\mathsf{pa}(i),\mathbf{a}}(t) - X_{\mathsf{pa}(i),\mathbf{a}}(t)\right)\right| \leq \sum_{j \in \mathsf{pa}(i)}|[\mathbf{B}_{\mathbf{a}}(t)]_{j,i}|\left|\widehat{X}_{j,\mathbf{a}}(t) - X_{j,\mathbf{a}}(t)\right|. \tag{143}$$

From induction, we know that for all $j \in \mathsf{pa}(i)$, with probability $1 - \delta$ we have

$$\left|\widehat{X}_{j,\mathbf{a}}(t) - X_{j,\mathbf{a}}(t)\right| \leq \widehat{w}_{j,\mathbf{a}}(t). \tag{144}$$

Thus, we obtain

$$\left|\widehat{X}_{i,\mathbf{a}}(t) - X_{i,\mathbf{a}}(t)\right| \leq \sum_{j \in \mathsf{pa}(i)}\left|[\mathbf{B}(t)]_i^{\top}\right|w_{j,\mathbf{a}} + \alpha\|X_{\mathsf{pa}(i)}\|_{[\mathbf{V}_{i,a_i}(t)]^{-1}} \tag{145}$$

$$\leq \sum_{j \in \mathsf{pa}(i)}\widehat{w}_{j,\mathbf{a}} + \alpha\|X_{\mathsf{pa}(i)}\|_{[\mathbf{V}_{i,a_i}(t)]^{-1}}. \tag{146}$$

Hence, we conclude the proof. $\blacksquare$

## E.2 Bound sum of width of UCB

To bound the sum of UCB width, we first need to bound the sum of the exploration bonuses, as presented in the following lemma.

**Lemma 5.** *For all $i \in [N]$ with $L_i = \ell$, with probability at least $1 - \delta$ we have*

$$\sum_{t=1}^{T}\left\|X_{\mathsf{pa}(i)}(t)\right\|_{[\mathbf{V}_{i,a_i(t)}(t)]^{-1}} \leq 2\sqrt{5\frac{m_{\mathsf{pa},\ell}^2}{\log(m_{\mathsf{pa},\ell}^2/d+1)}\psi\log\left(\frac{m_{\mathsf{pa},\ell}^2}{2d}\psi+1\right)}, \tag{147}$$

*where $m_{\mathsf{pa},\ell} \triangleq \max_{i \in [N], L_i = \ell}\left\|X_{\mathsf{pa}(i)}\right\|$.*

*Proof:* This proof will use some intermediate steps from proof of Lemma 4. To proceed, we need the following lemma to bound the exploration bonus $\left\|X_{\mathsf{pa}(i)}(t)\right\|_{[\mathbf{V}_{i,a_i(t)}(t)]^{-1}}$ in terms of eigenvalues.

**Lemma 6.** *Let* $\{\lambda_{a_i(t),j}(t), j \in [N]\}$ *denote the ordered eigenvalues of* $\mathbf{V}_{i,a_i(t)}(t)$ *such that* $\lambda_{a_i(t),j}(t) \leq \lambda_{a_i(t),j+1}(t)$ *for* $j \in [N-1]$. *Then we have*

$$\left\|X_{\mathsf{pa}(i)}(t)\right\|^2_{[\mathbf{V}_{i,a_i(t)}(t)]^{-1}} \leq 10 \sum_{j=1}^{d} \frac{\lambda_{a_i(t),j}(t+1) - \lambda_{a_i(t),j}(t)}{\lambda_{a_i(t),j}(t)} . \tag{148}$$

*Proof:* The proof is similar to [37, Lemma 11] and [38, Lemma 2] with minor modifications to reflect bounded assumptions (the effect of $m$) and causal mechanisms (post-intervention distributions). We provide the proof for fixing $i \in [N]$ and $a_i = 0$ as it can be readily generalized to all cases $\mathbf{a} \in \mathcal{A}$. To proceed, we need the following lemmas.

**Lemma 7.** *[37, Lemma 17] For any* $\lambda_1 \geq \lambda_2$, $a \in \mathbb{R}$, *we have*

$$\begin{pmatrix} \lambda_1 & a \\ a & \lambda_2 \end{pmatrix} = \mathbf{U}^\top \begin{pmatrix} \lambda_1 + y & 0 \\ 0 & \lambda_2 - y \end{pmatrix} \mathbf{U} \tag{149}$$

*for some* $0 \leq y \leq \frac{a^2}{\lambda_1 - \lambda_2}$ *and some orthogonal matrix* $\mathbf{U}$.

**Lemma 8.** *Let* $\lambda_1 \geq \cdots \geq \lambda_d \geq 1$. *And let* $\nu_1, \geq \cdots \geq \nu_d$ *denote the effective eigenvalues of matrix* $\Delta(\lambda_1, \ldots, \lambda_d, 1 \cdots 1) + \mathbf{z} \cdot \mathbf{z}^\top$, *where* $\mathbf{z} \in \mathbb{R}^N$, $\|\mathbf{z}\| \leq m_{\mathsf{pa},L_i}$, *and* $\mathrm{supp}(\mathbf{z}) = [d]$. *There exists* $y_{h,j} \geq 0, 1 \leq h < j \leq d$, *and the following holds:*

$$\nu_j \geq \lambda_j , \tag{150}$$

$$\nu_j = \lambda_j + z_j^2 - \sum_{h=1}^{j-1} y_{h,j} + \sum_{h=j+1}^{d} y_{j,h} , \tag{151}$$

$$\sum_{h=1}^{j-1} y_{h,j} \leq z_j^2 , \tag{152}$$

$$\sum_{h=j+1}^{d} y_{j,h} \leq \nu_j - \lambda_j , \tag{153}$$

$$\sum_{j=1}^{d} \nu_j = \sum_{j=1}^{d} \lambda_j + \|\mathbf{z}\|^2 . \tag{154}$$

*If* $\lambda_h > \lambda_j + m^2_{\mathsf{pa},L_i}$ *then*

$$y_{h,j} \leq \frac{z_j^2 z_h^2}{\lambda_h - \lambda_j - m^2_{\mathsf{pa},L_i}} . \tag{155}$$

*Proof:* Clearly (151) implies (154) and (151); (152) imply (153). We prove the lemma by a recursive methods similar to induction on the dimension $d$.

**Base step:** We apply Lemma 7 to obtain the following transformation:

$$\Delta(\lambda_1, \ldots, \lambda_d, 1, \cdots, 1) + \mathbf{z} \cdot \mathbf{z}^\top \tag{156}$$

$$= \begin{pmatrix} \lambda_1 + z_1^2 & \cdots & z_1 z_{d-1} & z_1 z_d & \\ \vdots & \ddots & \vdots & \vdots & \\ z_1 z_{d-1} & \cdots & \lambda_{d-1} + z_{d-1}^2 & z_{d-1} z_d & \\ z_1 z_d & \cdots & z_{d-1} z_d & \lambda_d + z_d^2 & \\ & & & & \mathbf{I}_{N-d} \end{pmatrix} \tag{157}$$

$$= \mathbf{U}_d^\top \begin{pmatrix} \tilde{\lambda}_1 + z_1^2 & \cdots & z_1 z_{d-1} & 0 & \\ \vdots & \ddots & \vdots & \vdots & \\ z_1 z_{d-1} & \cdots & \tilde{\lambda}_{d-1} + z_{d-1}^2 & 0 & \\ 0 & \cdots & 0 & \tilde{\lambda}_d & \\ & & & & \mathbf{I}_{N-d} \end{pmatrix} \mathbf{U}_d , \tag{158}$$

where $\tilde{\lambda}_h = \lambda_h + y_{h,d}$, and $y_{h,d} \geq 0$, for $h = 1, \ldots, d-1$, and $\tilde{\lambda}_d = \lambda_d + z_d^2 - \sum_{h=1}^{d-1} y_{h,d}$. And we have the fact that $\sum_{h=1}^{d-1} y_h \leq z_d^2$. Thus $\tilde{\lambda}_j \geq \lambda_j$ for $j = 1, \ldots, d$.

**Induction Hypothesis:** Assume that we can apply the base step to dimension $\{d', \cdots, d\}$ and get $\{\mathbf{U}_{d'}, \cdots \mathbf{U}_d\}$.

**Induction Step:** We proceed by applying Lemma 7 with the upper left sub-matrix

$$\Delta\left(\tilde{\lambda}_1, \ldots, \tilde{\lambda}_{d'}\right) + [z_1, \cdots z_{d'}] \cdot [z_1, \cdots z_{d'}]^\top \tag{159}$$

$$= \begin{pmatrix} \tilde{\lambda}_1 + z_1^2 & \cdots & z_1 z_{d'} \\ \vdots & \ddots & \vdots \\ z_1 z_{d'} & \cdots & \tilde{\lambda}_{d'} + z_{d'}^2 \end{pmatrix} \tag{160}$$

$$= \mathbf{U}_{d'} \begin{pmatrix} \tilde{\lambda}_1 + z_1^2 & \cdots & z_1 z_{d'-1} & 0 \\ \vdots & \ddots & \vdots & \vdots \\ z_1 z_{d'-1} & \cdots & \tilde{\lambda}_{d'-1} + z_{d'-1}^2 & 0 \\ 0 & \cdots & 0 & \tilde{\lambda}'_d \end{pmatrix} \mathbf{U}_{d'} . \tag{161}$$

to tackle the $d'$ row and column. Then (155) follows from Lemma 7 since all elements in the diagonal are increasing with the induct step but no element grows by more than $m_{\mathsf{pa},\ell}^2$ since $\|\mathbf{z}\| \leq m_{\mathsf{pa},\ell}^2$. ■

Now, we are ready to prove the Lemma 5. From the definition of $\mathbf{V}_{i,a_i(t)}(t+1)$ we get

$$\mathbf{V}_{i,a_i(t)}(t+1) = \mathbf{V}_{i,a_i(t)}(t) + X_{\mathsf{pa}(i)}(t)X_{\mathsf{pa}(i)}^\top(t) \tag{162}$$

$$= \mathbf{U}(t)^\top \Delta\left(\lambda_1(t), \ldots, \lambda_d(t), 1, \cdots, 1\right) \mathbf{U}(t)$$
$$+ \mathbf{U}(t)^\top \widetilde{X}_{\mathsf{pa}(i)}(t) \widetilde{X}_{\mathsf{pa}(i)}^\top(t) \mathbf{U}(t) , \tag{163}$$

where in (163) we apply Lemma 8 on $\mathbf{V}_{i,a_i(t)}(t)$ and we define $\widetilde{X}_{\mathsf{pa}(i)}(t) = \mathbf{U}(t) X_{\mathsf{pa}(i)}(t)$. Thus, the effective eigenvalues of $\mathbf{V}_{i,a_i(t)}(t+1)$ are the effective eigenvalues of the matrix

$$\Delta\left(\lambda_1(t), \ldots, \lambda_d(t), 1 \cdots, 1\right) + \widetilde{X}_{\mathsf{pa}(i)}(t) \cdot \widetilde{X}_{\mathsf{pa}(i)}^\top(t) . \tag{164}$$

Using the notation of Lemma 8, let $\lambda_1 \geq \cdots \geq \lambda_d \geq 1$ be the eigenvalues of $\mathbf{V}_{i,a_i(t)}(t)$, $\{\nu_1, \ldots, \nu_d\}$ be the eigenvalues of $\mathbf{V}_{i,a_i(t)}(t+1)$, and $\mathbf{z} = \widetilde{X}_{\mathsf{pa}(i)}(t)$. For these choices we have the following property

$$X_{\mathsf{pa}(i)}^\top(t)[\mathbf{V}_{i,a_i(t)}(t)]^{-1} X_{\mathsf{pa}(i)}(t) = \sum_j \frac{z_j^2}{\lambda_j} . \tag{165}$$

To bound $z_j^2$, we use (151) in Lemma 8 and obtain

$$z_j^2 \leq \nu_j - \lambda_j + \sum_{h=1}^{j-1} y_{h,j} . \tag{166}$$

For $\lambda_h > \lambda_j + 3m_{\mathsf{pa},L_i}^2$ from (155) we obtain

$$y_{h,j} \leq \frac{z_j^2 z_h^2}{\lambda_h - \lambda_j - m_{\mathsf{pa},L_i}^2} \leq \frac{z_j^2 z_h^2}{2m_{\mathsf{pa},L_i}^2} , \tag{167}$$

and

$$\sum_{h:\lambda_h > \lambda_j + 3m_{\mathsf{pa}(i),L_i}^2} y_{h,j} \leq \frac{z_j^2}{2m_{\mathsf{pa},L_i}^2} \sum_{h:\lambda_h > \lambda_j + 3m_{\mathsf{pa},L_i}^2} z_h^2 \leq \frac{z_j^2}{2} , \tag{168}$$

since $\|\mathbf{z}\| \leq m_{\mathsf{pa},L_i}^2$. Hence, by combining (166) and (168) we get

$$z_j^2 \leq \nu_j - \lambda_j + \sum_{h=1}^{j-1} y_{h,j} \leq \nu_j - \lambda_j + z_j^2/2 + \sum_{h<j:\lambda_h \leq \lambda_j + 3m_{\mathsf{pa},L_i}^2} y_{h,j} , \tag{169}$$

and, subsequently

$$z_j^2 \le 2\left[\nu_j - \lambda_j + \sum_{h<j:\lambda_h \le \lambda_j + 3m_{\mathsf{pa},L_i}^2} y_{h,j}\right]. \tag{170}$$

If $\lambda_j \ge m_{\mathsf{pa},L_i}^2$ and $\lambda_h \le \lambda_j + 3m_{\mathsf{pa},L_i}^2$ when $\lambda_j \ge \lambda_h/4$ and we have

$$\sum_j \sum_{h<j:\lambda_h \le \lambda_j + 3m_{\mathsf{pa},L_i}^2} \frac{y_{h,j}}{\lambda_j} \le 4\sum_j \sum_{h<j:\lambda_h \le \lambda_j + 3m_{\mathsf{pa},L_i}^2} \frac{y_{h,j}}{\lambda_h} \tag{171}$$

$$\le 4\sum_h \sum_{j=h+1}^d \frac{y_{h,j}}{\lambda_h} \tag{172}$$

$$\le 4 \sum_{h:\lambda_h \ge 1} \frac{\nu_h - \lambda_h}{\lambda_h}, \tag{173}$$

where (171) holds as we relax the sum to include more terms and (173) holds due to (153). Thus, by applying (170) and (173) to (165), we obtain

$$\left\|X_{\mathsf{pa}(i)}(t)\right\|_{[\mathbf{V}_{i,a_i(t)}(t)]^{-1}}^2 = \sum_j \frac{z_j^2}{\lambda_j} \tag{174}$$

$$\le 2\sum_j \frac{\nu_j - \lambda_j}{\lambda_j} + 2\sum_j \sum_{h<j:\lambda_h \le \lambda_j + 3m_{\mathsf{pa},L_i}^2} \frac{y_{h,j}}{\lambda_j} \tag{175}$$

$$\le 2\sum_j \frac{\nu_j - \lambda_j}{\lambda_j} + 8\sum_h \frac{\nu_h - \lambda_h}{\lambda_h} \tag{176}$$

$$\le 10\sum_j \frac{\nu_j - \lambda_j}{\lambda_j}. \tag{177}$$

∎

So far, we have characterized a bound for confidence width. Next in terms of eigenvalues, we proceed to bound the sum in term of time instances the following lemma.

**Lemma 9.** *For all $i \in [N]$ with $L_i = \ell$ we have*

$$\sum_{t=1}^T \left\|X_{\mathsf{pa}(i)}(t)\right\|_{[\mathbf{V}_{i,a_i(t)}(t)]^{-1}}^2 \le 2\sqrt{5\frac{m_{\mathsf{pa},\ell}^2}{\log(m_{\mathsf{pa},\ell}^2/d+1)}T\log\left(\frac{m_{\mathsf{pa},\ell}^2}{2d}T+1\right)}. \tag{178}$$

*Proof:* The proof is similar to the proof of [37, Lemma 13], but modified to handle the difference between our algorithms and the difference of the weights. Lemma 6 implies

$$\sum_{t=1}^T \left\|X_{\mathsf{pa}(i)}(t)\right\|_{[\mathbf{V}_{i,a_i(t)}(t)]^{-1}}^2 = \sum_{t=1}^T \sqrt{10\sum_{j=1}^d \left(\frac{\lambda_{a_i(t),j}(t+1)}{\lambda_{a_i(t),j}(t)} - 1\right)}. \tag{179}$$

Since $a_i(t) \in \{0,1\}$ we have

$$\sum_{t=1}^T \left\|X_{\mathsf{pa}(i)}(t)\right\|_{[\mathbf{V}_{i,a_i(t)}(t)]^{-1}}^2 = \sum_{t=1}^T \mathbb{1}\{a_i(t) = 0\}\sqrt{10\sum_{j=1}^d \left(\frac{\lambda_{0,j}(t+1)}{\lambda_{0,j}(t)} - 1\right)}$$

$$+ \sum_{t=1}^T \mathbb{1}\{a_i(t) = 1\}\sqrt{10\sum_{j=1}^d \left(\frac{\lambda_{1,j}(t+1)}{\lambda_{1,j}(t)} - 1\right)}. \tag{180}$$

To bound the width sum, we leverage the closed-form solution of the following optimization problem. We define the instances at which node $i$ is intervened and not intervened as follows.

$$T_i(t) = \{\tau \in [t] \mid a_i(\tau) = 0\}, \quad T_i^*(t) = \{\tau \in [t] \mid a_i(\tau) = 1\}. \tag{181}$$

**Lemma 10.** *[38, Lemma 8] The solution to the following optimization problem with a set of time instances $\Psi \subset [T]$ and $C > d$*

$$\begin{cases} \max_{\{c_{tj} \in \mathbb{R}_+\}} & \sum_{t \in \Psi} \sqrt{\sum_{j=1}^{d} c_{tj}} \\ \text{s.t.} & \sum_{j=1}^{d} \prod_{t \in \Psi} (c_{tj} + 1) \leq C \end{cases}, \tag{182}$$

*is*

$$c_{tj} = \left(\frac{C}{d}\right)^{1/|\Psi|} - 1, \quad \forall\, t \in \Psi, j \in [d] . \tag{183}$$

We have the following property for the constraints.

$$\sum_{j=1}^{d} \prod_{t \in T_i(t)} \frac{\lambda_{0,j}(t+1)}{\lambda_{0,j}(t)} = \sum_{j=1}^{d} \lambda_{0,j}(T+1) \tag{184}$$

$$= \sum_{t \in T_i(t)} \left\| X_{\mathsf{pa}(i)}(t) \right\|^2 + d \tag{185}$$

$$\leq m_{\mathsf{pa},\ell}^2 |T_i(t)| + d , \tag{186}$$

and $$\sum_{j=1}^{d} \prod_{t \in T_i(t)} \frac{\lambda_{0,j}(t+1)}{\lambda_{0,j}(t)} \leq m_{\mathsf{pa},\ell}^2 |T_i^*(t)| + d , \tag{187}$$

where (186) is due to the definition of $m_{\mathsf{pa},\ell}$. Hence, by applying Lemma 10 in (180) we obtain

$$\sum_{t \in T_i(t)} \left\| X_{\mathsf{pa}(i)}(t) \right\|^2_{[\mathbf{V}_{i,a_i(t)}(t)]^{-1}} \leq |T_i(t)| \sqrt{10d} \sqrt{\left(\frac{m_{\mathsf{pa},\ell}^2}{d}|T_i(t)| + 1\right)^{\frac{1}{|T_i(t)|}} - 1}$$

$$+ |T_i^*(t)| \sqrt{10d} \sqrt{\left(\frac{m_{\mathsf{pa},\ell}^2}{d}|T_i^*(t)| + 1\right)^{\frac{1}{|T_i^*(t)|}} - 1} . \tag{188}$$

Next, we use the following lemma to bound (188).

**Lemma 11.** *If $\psi \geq 1$, the following inequality holds*

$$\left(\frac{m_{\mathsf{pa},\ell}^2}{d}\psi + 1\right)^{1/\psi} - 1 \leq \frac{m_{\mathsf{pa},\ell}^2/\hat{d}}{log(m_{\mathsf{pa},\ell}^2/\hat{d} + 1)} \frac{1}{\psi} log\left(\frac{m_{\mathsf{pa},\ell}^2}{\hat{d}}\psi + 1\right) . \tag{189}$$

*Proof:* Let $a \triangleq \frac{m_{\mathsf{pa},\ell}^2/\hat{d}}{log(m_{\mathsf{pa},\ell}^2/\hat{d}+1)}$, showing (189) is equivalent to showing

$$\frac{1}{\psi} log\left(\frac{m_{\mathsf{pa},\ell}^2}{d}\psi + 1\right) \leq log\left(1 + a\frac{1}{\psi}log\left(\frac{m_{\mathsf{pa},\ell}^2}{d}\psi + 1\right)\right) . \tag{190}$$

Define

$$g(\psi) \triangleq \frac{1}{\psi} log\left(\frac{m_{\mathsf{pa},\ell}^2}{d}\psi + 1\right) , \tag{191}$$

and

$$h(\psi) \triangleq log(1 + ag(\psi)) . \tag{192}$$

Since $h(\psi) > 0$. Showing (189) is equivalent to showing that for all $\psi \geq 1$ we have

$$\frac{g(\psi)}{h(\psi)} \leq 1 . \tag{193}$$

We show it in two steps. First, we show when $\psi = 1$, $\frac{g(\psi)}{h(\psi)}$ is less than 1. Second, we show the derivative of the function $\frac{g(\psi)}{h(\psi)}$ is negative for $\psi \geq 1$. To proceed, we use the following properties of $g(\psi)$ and $h(\psi)$.

$$\lim_{\psi \to \infty} g(\psi) = \lim_{\psi \to \infty} \frac{1}{\psi} \log \left( \frac{m_{\mathsf{pa},\ell}^2}{d} \psi + 1 \right) \tag{194}$$

$$= \lim_{\psi \to \infty} \frac{\frac{m_{\mathsf{pa}(i),\ell}^2}{d}}{\frac{m_{\mathsf{pa},\ell}^2}{d} \psi + 1} \tag{195}$$

$$= 0 \,, \tag{196}$$

where (195) is due to L'Hôpital's rule. Similarly, we have

$$\lim_{\psi \to \infty} h(\psi) = 0 \,. \tag{197}$$

Besides, the derivative of $g(\psi)$ is

$$g'(\psi) = \frac{\frac{m_{\mathsf{pa},\ell}^2}{d} \psi - \log \left( \frac{m_{\mathsf{pa},\ell}^2}{d} \psi + 1 \right) \left( \frac{m_{\mathsf{pa},\ell}^2}{d} \psi + 1 \right)}{\psi^2 \left( \frac{m_{\mathsf{pa},\ell}^2}{d} \psi + 1 \right)} \,. \tag{198}$$

Now, let

$$k(\psi) \triangleq \frac{m_{\mathsf{pa},\ell}^2}{d} \psi - \log \left( \frac{m_{\mathsf{pa},\ell}^2}{d} \psi + 1 \right) \left( \frac{m_{\mathsf{pa},\ell}^2}{d} \psi + 1 \right) \,. \tag{199}$$

The sign of $g'(\psi)$ is the same as $k(\psi)$ when $\psi > 0$. Furthermore, $k(1) \leq 0$ and

$$k'(\psi) = -\frac{m_{\mathsf{pa},\ell}^2}{d} \log \left( \frac{m_{\mathsf{pa},\ell}^2}{d} \psi + 1 \right) < 0 \,. \tag{200}$$

Thus, we can conclude that $k(\psi) < 0$ for all $\psi \geq 1$. Therefore, using (198) we find

$$g'(\psi) < 0 \,. \tag{201}$$

*Step 1:* When $\psi = 1$, we have

$$\frac{g(1)}{h(1)} = \frac{\log(m_{\mathsf{pa},\ell}^2/d + 1)}{\log(1 + a \log(m_{\mathsf{pa},\ell}^2/d + 1))} \,. \tag{202}$$

To show $\frac{g(1)}{h(1)} \leq 1$, we equivalently show

$$m_{\mathsf{pa},\ell}^2/d + 1 \leq 1 + a \log(m_{\mathsf{pa},\ell}^2/d + 1) \,, \tag{203}$$

which is obvious since $a = \frac{m_{\mathsf{pa},\ell}^2/\hat{d}}{\log(m_{\mathsf{pa},\ell}^2/\hat{d}+1)}$. Thus, we have

$$\frac{g(1)}{h(1)} \leq 1 \,. \tag{204}$$

*Step 2:* The gradient of $\frac{g(\psi)}{h(\psi)}$ can be calculated as

$$\left( \frac{g(\psi)}{h(\psi)} \right)' = \frac{g'(\psi)h(\psi) - h'(\psi)g(\psi)}{h^2(\psi)} \tag{205}$$

$$= \frac{g'(\psi) \log(1 + ag(\psi)) - \frac{ag'(\psi)}{1+ag(\psi)} g(\psi)}{h^2(\psi)} \tag{206}$$

$$= \frac{g'(\psi)}{h^2(\psi) \left( 1 + ag(\psi) \right)} \left( \left( 1 + ag(\psi) \right) \log(1 + ag(\psi)) - ag(\psi) \right) \,. \tag{207}$$

We have $g'(\psi) < 0$, $1 + ag(\psi) > 0$, and $h(\psi) > 0$. Thus, we only need to show that $m(\psi) = \big(1 + ag(\psi)\big)\log(1 + ag(\psi)) - ag(\psi) > 0$. We show this by noting that

$$\lim_{\psi \to \infty} m(\psi) = 0 \,, \tag{208}$$

and

$$m'(\psi) = ag'(\psi)\log(1 + ag(\psi)) + ag'(\psi) - ag'(\psi) = ag'(\psi)\log(1 + ag(\psi)) < 0 \,. \tag{209}$$

Thus, we conclude that $m(\psi) > 0$ for $\psi \geq 1$ and

$$\left(\frac{g(\psi)}{h(\psi)}\right)' < 0 \,. \tag{210}$$

Combine the results in (201), (204) and (210) we show the inequality in (189). ∎

Now, by applying lemma 11 to (188), we obtain

$$\sum_{t=1}^{T}\big\|X_{\mathsf{pa}(i)}(t)\big\|_{[\mathbf{V}_{i,a_i(t)}(t)]^{-1}} \leq \sqrt{10\frac{m_{\mathsf{pa},\ell}^2}{\log(m_{\mathsf{pa},\ell}^2/d+1)}|T_i(t)|\log\left(\frac{m_{\mathsf{pa},\ell}^2}{d}|T_i(t)|+1\right)}$$
$$+\sqrt{10\frac{m_{\mathsf{pa},\ell}^2}{\log(m_{\mathsf{pa},\ell}^2/d+1)}|T_i^*(t)|\log\left(\frac{m_{\mathsf{pa},\ell}^2}{d}|T_i^*(t)|+1\right)} \,. \tag{211}$$

Since the function $g(t) = \sqrt{10\frac{m_{\mathsf{pa},\ell}^2}{\log(m_{\mathsf{pa},\ell}^2/d+1)}t\log\left(\frac{m_{\mathsf{pa},\ell}^2}{d}t+1\right)}$ is concave function in $t$ and $|T_i(t)| + |T_i^*(t)| = T$, we have

$$\sum_{t=1}^{T}\big\|X_{\mathsf{pa}(i)}(t)\big\|_{[\mathbf{V}_{i,a_i(t)}(t)]^{-1}} \leq 2\sqrt{5\frac{m_{\mathsf{pa},\ell}^2}{\log(m_{\mathsf{pa},\ell}^2/d+1)}T\log\left(\frac{m_{\mathsf{pa},\ell}^2}{2d}T+1\right)} \,. \tag{212}$$

∎

### E.3 Bounding the time periods for elimination

Based on Lemma 9, we are able to bound the following lemma, which provide three important properties.

**Lemma 12.** *With probability at least $1 - 2\delta$, the following properties hold:*

1. *$\forall \mathbf{a} \in \mathcal{A}, t \in [T]$: $|\hat{\mu}_{N,\mathbf{a}}(t) - \mu_{N,\mathbf{a}}| \leq w_{N,\mathbf{a}}(t)$.*

2. *$\forall s \in [S]$: $\mathbf{a}^* \in \hat{\mathcal{A}}_s$.*

3. *$\forall \mathbf{a} \in \hat{\mathcal{A}}_s$: $\mu_{N,\mathbf{a}^*} - \mu_{N,\mathbf{a}} \leq m2^{3-s}$.*

*Proof:* We prove the properties when the properties in Theorem 1 hold. We prove the first property using the triangle inequality. Specifically,

$$\big\|X_{\mathsf{pa}(i)}(t)\big\|_{[\mathbf{V}_{i,a_i(t)}(t)]^{-1}} \leq \|\hat{\boldsymbol{\mu}}_{\mathsf{pa}(i)}(t)\|_{[\mathbf{V}_{i,a_i(t)}(t)]^{-1}} + \|X_{\mathsf{pa}(i)}(t) - \hat{\boldsymbol{\mu}}_{\mathsf{pa}(i)}(t)\|_{[\mathbf{V}_{i,a_i(t)}(t)]^{-1}} \tag{213}$$

$$\leq \|\hat{\boldsymbol{\mu}}_{\mathsf{pa}(i)}(t)\|_{[\mathbf{V}_{i,a_i(t)}(t)]^{-1}} + \sqrt{2}m_{\mathsf{pa},L_i}\lambda_{\min}^{-1/2}\big(\mathbf{V}_{i,a_i(t)}(t)\big) \,. \tag{214}$$

Therefore, using the definition in (19) and (117), we obtain

$$\widehat{w}_{N,\mathbf{a}}(t) \leq w_{N,\mathbf{a}}(t) \,. \tag{215}$$

Thus, by using Lemma 4 and summing over $t \in [T]$, with probability at least $1 - \delta$ we have

$$|\hat{\mu}_{N,\mathbf{a}}(t) - \mu_{N,\mathbf{a}}| = \left|\widehat{X}_{N,\mathbf{a}}(t) - X_{N,\mathbf{a}}(t)\right| \leq \widehat{w}_{N,\mathbf{a}}(t) \leq w_{N,\mathbf{a}}(t) \,. \tag{216}$$

The second property holds since when the first property holds, $\mathbf{a}^*$ always satisfies the condition in the line 16-17 in GA-LCB , i.e.

$$\mathrm{UCB}_{\mathbf{a}^*}(t) \geq \max_{\mathbf{a} \in \hat{\mathcal{A}}_s} \mathrm{UCB}_{\mathbf{a}}(t) - m2^{1-s}. \tag{217}$$

Next, we prove the third one by induction.

**Base Step:** When $s = 1$ it is obvious since we have $|\mu_{N,\mathbf{a}^*}| \leq m$ and $|\mu_{N,\mathbf{a}}| \leq m$.

**Induction Hypothesis:** Assume it holds for $s \leq s'$.

**Induction:** This indicates that for $s = s' + 1$ we have $\hat{\mathcal{A}}_s \subset \hat{\mathcal{A}}_{s-1}$. Additionally the condition in line 16 implies that $w_{\mathbf{a}}^{s-1} \leq m2^{-(s-1)}$ for $\mathbf{a} \in \hat{\mathcal{A}}_s$ and $w_{\mathbf{a}^*}^{s-1} \leq m2^{-(s-1)}$. Furthermore, the description of $\mathcal{A}_s$ in (22) implies

$$\mathrm{UCB}_{\mathbf{a}}^{(s-1)}(t) \geq \mathrm{UCB}_{\mathbf{a}^*}^{(s-1)}(t) - m2^{1-(s-1)} . \tag{218}$$

Thus, for any $\mathbf{a} \in \hat{\mathcal{A}}_{s-1}$ we have

$$\mu_{N,\mathbf{a}} \geq \mathrm{UCB}_{\mathbf{a}}^{(s-1)}(t) - m2 \cdot 2^{-(s-1)} \tag{219}$$

$$\geq \mathrm{UCB}_{\mathbf{a}^*(t)}^{(s-1)}(t) - m4 \cdot 2^{-(s-1)} \tag{220}$$

$$\geq \mu_{N,\mathbf{a}^*} - m4 \cdot 2^{-(s-1)} . \tag{221}$$

where (219) holds due to $\mathbf{a} \in \hat{\mathcal{A}}_{s-1}$, (220) holds due to (218) and (221) holds due to the definition of UCB. ∎

The next lemma provides an upper bound on the number of trials for which an alternative is chosen at the time $T_s$ for $s \in [S]$.

**Lemma 13.** *If we define $T_s$ as the time when $\hat{\mathcal{A}}_s$ is ended and $\hat{\mathcal{A}}_{s+1}$ starts, for all $s \in [S]$ we have*

$$
T_s \leq \frac{2^s}{m} \times \left( (\alpha + \sqrt{d}) \sum_{\ell=1}^{L} d^{\ell-1} 2 \sqrt{5 \frac{m_{\mathsf{pa},\ell}^2}{\log(m_{\mathsf{pa},\ell}^2/d + 1)} T_s \log \left( \frac{m_{\mathsf{pa},\ell}^2}{2d} T_s + 1 \right)} \right.
$$
$$
\left. + (\alpha + \sqrt{d}) 2\sqrt{2} \sum_{\ell=1}^{L} d^{\ell-1} m_{\mathsf{pa},\ell} \left( \sqrt{\frac{2}{\kappa_{\min}}} \sqrt{T_s} + 8\tau + 1 \right) \right) . \tag{222}
$$

*Proof:* Note that similar to (213), we apply triangle inequality to get

$$\|\hat{\boldsymbol{\mu}}_{\mathsf{pa}(i)}(t)\|_{[\mathbf{V}_{i,a_i(t)}(t)]^{-1}} \leq \|X_{\mathsf{pa}(i)}(t)\|_{[\mathbf{V}_{i,a_i(t)}(t)]^{-1}} + \|\hat{\boldsymbol{\mu}}_{\mathsf{pa}(i)}(t) - X_{\mathsf{pa}(i)}(t)\|_{[\mathbf{V}_{i,a_i(t)}(t)]^{-1}} \tag{223}$$

$$\leq \|X_{\mathsf{pa}(i)}(t)\|_{[\mathbf{V}_{i,a_i(t)}(t)]^{-1}} + \sqrt{2} m_{\mathsf{pa},L_i} \lambda_{\min}^{-1/2} \left( \mathbf{V}_{i,a_i(t)}(t) \right) . \tag{224}$$

*Step 1:* To proceed, we first bound the second term in (224): summation regarding the eigenvalues, in the following lemma.

**Lemma 14.** *We have the following upper bound for the eigenvalues*

$$\mathbb{E}\left[ \sum_{t=1}^{T} \lambda_{\min}^{-1/2} \left( \mathbf{V}_{i,a_i(t)}(t) \right) \right] \leq \sqrt{\frac{2}{\kappa_{\min}}} \sqrt{T} + 8\tau + 1 , \tag{225}$$

*where $\tau \triangleq \frac{\iota^2 m^4}{\kappa_{\min}^2}$ and $\iota \triangleq \sqrt{\frac{16}{3} \log(2dNT^2(T+1))}$.*

*Proof:* In order to proceed, we need upper and lower bounds on the maximum and minimum singular values of $\mathbf{U}_{i,a(t)}(t)$. However, these bounds depend on the number of non-zero rows of $\mathbf{U}_{i,a(t)}(t)$ matrices, which equals the values of the random variable $N_{i,a(t)}(t)$. Let us define the weighted constant

$$\gamma_n \triangleq \max \left\{ \iota m^2 \sqrt{n}, \iota^2 m^2 \right\} , \tag{226}$$

Then for every $t \in [T]$, and $n \in [t]$, we define the error events corresponding to the maximum and minimum singular values of $\mathbf{U}_i(t)$ and $\widetilde{\mathbf{U}}_i(t)$ as

$$\mathcal{E}_{i,n}(t) \triangleq \left\{ N_i(t) = n \quad \text{and} \quad \left\{ \sigma_{\min}\left(\mathbf{U}_i(t)\right) \leq \sqrt{\max\left\{0, n\kappa_{\min} - \gamma_n\right\}} \right.\right.$$

$$\left.\left. \text{or } \sigma_{\max}\left(\mathbf{U}_i(t)\right) \geq \sqrt{n\kappa_{\max} + \gamma_n} \right\} \right\}, \tag{227}$$

$$\mathcal{E}_{i,n}^*(t) \triangleq \left\{ N_i^*(t) = n \quad \text{and} \left\{ \sigma_{\min}\left(\mathbf{U}_i^*(t)\right) \leq \sqrt{\max\left\{0, n\kappa_{\min} - \gamma_n\right\}} \right.\right.$$

$$\left.\left. \text{or } \sigma_{\max}\left(\mathbf{U}_i^*(t)\right) \geq \sqrt{n\kappa_{\max} + \gamma_n} \right\} \right\}. \tag{228}$$

**Lemma 15.** *[15, Lemma 8] The probability of the error events $\mathcal{E}_{i,n}(t)$ and $\mathcal{E}_{i,n}^*(t)$ are upper bounded as*

$$\max\left\{\mathbb{P}(\mathcal{E}_{i,n}(t)), \mathbb{P}(\mathcal{E}_{i,n}^*(t))\right\} \leq d\exp\left(-\frac{3\iota^2}{16}\right). \tag{229}$$

Then we define the union error event $\mathcal{E}_{i,\cup}$ as

$$\mathcal{E}_{i,\cup} \triangleq \left\{ \exists\,(t,n) : t \in [T], n \in [t],\ \mathcal{E}_{i,n}(t) \text{ or } \mathcal{E}_{i,n}^*(t) \right\}. \tag{230}$$

By taking a union bound and using Lemma 3, we have

$$\mathbb{P}(\mathcal{E}_{i,\cup}) \leq \sum_{i=1}^{N}\sum_{t=1}^{T} 2d\exp\left(-\frac{3\iota^2}{16}\right) \tag{231}$$

$$\leq T(T+1)d\exp\left(-\frac{3\iota^2}{16}\right). \tag{232}$$

Now we turn back to $\mathbb{E}\left[\sum_{t=1}^{T}\lambda_i(t)\right]$ to analyze it under the complementary events $\mathcal{E}_{i,\cup}$ and $\mathcal{E}_{i,\cup}^{\mathrm{c}}$.

**Bounding term** $\mathbb{E}\left[\mathbb{1}\{\mathcal{E}_{i,\cup}\}\sum_{t=1}^{T}\lambda(t)\right]$. Since $\lambda_{\min}\left(\mathbf{V}_{i,a_i(t)}(t)\right) \geq 1$, we have the following upper bound.

$$\lambda_i(t) = \lambda_{\min}^{-1/2}(\mathbf{V}_{i,a_i(t)}(t)) \leq 1. \tag{233}$$

We have

$$\mathbb{E}\left[\mathbb{1}\{\mathcal{E}_{i,\cup}\}\sum_{t=1}^{T}\lambda_i(t)\right] \overset{(233)}{\leq} \mathbb{E}\left[\mathbb{1}\{\mathcal{E}_{i,\cup}\}T\right] = \mathbb{P}(\mathcal{E}_{i,\cup})T. \tag{234}$$

By setting $\iota = \sqrt{\frac{16}{3}\log(2dNT^2(T+1))}$, we obtain

$$\mathbb{E}\left[\mathbb{1}\{\mathcal{E}_{i,\cup}\}\sum_{t=1}^{T}\lambda_i(t)\right] \overset{(234)}{\leq} \mathbb{P}(\mathcal{E}_{i,\cup})T \overset{(232)}{\leq} \underbrace{\frac{NT(T+1)d}{\exp(\log(dNT^{5/2}(T+1)))}}_{=T^{-1}}T < 1. \tag{235}$$

**Bounding** $\mathbb{E}\left[\mathbb{1}\{\mathcal{E}_{i,\cup}^{\mathrm{c}}\}\sum_{t=1}^{T}\lambda_i(t)\right]$. Considering the event $\mathcal{E}_{i,\cup}^{\mathrm{c}}$, we can use the following bounds on the singular values

$$\sigma_{\min}\left(\mathbf{U}_{i,a_i(t)}(t)\right) \geq \sqrt{\max\left\{0, N_{i,a(t)}(t)\kappa_{\min} - \gamma_n\right\}}. \tag{236}$$

Thus, the target sum can be upper-bounded as

$$\mathbb{E}\left[\mathbb{1}\{\mathcal{E}_{i,\cup}^{\mathrm{c}}\}\sum_{t=1}^{T}\lambda(t)\right] = \mathbb{E}\left[\mathbb{1}\{\mathcal{E}_{i,\cup}^{\mathrm{c}}\}\sum_{t=1}^{T}\frac{1}{\sqrt{\sigma_{\min}^2\left(\mathbf{U}_{i,a(t)}(t)\right) + 1}}\right] \tag{237}$$

$$\leq \mathbb{E}\left[\sum_{t=1}^{T}\frac{1}{\sqrt{\max\{N_{i,a(t)}(t)\kappa_{\min} - \gamma_n\} + 1}}\right]. \tag{238}$$

It is noteworthy that the term in the sum in (238) has a critical point, and we bound the two regions separately. To this end, we define the function

$$n(x) \triangleq \frac{1}{\sqrt{\max\{0, x\kappa_{\min} - \gamma_n\} + 1}} , \quad x > 0 . \tag{239}$$

In order to analyze the behavior of the function $n$, we introduce $\tau \triangleq \frac{\iota^2 m^4}{\kappa_{\min}^2}$ as the critical point. Note that when $x \le \tau$, we have $x\kappa_{\min} < \gamma_n$. In this case,

$$n(x) = 1 , \tag{240}$$

which is an increasing function over the region. Now, we are ready to bound the last term

$$\mathbb{E}\left[\mathbb{1}\{\mathcal{E}_{i,\cup}^c\} \sum_{t=1}^T \lambda_i(t)\right] \le \mathbb{E}\left[\sum_{t=1}^T h(N_{i,a(t)}(t))\right] . \tag{241}$$

We define the set of time indices at which the chosen interventions are under-explored as

$$\mathcal{H}_i \triangleq \left\{t \in [T] \mid N_{i,\mathbf{a}(t)}(t) \le 4\tau\right\} . \tag{242}$$

It can be readily verified that $|\mathcal{H}_i| \le 8\tau$ since for node $i$ we have $a_i \in \{0,1\}$. Furthermore, when $x \in \mathcal{H}_i$, we have

$$h(x) = 1, \ x \le \tau . \tag{243}$$

Then we can bound the sum in (241) when $\mathcal{H}_i$ occurs as follows.

$$\mathbb{E}\sum_{t=1}^T \mathbb{1}\{t \in \mathcal{H}_i\}h(N_{i,a(t)}(t)) \le 8\tau . \tag{244}$$

Next, we only need to bound the remaining part when $t \notin \mathcal{H}_i$

$$\mathbb{E}\sum_{t=1}^T \mathbb{1}\{t \in \mathcal{H}_i^c\}h(N_{i,a(t)}(t)) . \tag{245}$$

Note that when $t \in \mathcal{H}_i^c$, we have $N_{i,a(t)}(t) > \tau$ and $n$ is a decreasing function. Hence,

$$\sum_{t=1}^T \mathbb{1}\{t \in \mathcal{H}_i^c\}h(N_{i,a(t)}(t)) \le \sum_{s=4\tau+1}^{N_i(T)+4\tau} n(s) + \sum_{s=4\tau+1}^{N_i^*(T)+4\tau} n(s) . \tag{246}$$

We bound the discrete sums through integrals and define

$$H_\tau(y) = \int_{x=4\tau}^y n(x)dx , \quad y \ge 4\tau . \tag{247}$$

Since $g(x)$ is a positive, non-increasing function, for any $k \in \mathbb{N}, k \ge 4\tau + 1$ we have

$$\sum_{s=4\tau+1}^k n(s) \le \int_{s=4\tau}^k n(s)\mathrm{d}s = H_\tau(k) . \tag{248}$$

Then, the sum in (246) is upper bounded by

$$\sum_{s=4\tau+1}^{N_i(T)+4\tau} n(s) + \sum_{s=4\tau+1}^{N_i^*(T)+4\tau} n(s) \le H_\tau(N_i(t) + 4\tau) + H_\tau(N_i^*(t) + 4\tau) . \tag{249}$$

Since $h(x)$ is positive and decreasing, and $H(y)$ is defined as an integral of the $n$ function with a positive first derivative and negative second derivative, it can be deduced that $H$ is a concave function. Thus, by applying the concavity property we have

$$H_\tau(N_i(t) + 4\tau) + H_\tau(N_i^*(t) + 4\tau) \le 2H_\tau\left(\frac{T}{2} + 4\tau\right) . \tag{250}$$

Next, we proceed to establish an upper bound on the function $H$ as follows.

$$H_\tau\left(\frac{T}{2} + 4\tau\right) = \int_{x=4\tau}^{\frac{T}{2}+4\tau} h(x)\mathrm{d}x \tag{251}$$

$$= \int_{x=4\tau}^{\frac{T}{2}+4\tau} \frac{1}{\sqrt{x\kappa_{\min} - \gamma_n + 1}}\mathrm{d}x . \tag{252}$$

$$= \int_{x=4\tau}^{\frac{T}{2}+4\tau} \frac{1}{\sqrt{x\kappa_{\min} - \gamma_n + 1}}\mathrm{d}x \tag{253}$$

$$= \frac{2\sqrt{\kappa_{\min}(\frac{T}{2} + 4\tau) - \gamma_n + 1} - 2\sqrt{4\kappa_{\min}\tau - \gamma_n + 1}}{\kappa_{\min}} \tag{254}$$

$$\leq \sqrt{\frac{2}{\kappa_{\min}}}\sqrt{T} , \tag{255}$$

where we have used $\int \frac{1}{\sqrt{ax-b}}dx = \frac{2\sqrt{ax-b}}{a} +$ constant . Combining the results in (235), (244) and (255), the final result for the bound is

$$\mathbb{E}\left[\sum_{t=1}^{T} \lambda_i(t)\right] \leq \sqrt{\frac{2}{\kappa_{\min}}}\sqrt{T} + 8\tau + 1 . \tag{256}$$

∎

*Step 2:* Now we bound the first term in second term in (224). We proceed by proving the following inequalities by induction. For $i \in \hat{\pi}$ with causal depth $L_i$, we prove

$$\sum_{t=1}^{T_s} w_{i,\mathbf{a}(\tau)}^{(s)} \leq (\alpha + \sqrt{d}) \sum_{\ell=1}^{L_i} d^{\ell-1}2\sqrt{5\frac{m_{\mathsf{pa},\ell}^2}{\log(m_{\mathsf{pa},\ell}^2/d+1)}T_s \log\left(\frac{m_{\mathsf{pa},\ell}^2}{2d}T_s + 1\right)}$$

$$+ (\alpha + \sqrt{d})2\sqrt{2} \sum_{\ell=1}^{L_i} d^{\ell-1}m_{\mathsf{pa},\ell}\left(\sqrt{\frac{2}{\kappa_{\min}}}\sqrt{T_s} + 8\tau + 1\right) . \tag{257}$$

**Base step:** when $i \in [N]$ and $L_i = 1$, we have

$$w_{i,\mathbf{a}(\tau)}^{(s)} \leq \|X_{\mathsf{pa}(i)}(t)\|_{[\mathbf{V}_{i,a_i}(t)]^{-1}} + 2\sqrt{2}m_{\mathsf{pa},L_i}\lambda_{\min}^{-1/2}\left(\mathbf{V}_{i,\mathbf{a}(\tau)}(t)\right) . \tag{258}$$

Hence, applying Lemma 9 and Lemma 14, we have

$$\sum_{t=1}^{T_s} w_{i,\mathbf{a}(\tau)}^{(s)} \leq 2\sqrt{5\frac{m_{\mathsf{pa},L_i}^2}{\log(m_{\mathsf{pa},L_i}^2/d+1)}T_s \log\left(\frac{m_{\mathsf{pa},L_i}^2}{2d}T_s + 1\right)}$$

$$+ 2\sqrt{2}m_{\mathsf{pa},L_i}\left(\sqrt{\frac{2}{\kappa_{\min}}}\sqrt{T_s} + 8\tau + 1\right) . \tag{259}$$

**Induction Step**: Assume (257) holds for nodes $i \in [N]$ with $L_i = k - 1$. Then for $i \in [N]$ and $L_i = k$, we have

$$
\sum_{t=1}^{T_s} w_{i,\mathbf{a}(\tau)}^{(s)} = \sum_{t=1}^{T_s} \sum_{j \in \mathsf{pa}(i)} w_{j,\mathbf{a}(\tau)}^{(s)}
$$

$$
+ \sum_{t=1}^{T_s} (\alpha + \sqrt{d})(\|X_{\mathsf{pa}(i)}(t)\|_{[\mathbf{V}_{i,a_i}(t)]^{-1}} + 2\sqrt{2} m_{\mathsf{pa},L_i} \lambda_{\min}^{-1/2}(\mathbf{V}_{i,\mathbf{a}(\tau)}(t))) \quad (260)
$$

$$
\leq d \times (\alpha + \sqrt{d}) \sum_{\ell=1}^{k-1} d^{\ell-1} \sqrt{10 \frac{m_{\mathsf{pa},\ell}^2}{\log(m_{\mathsf{pa},\ell}^2/d+1)} T_s \log\left(\frac{m_{\mathsf{pa},\ell}^2}{d} T_s + 1\right)}
$$

$$
+ d \times (\alpha + \sqrt{d}) 2\sqrt{2} \sum_{\ell=1}^{k-1} d^{\ell-1} m_{\mathsf{pa},\ell} \left(\sqrt{\frac{2}{\kappa_{\min}}} \sqrt{T_s} + 8\tau + 1\right)
$$

$$
+ (\alpha + \sqrt{d}) \sqrt{10 \frac{m_k^2}{\log(m_k^2/d+1)} T_s \log\left(\frac{m_{\mathsf{pa},\ell}^2}{d} T_s + 1\right)}
$$

$$
+ (\alpha + \sqrt{d}) 2\sqrt{2} m_k \left(\sqrt{\frac{2}{\kappa_{\min}}} \sqrt{T_s} + 8\tau + 1\right) \quad (261)
$$

$$
= (\alpha + \sqrt{d}) \sum_{\ell=1}^{L_i} d^{\ell-1} \sqrt{10 \frac{m_{\mathsf{pa},\ell}^2}{\log(m_{\mathsf{pa},\ell}^2/d+1)} T_s \log\left(\frac{m_{\mathsf{pa},\ell}^2}{d} T_s + 1\right)}
$$

$$
+ (\alpha + \sqrt{d}) 2\sqrt{2} \sum_{\ell=1}^{L_i} d^{\ell-1} m_{\mathsf{pa},\ell} \left(\sqrt{\frac{2}{\kappa_{\min}}} \sqrt{T_s} + 8\tau + 1\right) , \quad (262)
$$

where in (260) we use the definition of width, (261) holds due to induction and Lemma 14. Hence, we have proved the following inequality for the reward node.

$$
\sum_{t=1}^{T_s} w_{N,\mathbf{a}(\tau)}^{(s)} \leq (\alpha + \sqrt{d}) \sum_{\ell=1}^{L} d^{\ell-1} 2 \sqrt{5 \frac{m_{\mathsf{pa},\ell}^2}{\log(m_{\mathsf{pa},\ell}^2/d+1)} T_s \log\left(\frac{m_{\mathsf{pa},\ell}^2}{2d} T_s + 1\right)}
$$

$$
+ (\alpha + \sqrt{d}) 2\sqrt{2} \sum_{\ell=1}^{L} d^{\ell-1} m_{\mathsf{pa},\ell} \left(\sqrt{\frac{2}{\kappa_{\min}}} \sqrt{T_s} + 8\tau + 1\right) . \quad (263)
$$

By the condition of line 18 of GA-LCB , we have

$$
\sum_{t=1}^{T_s} w_{N,\mathbf{a}(\tau)}^{(s)}(\tau) \geq m 2^{-s} T_s . \quad (264)
$$

Combining (263) and (264) we obtain

$$
T_s \leq \frac{2^s}{m} \left( (\alpha + \sqrt{d}) \sum_{\ell=1}^{L} d^{\ell-1} 2 \sqrt{5 \frac{m_{\mathsf{pa},\ell}^2}{\log(m_{\mathsf{pa},\ell}^2/d+1)} T_s \log\left(\frac{m_{\mathsf{pa},\ell}^2}{2d} T_s + 1\right)} \right.
$$

$$
\left. + (\alpha + \sqrt{d}) 2\sqrt{2} \sum_{\ell=1}^{L} d^{\ell-1} m_{\mathsf{pa},\ell} \left(\sqrt{\frac{2}{\kappa_{\min}}} \sqrt{T_s} + 8\tau + 1\right) \right) . \quad (265)
$$

### E.4 Bounding the regret

Lastly, we define $\Psi_0$ as the time instance that GA-LCB performs UCB to select interventions. Now, we are ready to prove the final theorem. We start from the decomposition of the regret as follows

$$\mathbb{E}[\mathcal{R}(T)] = T\mu_{\mathbf{a}^*} - \sum_{t=1}^{T} \mu_{\mathbf{a}(t)} \tag{266}$$

$$= \sum_{t \in \Psi_0} \left(\mu_{\mathbf{a}^*} - \mu_{\mathbf{a}(t)}\right) + \sum_{s=1}^{S} \sum_{t=T_s-1+1}^{T_s} \left(\mu_{\mathbf{a}^*} - \mu_{\mathbf{a}(t)}\right) \tag{267}$$

$$\leq \frac{2m}{\sqrt{T}}|\Psi_0| + \sum_{s=1}^{S} 8m2^{-s}T_s \tag{268}$$

$$\leq \frac{2m}{\sqrt{T}}|\Psi_0| + \sum_{s=1}^{S} 8\left((\alpha + \sqrt{d})\sum_{\ell=1}^{L} d^{\ell-1}2\sqrt{5\frac{m_{\mathsf{pa},\ell}^2}{\log(m_{\mathsf{pa},\ell}^2/d+1)}T_s \log\left(\frac{m_{\mathsf{pa},\ell}^2}{2d}T_s + 1\right)}\right.$$

$$\left. + (\alpha + \sqrt{d})2\sqrt{2}\sum_{\ell=1}^{L} d^{\ell-1}m_{\mathsf{pa},\ell}\left(\sqrt{\frac{2}{\kappa_{\min}}}\sqrt{T_s} + 8\tau + 1\right)\right) \tag{269}$$

$$\leq 2m\sqrt{T} + 8(\alpha + \sqrt{d})S\sum_{\ell=1}^{L} d^{\ell-1}m_{\mathsf{pa},\ell}\left(2\sqrt{\frac{5}{\log(m_{\mathsf{pa},\ell}^2/d+1)}T \log\left(\frac{m_{\mathsf{pa},\ell}^2}{2d}T + 1\right)}\right.$$

$$\left. + 2\sqrt{2}\left(\sqrt{\frac{2}{\kappa_{\min}}}\sqrt{T} + 8\tau + 1\right)\right), \tag{270}$$

where (267) is due to $2^{-S} \leq \frac{1}{\sqrt{T}}$ and Lemma 13. Then we use the fact that for $d \geq 2$ we have

$$\sum_{\ell=1}^{L} d^{\ell-1} = \frac{d^L - 1}{d - 1} \leq 2d^{L-1} = \mathcal{O}(d^{L-1}). \tag{271}$$

Since $S = \mathcal{O}(\log T)$ and $m_{\mathsf{pa},\ell} \leq m$, we have

$$\mathbb{E}[\mathcal{R}(T)] \leq \tilde{\mathcal{O}}\left(d^{L-\frac{1}{2}}\sqrt{T}\right). \tag{272}$$

Finally, to get the regret bound in Theorem 2, we combine the results in Theorem 1 and (272), we conclude that with probability at least $1 - 3\delta$, we have

$$\mathbb{E}[\mathcal{R}(T)] \leq \tilde{\mathcal{O}}\left(d^{L-\frac{1}{2}}\sqrt{T} + RN + d\right). \tag{273}$$

## F  Notes on proof of Corollary 1 and Corollary 2

In the setting of Corollary 1, the causal graph $\mathcal{G}$ is known. We do not need to use GA-LCB-SL algorithm to do structure learning. Instead, we can directly employ the GA-LCB-ID algorithm with the true parent sets $\{\mathsf{pa}(i) \mid i \in [N]\}$. Hence, we can follow the proof of Theorem 2 to prove Corollary 1. The difference is that we will use the exact maximum in-degree $d$ instead of $\kappa d$ as we do not have that error due to the imperfect graph structure learning. Consequently, the regret bound simplifies, and we obtain the result stated in Corollary 1.

For Corollary 2, we have additional knowledge of the *effective* maximum in-degree $d_e$ and the causal depth $L_e$. If we define $\widehat{d}_e = \max i \in \mathsf{an}(N)|\widehat{\mathsf{pa}}(i)|$, from Theorem 1 we have $\widehat{d}_e \leq \kappa d_e$. At the same time, we can obtain a valid topological ordering $\widehat{\pi}_e$ that only contains $\widehat{\mathsf{an}}(N)$. Hence, we can follow the proof of Theorem 2 with maximum in-degree $d_e$ and causal depth $L_e$ and induction on $\pi_e$ to prove Corollary 2.

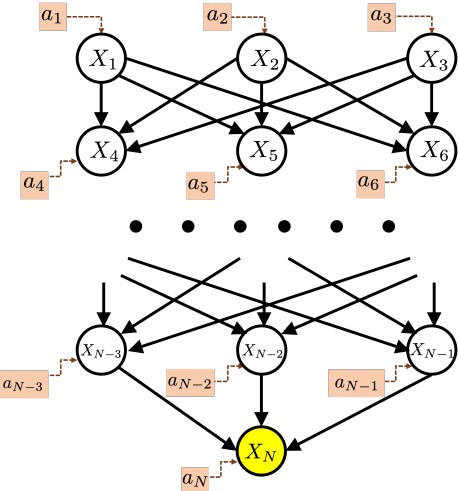

Figure 8: Sample hierarchical graph used in the proof of Theorem 3.

## G   Proof of Theorem 3 (Lower Bound)

Let $\Pi$ be the set of all policies on the set of stochastic bandit environments $\mathcal{I}$. The minimax regret is defined as

$$\inf_{\pi \in \Pi} \sup_{\mathcal{I}_0 \in \mathcal{I}} \mathbb{E}_{\pi, \mathcal{I}_0}[\mathcal{R}(T)] , \tag{274}$$

where $\mathbb{E}_{\pi, \mathcal{I}_0}[\mathcal{R}(T)]$ denotes the expected regret of policy $\pi$ on the bandit instance $\mathcal{I}_0$. We will consider a set $\tilde{\mathcal{I}}$, instead of $\mathcal{I}$, that contains two bandit instances. By definition of minimax regret, a lower bound for the regret of any policy on $\tilde{\mathcal{I}}$ also is a lower bound for the minimax regret since

$$\inf_{\pi \in \Pi} \sup_{\mathcal{I}_0 \in \mathcal{I}} \mathbb{E}_{\pi, \mathcal{I}_0}[\mathcal{R}(T)] \geq \inf_{\pi \in \Pi} \sup_{\mathcal{I}_0 \in \tilde{\mathcal{I}}} \mathbb{E}_{\pi, \mathcal{I}_0}[\mathcal{R}(T)] . \tag{275}$$

Following this property, the central idea of the proof is as follows. Consider two linear SEM causal bandit instances that differ by a small fraction and are hard to distinguish. At the same time, we can construct them to have different optimal interventions, indicating that a selection policy cannot incur small regret for both at the same time under the same data realization. Note that, the difference of the rewards, or equivalently the regrets, observed by these two bandit instances under the same intervention can be computed by tracing the effect of the differing edge parameter over all the paths that end at the reward node. We carefully build graphs to maximize the number of such paths for given $d$ and $L$. In this section, we provide details of these steps.

We consider the hierarchical graph as depicted in Figure 8, which consists of $L$ layers each with $d$ nodes. Adjacent layers are fully connected. There exists a final layer with one node fully connected to layer $L$. We label the $j$-th node on layer $\ell$ by $X_{(\ell-1)d+j}$ and the reward node by $X_N$.

We consider two linear SEM causal bandit instances $I$ and $\bar{I}$ that share the same graph $\mathcal{G}$. $I$ is parameterized by $I \triangleq \{\mathbf{B}, \mathbf{B}^*, \epsilon\}$ and $\bar{I}$ is parameterized by $I \triangleq \{\bar{\mathbf{B}}, \bar{\mathbf{B}}^*, \epsilon\}$. For each instance $I \in \tilde{\mathcal{I}}$ and for edges $(i \rightarrow j) \in \mathcal{E}$ with causal depth $L_i = 0$ or $L_i > 1$, let the weights of all observational edges be $m_{\mathbf{B}}$, and all interventional edges be $m_{\mathbf{B}} - \delta$ where $\delta \in (0, m_{\mathbf{B}})$ will be determined later. In other words, for $i, j \in [N]$, if $L_i = 0$ or $L_i > 1$ and $(i \rightarrow j) \in \mathcal{E}$, then

$$[\mathbf{B}]_{i,j} = [\bar{\mathbf{B}}]_{i,j} = m_{\mathbf{B}} , \qquad \text{and} \qquad [\mathbf{B}^*]_{i,j} = [\bar{\mathbf{B}}^*]_{i,j} = m_{\mathbf{B}} - \delta . \tag{276}$$

Let noise terms follow the standard Gaussian distribution for nodes except the first layer, i.e., $\epsilon_i \sim \mathcal{N}(0, 1)$ for all $i \in [N], i > d$. Furthermore, we let the noise in the first layer follow a Gaussian distribution $\mathcal{N}(1, 1)$ for all $i \in [d]$.

The only difference between instances in $I$ and $\bar{I}$ is in the weights for the nodes with causal depth $L_i = 1$. Note that nodes are labeled from $d + 1$ to $2d$. We have for $i \in [d+1, 2d]$ and $j \in [d]$

$$[\mathbf{B}]_{i,d+j} = [\bar{\mathbf{B}}^*]_{i,d+j} = m_{\mathbf{B}}, \qquad \text{and} \qquad [\mathbf{B}^*]_{i,d+j} = [\bar{\mathbf{B}}]_{i,j} = m_{\mathbf{B}} - \delta . \tag{277}$$

Next, consider a fixed bandit policy $\pi$ that generates the following filtration over time

$$\mathcal{F}_t \triangleq \{\mathbf{a}(1), X(1), \ldots, \mathbf{a}(t), X(t)\} . \tag{278}$$

The decision of $\pi$ at time $t$ is $\mathcal{F}_{t-1}$-measurable. Accordingly, define $\mathbb{P}_t$ and $\bar{\mathbb{P}}_t$ as the probability measures induced by $\mathcal{F}_t$ by $t$ rounds of interaction between $\pi$ and the two bandit instances $I$ and $\tilde{I}$. When it is clear from context, we use the shorthand terms $\mathbb{P}$ and $\bar{\mathbb{P}}$ for $\mathbb{P}_T$ and $\bar{\mathbb{P}}_T$, respectively. We will show that $\pi$ cannot suffer small regret in both instances at the same time and under the same filtration $\mathcal{F}_T$.

By Lemma 1, since all the elements of observational and interventional weights are non-negative, the optimal intervention is the one that maximizes the value of each entry of $\mathbf{B_a}$ and $\bar{\mathbf{B}}_{\mathbf{a}}$. The optimal action between two bandit instances only differs in nodes with $L_j = 2$. This means optimal intervention should include node $j$ if elements of $[\mathbf{B}]_j$ are $m_{\mathbf{B}}$ and not include $j$ otherwise, for $j \in [d+1, 2d]$. As a result, we have $\mathbf{a}_I^* = \emptyset$ and $\mathbf{a}_{\tilde{I}}^* = [d+1, 2d]$. Define $\mathcal{E}_{\text{lb}}^j$ as the event in which the decision on node $d$ is sup-optimal at least $\frac{T}{2}$ times after $T$ rounds on bandit instance $I$, i.e.,

$$\mathcal{E}_{\text{lb}}^j \triangleq \left\{ N_{d+j}^*(T) \geq \frac{T}{2} \right\} , \quad \text{for } j \in [d] . \tag{279}$$

We note that the event $\mathcal{E}_{\text{lb}}^j$ is defined on the $\sigma$-algebra defined by the filtration $\mathcal{F}_t$, that induces both $\mathbb{P}_t$ and $\bar{\mathbb{P}}_t$. We compute the expected instantaneous regret when node $i \in [d+1, 2d]$ is chosen sub-optimal in the first bandit instance and the total regret is the summation over these nodes. Note that each path passes node that node $i$ contributes to the expected regret. Furthermore, since every weight is positive, in $\mathcal{I}$, we have when the intervention on node $\{d+1, \cdots, 2d\}$ is chosen to be suboptimal, the impact on the average regret is $\delta m_{\mathbf{B}}^{L-1} d^{L-1}$ since there are $d^{L-2}$ paths of length $L-1$ from node $j$ to $N$ and the difference between weights as $\delta$ is multiplied with a $m_{\mathbf{B}}$ factor for every edge along a path. Then, by the definition of $\mathcal{E}_{\text{lb}}$, we have

$$\mathbb{E}_{\mathbb{P}}[\mathcal{R}(t)] = \mathbb{E}_{\mathbb{P}} \left[ \sum_{t=1}^{T} r(t) \right] \tag{280}$$

$$= \mathbb{E}_{\mathbb{P}} \left[ \sum_{t=1}^{T} \sum_{j \in [d, 2d]} \mathbb{1}\{j \notin \mathbf{a}(t)\} \delta m_{\mathbf{B}}^{L-1} d^{L-1} \right] \tag{281}$$

$$\geq \sum_{j=1}^{d} \mathbb{P}(\mathcal{E}_{\text{lb}}^j) \frac{T}{2} \delta m_{\mathbf{B}}^{L-1} d^{L-1} , \tag{282}$$

where (281) holds as we break down the regret and (282) holds due to the definition of $\mathcal{E}_{\text{lb}}^j$ in (279).

Similarly, for $\bar{I}$, each node $\{d+1, \cdots, 2d\}$ that is not intervened, it will occur at least $\delta m_{\mathbf{B}}^{L-1} d^{L-1}$ regret. Applying the same steps as in (280),-(282), we obtain

$$\mathbb{E}_{\bar{\mathbb{P}}}[\mathcal{R}(t)] = \mathbb{E}_{\bar{\mathbb{P}}} \left[ \sum_{t=1}^{T} r(t) \right] \tag{283}$$

$$\geq \mathbb{E}_{\bar{\mathbb{P}}} \left[ \sum_{t \in [T]} \sum_{j \in [d, 2d]} \mathbb{1}\{j \in \mathbf{a}(t)\} \delta m_{\mathbf{B}}^{L-1} d^{L-1} \right] \tag{284}$$

$$\geq \sum_{j=1}^{d} \bar{\mathbb{P}}(\mathcal{E}_{\text{lb}}^{j;\text{c}}) \frac{T}{2} \delta m_{\mathbf{B}}^{L-1} d^{L-1} . \tag{285}$$

By combining (282) and (285) we have

$$\mathbb{E}_{\mathbb{P}}[\mathcal{R}(t)] + \mathbb{E}_{\bar{\mathbb{P}}}[\mathcal{R}(t)] \geq \frac{T}{2} \delta m_{\mathbf{B}}^{L-1} d^{L-1} \sum_{j=1}^{d} [\mathbb{P}(\mathcal{E}_{\text{lb}}^j) + \bar{\mathbb{P}}(\mathcal{E}_{\text{lb}}^{j;\text{c}})] . \tag{286}$$

Next, we characterize a lower bound on $\mathbb{P}(\mathcal{E}_{\text{lb}}) + \bar{\mathbb{P}}(\mathcal{E}_{\text{lb}}^{\text{c}})$, which involves the Kullback-Leibler (KL) divergence between $\mathbb{P}$ and $\bar{\mathbb{P}}$, denoted by $D_{\text{KL}}(\mathbb{P} \| \bar{\mathbb{P}})$. For this purpose, we leverage the following theorem.

**Theorem 4** (Bretagnolle-Huber inequality). *Let $\mathbb{P}$ and $\bar{\mathbb{P}}$ be probability measures on the same measurable space $(\Omega, \mathcal{F})$ and let $A \in \mathcal{F}$ be an arbitrary event. Then,*

$$\mathbb{P}(A) + \bar{\mathbb{P}}(A^{\mathrm{c}}) \geq \frac{1}{2} \exp(-\mathrm{D}_{\mathrm{KL}}(\mathbb{P} \parallel \bar{\mathbb{P}})) . \tag{287}$$

By invoking Theorem 4, from (286) we obtain

$$\mathbb{E}_{\mathbb{P}}[\mathcal{R}(t)] + \mathbb{E}_{\bar{\mathbb{P}}}[\mathcal{R}(t)] \geq \frac{T}{2} \, \delta m_{\mathbf{B}}^{L-1} d^{L-1} \sum_{j=1}^{d} [\mathbb{P}(\mathcal{E}_{\mathrm{lb}}^{j}) + \bar{\mathbb{P}}(\mathcal{E}_{\mathrm{lb}}^{j,\mathrm{c}})] \tag{288}$$

$$\geq \frac{T}{4} \, \delta m_{\mathbf{B}}^{L-1} d^{L-1} \, d \exp(-\mathrm{D}_{\mathrm{KL}}(\mathbb{P} \parallel \bar{\mathbb{P}})) . \tag{289}$$

It remains to compute $\exp(-\mathrm{D}_{\mathrm{KL}}(\mathbb{P} \parallel \bar{\mathbb{P}}))$ to conclude our proof, for which we leverage the following result.

**Lemma 16.** *The KL divergence between $\mathbb{P}$ and $\bar{\mathbb{P}}$, the probability measures induced by $\mathcal{F}_t$ on $I$ and $\tilde{I}$, is equal to*

$$\mathrm{D}_{\mathrm{KL}}(\mathbb{P} \parallel \bar{\mathbb{P}}) = T d^2 (1+d) \delta^2 . \tag{290}$$

*Proof:* Note that a Bayesian network factorizes as

$$\mathbb{P}(X_1, \ldots, X_N) = \prod_{i=1}^{N} \mathbb{P}(X_i \mid X_{\mathsf{pa}(i)}) . \tag{291}$$

Additionally, the two bandit instances differ only in the mechanism of the first layer. Then, $\mathrm{D}_{\mathrm{KL}}(\mathbb{P} \parallel \bar{\mathbb{P}})$ can be decomposed as

$$\mathrm{D}_{\mathrm{KL}}(\mathbb{P} \parallel \bar{\mathbb{P}}) = \sum_{i=1}^{N} \mathrm{D}_{\mathrm{KL}}\big(\mathbb{P}(X_i \mid X_{\mathsf{pa}(i)}) \parallel \bar{\mathbb{P}}(X_i \mid X_{\mathsf{pa}(i)})\big) \tag{292}$$

$$= \sum_{j=1}^{d} \mathrm{D}_{\mathrm{KL}}\big(\mathbb{P}(X_{d+j}) \parallel \bar{\mathbb{P}}(X_{d+j}) \mid X_{\mathsf{pa}(d+j)}\big) . \tag{293}$$

Hence, we only need to analyze $\mathrm{D}_{\mathrm{KL}}(\mathbb{P}(X_{d+j}) \parallel \bar{\mathbb{P}}(X_{d+j}) \mid X_{\mathsf{pa}(d+j)})$ under two cases: (i) when node $(d+j)$ is observed, and (ii) node $(d+j)$ is intervened. We have that

$$X_{j+d} \sim \begin{cases} \mathcal{N}\big(m_{\mathbf{B}} \sum_{i=1}^{d} X_i, 1\big) , & \text{under } \mathbb{P} \text{ when } j+d \notin a \\ \mathcal{N}\big((m_{\mathbf{B}} - \delta) \sum_{i=1}^{d} X_i, 1\big) , & \text{under } \mathbb{P} \text{ when } j+d \in a \\ \mathcal{N}\big((m_{\mathbf{B}} - \delta) \sum_{i=1}^{d} X_i, 1\big) , & \text{under } \bar{\mathbb{P}} \text{ when } j+d \notin a \\ \mathcal{N}\big(m_{\mathbf{B}} \sum_{i=1}^{d} X_i, 1\big) , & \text{under } \bar{\mathbb{P}} \text{ when } j+d \in a \end{cases} . \tag{294}$$

By noting that

$$\mathrm{D}_{\mathrm{KL}}\Big(\mathcal{N}\big(m_{\mathbf{B}} \sum_{i=1}^{d} X_i, 1\big) \parallel \mathcal{N}\big((m_{\mathbf{B}} - \delta) \sum_{i=1}^{d} X_i, 1\big)\Big) \tag{295}$$

$$= \mathrm{D}_{\mathrm{KL}}\Big(\mathcal{N}\big((m_{\mathbf{B}} - \delta) \sum_{i=1}^{d} X_i, 1\big) \parallel \mathcal{N}\big(m_{\mathbf{B}} \sum_{i=1}^{d} X_i, 1\big)\Big) \tag{296}$$

$$= \frac{\delta^2 (\sum_{i=1}^{d} X_i)^2}{2} , \tag{297}$$

from (294) we obtain that for $j \in [d]$

$$D_{\text{KL}}(\mathbb{P}(X_{j+d}) \,\|\, \bar{\mathbb{P}}(X_{d+j})|X_{\mathsf{pa}(j+d)})$$

$$= \sum_{t \in [T]: j+d \notin \mathbf{a}(t)} D_{\text{KL}}(\mathcal{N}(m_{\mathbf{B}} \sum_{i=1}^{d} X_i, 1), \mathcal{N}((m_{\mathbf{B}} - \delta) \sum_{i=1}^{d} X_i, 1)) \tag{298}$$

$$+ \sum_{t \in [T]: j+d \in \mathbf{a}(t)} D_{\text{KL}}(\mathcal{N}((m_{\mathbf{B}} - \delta) \sum_{i=1}^{d} X_i, 1), \mathcal{N}(m_{\mathbf{B}} \sum_{i=1}^{d} X_i, 1)) \tag{299}$$

$$= N_1^*(T) \, \mathbb{E} \frac{\delta^2 (\sum_{i=1}^{d} X_i)^2}{2} + (T - N_1^*(T)) \, \mathbb{E} \frac{\delta^2 (\sum_{i=1}^{d} X_i)^2}{2} \tag{300}$$

$$= T(d + d^2)\delta^2 \,. \tag{301}$$

$$\blacksquare$$

By applying Lemma 16 on (289) and setting $\delta = \frac{1}{\sqrt{d^2(1+d)T}}$, we obtain

$$\max\{\mathbb{E}_{\mathbb{P}}[\mathcal{R}(t)], \mathbb{E}_{\bar{\mathbb{P}}}[\mathcal{R}(t)]\} \geq \frac{1}{2}(\mathbb{E}_{\mathbb{P}}[\mathcal{R}(t)] + \mathbb{E}_{\bar{\mathbb{P}}}[\mathcal{R}(t)]) \tag{302}$$

$$\geq \frac{T}{8} \, \delta m_{\mathbf{B}}^{L-1} d^{L-1} \, d \exp(-2T(d+d^2)\delta^2) \tag{303}$$

$$= \frac{\exp(-2)}{8} \, m_{\mathbf{B}}^{L-1} \frac{d^{L-1}}{\sqrt{d+1}} \, \sqrt{T} \,. \tag{304}$$

Hence, the policy $\pi$ incurs a regret $\Omega(d^{L-\frac{3}{2}}\sqrt{T})$ in at least one of the two bandit instances.

