# OpenReview forum: "Linear Causal Bandits: Unknown Graph and Soft Interventions"
_NeurIPS.cc/2024/Conference — NeurIPS 2024 poster_

### Official Review · Reviewer_2rLb · 2024-07-11

**Soundness:** 4
**Presentation:** 4
**Contribution:** 3
**Rating:** 6
**Confidence:** 3

**Summary:**

The paper considered a specific setting in the causal bandits problem where 1) the graph is unknown, 2) the causal model is linear , and 3) the action set consists of $2^{N}$ soft interventions.
To tackle the problem, the author proposed an algorithm that first learns the causal structure and then uses UCB-based approaches to find the best action and minimize the regret. The regret of the algorithm is analyzed for both cases: with and without the knowledge of the graph.

**Strengths:**

1- The paper considers a novel setting where the causal graph in unknown and SEM is linear.

2- The flow of the paper is clear, and each part of the proposed algorithm is thoroughly discussed.

3- The upper bound of GA-LCB-ID, when the graph is known, demonstrates an improvement compared to related work. Also, it is closed to the minimax lower bound.

4-  GA-LCB-ID, in the main case when the causal graph is unknown, achieves a reasonable upper bound.

5- The related work is comprehensively discussed in the introduction.

**Weaknesses:**

1- The algorithm assumes that the causal graph does not contain latent variables.

2- The algorithm requires access to identifiability parameters, which is an unrealistic assumption.

3- The set of interventions is limited, for each node, we only have one soft intervention.

**Questions:**

Can you provide a discussion for the above points?

1- I expected in the regret bound, the identifiability parameter would take part similar to previous work in an unknown setting. What is the intuition behind excluding it in this setting?

2- I didn’t understand why you took an expectation in Theorem 2. The theorem should show that with high probability, the regret is less than a certain bound. Why is it necessary to take an expectation? Note that you defined regret with implicit expectation over the randomness of rewards in Equation 5.

**Limitations:**

Yes.

---

> ### Author Rebuttal · Authors · 2024-08-06
>
> **Comment:** The algorithm assumes that the causal graph does not contain latent variables. The algorithm requires access to identifiability parameters, which is an unrealistic assumption.
>
> **Response:** We thank the reviewer for the thoughtful comment.  In principle, we agree with the reviewer about the importance of relaxing these assumptions. However, we note that these are standard assumptions in the current causal bandit literature (e.g.,  [14,17,19,22]). The primary reason is that causal bandits are not fully understood even under such assumptions. Causal bandit literature has been evolving towards removing as many assumptions as possible about graph topology and interventional distribution. This is the first paper that has dropped both under general structures and interventional models (with the remaining piece being extending linear SEMs to non-linear SEMs). However, we certainly agree that next, it is also important to dispense with assumptions on latent variables and identifiable parameters.
>
> **Comment:** The set of interventions is limited, for each node, we only have one soft intervention.
>
> **Response:** Thanks for the note. For presentation clarity, we have focused on the binary intervention model. However, the algorithms can be readily extended beyond binary interventions where we can show that the regret will scale as $\sqrt{KT}$ when $K$ is the number of interventions per node. In the final version we can use the additional one page to provide a brief discussion on the extension to $K$ interventions per node. Importantly, we note that expanding the intervention space will not impose additional costs for identifying the causal structure, as one soft intervention is sufficient. We also note that under $do$ intervention, the regret scales with $\sqrt{KT}$ where $K$ is the number of possible $do$ interventions. This means that under a significantly more relaxed intervention model, we can achieve the same regret that $do$ interventions achieve.
>
> Questions:
>
> **Comment:** I expected in the regret bound, the identifiability parameter would take part similar to previous work in an unknown setting. What is the intuition behind excluding it in this setting?
>
> **Response:** We thank the reviewer for this question. If we define $R=(m / \eta)^2$ and incorporate it into the regret, the term $N$ in the regret order will appear as $RN$. We will include this in the updated version. The $RN$ term arises from the step of identifying the ancestor relationship. Notably, a similar term is observed in related literature [19,22], where the term $R$ is coupled with the degree as they only need to identify $pa(N)$. Besides, in previous work, $K$ represents the number of distinct interventions, and we will have similar behavior when generalized to multiple soft interventions.
>
> **Comment:** I didn’t understand why you took an expectation in Theorem 2. The theorem should show that with high probability, the regret is less than a certain bound. Why is it necessary to take an expectation? Note that you defined regret with implicit expectation over the randomness of rewards in Equation 5.
>
> **Response**: We thank the reviewer for the sharp observation. Having the expectation is a typo, and we will remove it.

---

> > ### Comment · Reviewer_2rLb · 2024-08-09
> >
> > Thanks for your response. I also have another question. How do you compare your algorithm and Corollary 1 with the Naive approaches (e.g., the ucb-based algorithm in classic bandit) on a specific graph like the following graph. X1 -> X2 -> .... -> Xn. Also for all $i$, Xi is the parent of Xn. It's a chain graph where all variables are parent of Xn.

---

> ### Author Response · Authors · 2024-08-09
>
> We thank the reviewer for this comparison question. A chain graph is a **special** simple graph with a maximum in-degree of $d=1$ and a maximum causal depth of $L=N-1$.
>
> **Regret Comparison:** For UCB-based algorithms in the classic bandit setting **without knowing** the graph,  the regret scales as $\tilde{\mathcal{O}}(\sqrt{|\mathcal{A}| T}) = \tilde{\mathcal{O}}(\sqrt{2^N T})$. In contrast when the the graph is known, the regret reduces **significantly** and it scales as  $\tilde{\mathcal{O}}(\sqrt{T})$ (e.g. Corollary 1). The reason for the higher regret of the UCB-based algorithm without knowing the graph is that due to the lack of graph structure, the algorithm treats each set of interventions as an independent arm. This means that there are $2^N$ possible intervention sets, leading to the regret mentioned above.
>
> **Computational Cost:** Here we compare the computational cost of three algorithms: UCB-1 (a UCB-based algorithm for the classic bandit problem), LinSEM-UCB [14] (a UCB-based algorithm for causal bandits), and our proposed algorithm. The UCB-1 algorithm incurs a computational cost of $\mathcal{O}(2^N)$ because it needs to select the best intervention from $\mathcal{A}$ with $|\mathcal{A}|=2^N$. The LinSEM-UCB algorithm, designed for causal bandits, has a computational cost of $\mathcal{O}(N \cdot 2^{2N})$, as each UCB iteration requires maximization over the vertices of an $N$-dimensional hypercube, with each vertex having a computational cost of $\mathcal{O}(N)$. Additionally, it selects the best intervention from $\mathcal{A}$. In contrast, our algorithm has a computational cost of $\mathcal{O}(N |\mathcal{A}_s|)$, where $\mathcal{A}_s$ represents the set of possible optimal interventions at time $t$ after elimination, with $|\mathcal{A}_s| \leq 2^N$ and $|\mathcal{A}_s| \rightarrow 1$ as $t \rightarrow \infty$. This is because the computational cost of the confidence width calculation in equation (19) is $\mathcal{O}(1)$ (recall that $d=1$), and the UCB only requires the estimated mean value (of order $\mathcal{O}(N)$) and $N$ confidence widths. Consequently, our algorithm has a slightly higher computational cost at the beginning but becomes more efficient as $|\mathcal{A}_s|$ decreases to $\frac{2^N}{N}$, which can be achieved in a few elimination steps.
>
> Overall, our approach achieves a substantial reduction in regret while maintaining a comparable computational cost to UCB-1 in the context of this simple graph.
>
> Finally, we emphasize that as the graph topology becomes more complex, the UCB-based algorithms further degrade. For example, in hierarchical graphs, the intervention set size is $2^{dL+1}$, leading to a regret of $\tilde{\mathcal{O}}(\sqrt{2^{dL+1} T})$, compared to our algorithm’s $\tilde{\mathcal{O}}(\sqrt{d^{L-\frac{1}{2}} T})$.

---

> > ### Comment · Reviewer_2rLb · 2024-08-09
> >
> > Sorry, it seems you misunderstood my example. My example was a chain graph where all variables also are parents of Xn (reward node). So, the graph has $2(n-1)$ edges (if I'm not wrong, $d = N -1$ and $L = N-1$)

---

> > > ### Author Response · Authors · 2024-08-10
> > >
> > > Thank you for the clarification. To lay some context for discussion, we would like to address this in the dichotomy of instance-dependent versus class-level regret analysis. The former analyzes the performance of a policy on a specific bandit instance while the latter captures the performance over a class of instances. In this context, for any causal bandit algorithm (including ours), one can discuss
> > > - Instance-dependent regret analysis, which involves analyzing a policy for a specific causal bandit instance (e.g., with a specific graph topology).
> > > - Class-level analysis, which involves analyzing a policy designed for a class of causal bandits (e.g., causal bandits with maximum degree $d$ and causal depth $L$, which is standard in causal bandit literature).
> > >
> > > Given this context, we address the regret of the example proposed by the reviewer in both settings:
> > >
> > > **Class-level regret for the class of graphs with maximum in-degree $d$ and causal depth $L$:** The UCB-1 algorithm can have a class-level regret that scales as $\tilde{\mathcal{O}} \big(2^{\frac{d^L}{2}} \sqrt{T}\big)$ (e.g., in the reverse tree graph). On the other hand, our class-regret scales as $\tilde{\mathcal{O}}\Big( d^{L - \frac{1}{2}} \sqrt{T}\Big)$.
> > >
> > > **Instance-dependent regret for the chain graph:** As the UCB-1 algorithm does not take the causal structure into account, its regret remains as $\tilde{\mathcal{O}}(2^{N/2}  \sqrt{T})$. On the other hand, we can readily analyze the instance-dependent regret of our algorithm, which scales as $(N-1)\sqrt{T}$. To get the instance-level regret of our algorithm, let's examine the source of the $d^{L-1/2}$ term in our regret:. if we define the maximum in-degree at each causal depth as $d_{(\ell)} = \max_{i\in [N], L_i=\ell} d_i$ for $\ell \in [L]$, then the $d^{L-1/2}$ term can be refined to the instance-dependent term $\sqrt{d} \prod_{\ell=1}^{N-1} d_{(\ell)}$ (in the proofs we need to replace $d$ with $d_{(\ell)}$). In the term $\sqrt{d} \prod_{\ell=1}^{N-1} d_{(\ell)}$, $\sqrt{d}$ reflects the complexity of the linear function, and the product accounts for the compounding effect where each causal depth contributes to $d_{(\ell)}$.

---

> ### Comment · Reviewer_2rLb · 2024-08-11
>
> Thanks for your response. based on the current result, the upper bound is $\mathcal{O} \left ( (N -1) ^{(N-3/2)}\sqrt{T} \right)$? If so, it seems that, compared to the classic UCB for this instance, the regret bound is not particularly advantageous. It may be beneficial to discuss the improved instance-dependent regret, as you mentioned, in the revised version.
>
> I intend to maintain my current score; however, I have concerns regarding soft interventions. In the title, abstract, and introduction, the authors claim to propose an algorithm for soft intervention cases. However, in the problem formulation, they define only a single soft intervention per node, which was unexpected.

---

> > ### Author Response · Authors · 2024-08-11
> >
> > As the reviewer points to, we have consistently mentioned that we focus on soft interventions. This is the most general form of intervention and subsumes hard and $do$ interventions.
> >
> > If the reviewer is asking about one versus multiple interventions per node, we re-emphasize that using one is standard in the causal bandit literature for notational simplicity. Extensions from one to multiple interventions is trivial. A constant 2 will be replaced by $K$ (number of interventions) in cardinality of the intervention space.
> >
> > We kindly ask the review to base their judgment on the theoretical contribution of this paper: this paper significantly extends the scope of the causal bandit literature by entirely removing the assumption about knowing the graph topology and general soft interventions. All our other assumptions are either in line with or more relaxed compared to those in the existing literature. We are glad that the reviewers have not pointed to any technical flaws in the analysis.

---

> ### Comment · Reviewer_2rLb · 2024-08-11
>
> Just for clarification regarding soft interventions. I expected that you define it in the following way. By soft intervention on node $i$, we can change any entries of $B_i$ to any values (with some restriction). But in your setting, by soft intervention, we can only change $B_i$ to $B^*_i$. Could you discuss the complexity of the former setting? Is it not learnable? I understand the theoretical contribution of the paper, and my judgment is based on that.

---

> > ### Author Response · Authors · 2024-08-11
> >
> > Thanks for giving us a chance to clarify this. Please note $B_i$ is *not* the interventional value of the random variable $X_i$ generated by node $i$. Rather, it specifies the mechanism for generating it.
> >
> > **Post-interventional random variables:** Specifically, upon intervening on node $i$, the random variable $X_i$ can take any arbitrary real value. Its value is not limited to only one or finite number of choices. Specifically, by applying a soft intervention on node $i$, we are changing its conditional distribution $\mathbb{P}(X_i \ | \  X_{{\sf pa}(i)})$ to a distinct conditional distribution. The post-intervention random variable $X_i$ will be generated according to the post-intervention conditional distribution. This is in contrast to $do$ interventions that specify a choice for $X_i$ (which essentially means placing the entire post-interventional conditional probability on a singular value).
> >
> > **Post-interventional SEMs:** Since we are working with linear SEMs, both pre- and post-interventional conditional distributions are fully specified by matrices $B$ and $B^*$ as well as the exogenous noise model. We are assuming that all these (i.e., $B$, $B^*$, and noise distribution) are fully *unknown*. This means that the pre- and post-interventional distributions are fully *unknown*. Our algorithm is learning all these, which includes learning  $B$ and $B^*$ (their support and their real-valued entries).  The complexity of learning all $2N$ such vectors $\{B_i,B^*_i:i\in[N]\} $is already included in the regret. We have considered one post-intervention SEM $B^*$. This can be readily extended to any finite number of post-intervention SEMs. Extensions to an infinite number of post-intervention models can be a potential extension. That necessitates learning an *infinite* number of interventional distributions, which to the best of our knowledge even for simpler settings (e.g., knowing the topology or even interventional distributions) is an open question.
> >
> > So, if we are understanding the reviewer’s concern correctly, having one single post-intervention $B_i^*$ does *not* mean that we are forcing $X_i$ to take only one value. Rather, we are changing the conditional distribution that generates $X_i$ and it still can take a real value generated by the interventional distribution (and this is a generalization of $do$ interventions that specify the $X_i$ to take a specific deterministic value – which in the context of causal bandits has a finite number of possibilities).

---

> > > ### Comment · Reviewer_2rLb · 2024-08-11
> > >
> > > Thanks for your reply. I understood the answer to my question in the following sentence: "Extensions to an infinite number of post-intervention models can be a potential extension. That necessitates learning an infinite number of interventional distributions, which to the best of our knowledge even for simpler settings (e.g., knowing the topology or even interventional distributions) is an open question.", Thanks for your clarification.

---

### Official Review · Reviewer_zJSz · 2024-07-12

**Soundness:** 2
**Presentation:** 3
**Contribution:** 3
**Rating:** 4
**Confidence:** 4

**Summary:**

This paper studies the linear scm setting for causal bandits. In particular, there are two vectors associated with the linear response of every node, and the learner may independently choose which of the two vectors to use. The value as well as the graph are unknown. Hence, the action space is 2**number_nodes. The authors provide an algorithm with a regret upper bound that nearly matches a lower bound, also presented.

**Strengths:**

The problem formulation is natural and seems to be “the right size;” it’s added generality compared to the previous state of the art is enough to be interested but still allow for a solvable problem. The presentation is generally clear and polished.

**Weaknesses:**

There are some technical concerns; please see the questions below.

**Questions:**

What does assuming that B_i and B_i^* have the same support buy you?

Is the Lasso calibrated for the max number of parents?

The bound you have on kappa in (27) is very big, O(m^2). Doesn’t this make Theorem 1, part 2 trivial?

Is assumption 5 necessary? Why can’t you argue that soft interventions that would break assumption 5 would have a minimal effect on the expected loss and therefore only contribute a small amount to the regret? One might even be able to have a bound that is adaptive to eta. Or do you need it for graph discovery?

What is c, or at least what does it depend on?

Why use ridge regression for estimating B, B*? They are assumed sparse, after all.

Why do you need to solve an optimization problem over 2^|A|? Won’t the solution be on a corner of the |A|-dimensional hypercube, so a continuous relaxation would suffice?

Can you provide some intuition for what the lower bound construction looks like?

**Limitations:**

No limitations discussed in the main body.

---

> ### Author Rebuttal · Authors · 2024-08-07
>
> **Support of weights matrices:** We clarify that $B_i$ and $B_i^*$ always have the same support under soft interventions. as such interventions only change the conditional distribution of node $i$ without affecting the topology. The causal topology remains intact, leading to $B_i$ and $B_i^*$ having the same support. As a side note, we also remark that under more restrictive types of interventions ($do$ and hard), an intervention on node $i$ removes its causal dependence on its parents, i.e., $pa(i)$. In such cases, the support of $B_i^*$ becomes a subset of $B_i$. However, even in such cases when the topology is unknown, assuming that the support of $B_i^*$ is a subset of $B_i$ does not provide any gain. Overall, since we are assuming soft interventions, $B_i$ and $B_i^*$ have the same support.
>
> **Lasso calibrated for the max number of parents:** Yes, the algorithm is designed for the entire class of causal models with maximum in-degree bound $d$. If more node-level information is available, the Lasso regression can be accordingly modified. We remark that the interest of the community working on causal bandits has been towards removing as much as possible assumptions about the graph structure and interventions. Our setting is in line with extending the state of the art by removing topology information.
>
> **Loose bounds on kappa:** We thank the reviewer for the sharp observation. The reviewer is right that this is not a tight bound. Nevertheless, we note that for achieving a tight regret, such a loose bound is inevitable, and further improving it compromises the regret. Here’s the reason: the regret is only a function of the topology information, and the topology should be learned only to the extent that it facilitates identifying the best intervention. In other words, learning the topology up to some level improves regret and, after that, starts compromising it since learning the topology requires collecting more samples, which implies less sample efficiency. Therefore, any algorithm that learns the topology accurately is over-learning, and our conjecture is that it will compromise the regret. Please note that the purpose of Theorem 1 is to provide the guarantees needed only for characterizing the regret, and it is not intended to claim that the topology is learned accurately.
>
> **Necessity of Assumption 5:** As the reviewer points out, this assumption is needed for graph recovery, and it is standard when graph skeletons is unknown [19,22,R1]. When a node acts as a confounder, intervening on that node might not impact certain intermediary nodes but still influences reward. Without Assumption 5, the confounder relationships cannot be identified, leading to an incorrect graph topology recovery. The reviewer’s point is correct: the interventions on some nodes might have minimal effect on the reward. However, when the topology is unknown, these nodes need to be identified and treated properly. The reason is that even a minimal effect on the regret can still accumulate linear regret if it is chosen linearly in $T$. So the algorithm needs to also properly identify these nodes to ensure they are selected sublinearly.
>
> [R1]: Elahi et. al. Partial Structure Discovery is Sufficient for No-regret Learning in Causal Bandits, ICML 2024 Workshop RLControl Theory.
>
> **Definition of $c$:** Thanks for the sharp observation. This constant is $c=\kappa$ with $\kappa$ is defined in (27). We recognize this notation is redundant and will change it to $\kappa$.
>
> **The weights matrices are assumed sparse:** Thank you for this question. We are not assuming sparsity for $B$ and $B^*$. The maximum in-degree $d$ can be arbitrarily large, i.e., it can be as large as  $N−1$ . This means the columns of $B$ and $B^*$ are $d$-sparse, where $d$ can be $N−1$. Nevertheless, the reviewer is correct that if we limit the scope of the problem to assume that $B$ and $B^*$ are sparse, then sparse-recovery methods can be more effective than ridge regression.
>
> **Continuous relaxation for optimization** We recognize that the reviewer might have an interesting point about solving a bandit combinatorial problem via continuous relaxation. We are unaware of such relaxation in the bandit literature. One challenge in adopting such an approach is that the utility to be optimized (i.e., UCB) does not take values at non-discrete values. Specifically, $UCB_{a}$ is not defined for $a_i\notin\{0,1\}$. As such, while we appreciate the reviewers' thoughts, we do not see an immediate way of using continuous relaxation in combinatorial bandit decisions.
>
> **Lower bound construction:** In this paper, we provide a minimax lower bound. This is done by constructing two bandit instances that are highly similar, and distinguishing them is difficult. Specifically, we construct two bandit instances that share the same hierarchical graph structure. These instances differ only in the parameters that switch the observational and interventional weights for the nodes with causal depth of 1, resulting in distinct optimal interventions. We identify the minimum samples that any algorithm will need to distinguish between the two bandit instances and the minimum regret it will incur on one of the instances. This serves as a lower bound on the minimax regret.
>
> **Final note about “soundness”:** We hope we have addressed the reviewer’s concerns about “soundness” (rate 2: fair) – especially why  $B_i$ and $B_i^{\star}$ have the same support, why the second part of Theorem 1 is sufficient for our regret minimization purposes, lack of sparsity in $B$ and $B^{\star}$, and continuous relaxation. We remark that this paper significantly extends the scope of the causal bandit literature by entirely removing the assumption about knowing the graph topology (a common assumption by extensive recent publications in JMLR, NeurIPS, ICML, and AISTATS). This is a major leap in causal bandits, and kindly request that the reviewer consider re-evaluating their rating of the manuscript.

---

> > ### Comment · Reviewer_zJSz · 2024-08-09
> >
> > Thanks for the detailed response, and I think many of my concerns are answered, concerning theorem 1. On a related note, how does the regret bound scale with m?

---

> > > ### Author Response · Authors · 2024-08-10
> > >
> > > **Scale with $m$:** We appreciate the reviewer for raising this important question. The regret bounds scale linearly in $m$. An intuitive reason is that if we multiply all the random variables of the system by a constant $M$ then the final reward will be scaled up with $M$ and the regret of any algorithm scales up with a constant $M$. As discussed at the end of Section 2, the dependence of the regret on $m$ is linear even in simpler linear bandit problems (e.g., [25]). An exact same linear dependence has been also reported in other causal bandit settings [14,17]. We note that in some literature such dependence might not appear explicitly since they normalize the range of the random variables or rewards to fall in the range [0,1], i.e., by setting $m=1$ (e.g., causal bandit with $do$ intervention [1,6] and linear bandits [R2]).
> > >
> > > [R2]: Abbasi-Yadkori et. al. ​​Improved algorithms for linear stochastic bandits, NeurIPS 2011
> > >
> > > **Other concerns:** We are glad that the reviewer’s concerns about Theorem 1 are addressed. We will be happy to also elaborate more on other concerns if there are still remaining ones.

---

> > > > ### Comment · Reviewer_zJSz · 2024-08-12
> > > > **Thanks for the response**
> > > >
> > > > Thanks, I agree that a linear scaling with m is not avoidable. I had a suspicion that it could have super-linear scaling, as the probability of error bounds degrade in m so I would expect some sample complexity terms to at least pick up a log(m) to compensate.

---

> > > > > ### Author Response · Authors · 2024-08-12
> > > > >
> > > > > Again, we thank the reviewer for the thoughtful point. We'll add a comment to the revised manuscript to emphasize linearity in $m$. We'll be also happy to address any remaining concerns.
> > > > >
> > > > > Based on the discussions, we'd be grateful if the reviewer considers re-evaluating their rating of the paper.

---

### Official Review · Reviewer_1kEj · 2024-07-14

**Soundness:** 3
**Presentation:** 2
**Contribution:** 4
**Rating:** 7
**Confidence:** 4

**Summary:**

The authors consider a stationary causal bandit problem for an unknown linear model with weight matrix $B \in \mathbb{R}^{N \times N}$ and noise vector $\epsilon \in \mathbb{R}^N$. In their setup, intervention on the node $X_i$ replaces all of the weights into $X_i$ with those from another unknown matrix $B^* \in \mathbb{R}^{N \times N}$.

The authors propose a two-phase approach: learning the causal structure by (1a) learning a valid topological ordering (1b) learning the causal DAG, and then (2) applying a phase elimination algorithm to learn the best pulls over a time horizon $T$.

The authors also include a minimax lower bound for their problem setup. For either bound, the authors argue that the maximum causal depth $L$ and a known upper bound (denoted here as $d^+$ for clarity) for the maximum in-degree $d$ are the relevant topological parameters.

It is assumed throughout that interventions do not affect the causal structure: $\mathrm{Supp}(B)=\mathrm{Supp}(B^*)$.

**Strengths:**

The authors identify a novel problem setup in the understudied field of causal bandits with soft interventions and unknown causal structure. Their results are significant to the causal bandit literature and have clear applications for real-world modelling.

**Weaknesses:**

The authors make an informal claim near the beginning of the paper, which I suspect affects the tightness of their bounds. They argue that all conditional independencies must be learned for regret minimisation with an unknown graph. However, it should be clear that only the nodes in $\mathrm{An}(Y)$ are relevant. In stage (1b), beginning with the reward node $N$ and iteratively learning the parent sets $\mathrm{pa}(N)$ then $\mathrm{pa}(\mathrm{pa}(N))$ should be a more efficient approach. This suggests to me that the relevant causal path depth in their bounds is actually $L_N$, while the relevant in-degrees in stages (1) and (2) of their algorithm are $d^+$ and $d^* := \mathrm{max}_{i \in an(N) \cup N} (d_i)$ respectively.

The authors also seem to ignore the parameter $R := (m/\eta)^2$ in their regret bounds. While it is not a topological property of the population DAG, it is certainly an important parameter in the problem setting and could be thought of as a kind of guaranteed minimal signal-to-noise ratio (SNR). From the definition of $T_1$ there should be a linear scaling of the regret bounds with $R$, up to a poly-log factor. This SNR is important for real-world modelling and could also offer insights into relaxing assumptions 3, 4 and/or 5.

Putting the above together, my estimation of the same upper regret bound is $\tilde{O}((cd^*)^{L_N - 1/2} \sqrt{T} + R + d^+ + N)$. Please correct me otherwise and/or update the manuscript. I will revise my score and evaluation of the paper's soundness following the authors' rebuttal.

The authors provide a very minimal experiment, relegated to the appendix. It would have been informative to see parameter sweeps and experiments with random DAGs to compare their bounds with empirical scalings.

The authors should make no mention of the possibility of hidden confounders and whether their models can incorporate this. In particular, the claim that only $\mathrm{pa}(N)$ are possibly optimal with hard interventions is false - see [1] for details.

[1] Sanghack Lee and Elias Bareinboim, *Structural Causal Bandits: Where to Intervene?*, NeurIPS 2018.

**Questions:**

Does your approach assume causal sufficiency? In particular, are noise terms assumed to be mutually independent (i.e., $\mathrm{Corr}(\epsilon_i, \epsilon_j) = \delta_{ij}$)?

I do not understand the first case in for the estimator for the ancestors of $i$ in equation (9): it appears to read "if $i$ is not the reward node but has decedents then it can't have any ancestors". Is this a typo?

In table 1, the meaning of $K$ in the benchmark approaches should be indicated. Is this the number of possible hard interventions on a node for categorical data or something else?

**Limitations:**

Yes, the authors provide some key underlying assumptions. Whether causal sufficiency is assumed should be indicated either way.

---

> ### Author Rebuttal · Authors · 2024-08-06
>
> We would like to thank the reviewer for the thoughtful questions, especially those pertinent to the instance-dependant regret and hidden confounders. We provide more discussions and we hope these clarify the reviewer’s technical concerns.
>
> **Dependence on $d$ and $L$:** The reviewer raises a good point about also characterizing instance-dependent regret bounds. This falls into the general dichotomy of class-level versus instance-level regret analysis. We have provided regret bounds for a class of bandits with a maximum in-degree $d$ and maximum causal length $L$. As the reviewer correctly points out, this can be further fine-tuned to recover instance-dependent regret bounds that use the instance-level information $L_N$ and $d^*$. Characterizing these bounds is verbatim similar to our analysis, and we will add a remark in the paper to state these instance-dependant regret bounds.
>
> The changes in the instance-dependent bounds can be done as follows:
>
> - **The effect of $L$ and $L_N$.** We note that neither Algorithm 1 nor Algorithm 2 requires the information of $L$. Algorithm 1 implicitly learns $L$ by learning the parent sets, which is then fed to Algorithm 2. For the regret bound proof, we use recursion along the longest causal path. Each layer in the causal path contributes a term $cd$ to the regret, the compound effect of all layers along the causal path becomes $(cd)^L$ in the class-level regret. For the instance-dependent regret on the reward node, it is sufficient to perform this process along the causal path on the subgraph that only contains $an(N)$, in which case the regret will depend on $L_N$. This subgraph is learned by Algorithm 1.
>
> - **The effect of $d$ and $d^{\star}$.** Similar to the previous point, when we evaluate the regret compounding along the causal paths formed by $an(N)$, the maximum in-degree will be $d^*$. Thus, the contribution of each layer to the regret will $cd^*$ instead of $cd$.
>
> - Algorithm 1 requires knowing $d$, and the iterative approach suggested by the reviewer can save some computational cost by applying Lasso estimators only on $an(i)$. But this approach does not tighten the regret since the same amount of observational data $T_2$ is needed to learn any parent sets.
>
>  **Parameter $R=(m / \eta)^2$:** We certainly agree with explicitly incorporating the effect of $R$ on the regret bounds. We agree with the reviewer’s point that introducing this notion of SNR allows for replacing/relaxing Assumptions 3-5 could be restated equivalently as an Assumption on the SNR term $R$.  We further elaborate on the dependence on $R$ in the answer to the next comment.
>
> We also note our objective has been primarily focused on the fundamental question of how the regret depends on (i) the graph topology and (ii) interventional distribution. Causal bandit literature has been evolving towards removing as many assumptions as possible about graph topology and interventional distribution. This is the first paper that has dropped both under general structures and interventional models (with the remaining piece being extending linear SEMs to non-linear SEMs). However, we certainly agree that it is also important to understand the dependence of regret on other model parameters.
>
>
> **Upper regret bound:** Based on the previous point, the correct instance-dependant regret bound is $\tilde{\mathcal{O}}((c d^*)^{L_N - 1/2} \sqrt{T} + d^{+} + RN)$. The term $RN$ term arises from the step of identifying the ancestor relationship. We note that a similar term appears in related literature [19,22], where the term $R$ is coupled with the degree as they only need to identify $pa(N)$.
>
> **Additional  experiment:** Please refer to the global response where we provide more experiments.
>
> **Hidden confounders:** Thank you for the comment. We note that all our statements are in the context of our specified setting (no hidden confounders). Under this setting (i.e., no hidden confounders and full intervention capability), only $pa(𝑁)$ are optimal. We will re-emphasize this as a remark in the paper to avoid confusion.
>
> We highlight that having no hidden confounders is a standard assumption in the existing literature on causal bandits (even under more restrictive $do$ interventions).  The primary reason is that causal bandits are not fully understood even when there are no hidden confounders. Nevertheless, we certainly agree that understanding the effect of unobservable confounders is also important. Including hidden confounders will introduce non-trivial challenges. For instance, it brings in bi-direction edges, as a result of which one cannot express the reward as a linear function of observable variables.
>
> **Causal sufficiency:** We thank the reviewer for this question. Yes, our approach does assume causal sufficiency, and we will include this in Assumption 4.
>
>  **first case in equation (9):** Thanks for the sharp observation. Yes, based on Assumption 5, indeed the condition part "if $\text { if } i<N \text { and }|\widehat{\operatorname{de}}(i)|>0$ Is a typo and should be removed.
>
> **$K$ in table 1:** Yes, $K$ represents the number of possible $do$ interventions on one node. We will add a comment to specify it.
>
> **Final note about “soundness”:** We hope we have addressed the reviewer’s concerns about “soundness” (rate 2: fair) – especially about having hidden confounders and class-level versus instance-level regret (which can be readily recovered from our result). We remark that this paper significantly extends the scope of the causal bandit literature by entirely removing the assumption about knowing the graph topology (a common assumption by extensive recent publications in JMLR, NeurIPS, ICML, and AISTATS). This is a major leap in causal bandits, and kindly request that the reviewer consider re-evaluating their rating of the manuscript.

---

> > ### Comment · Reviewer_1kEj · 2024-08-12
> >
> > Many thanks to the authors for their thoughtful and detailed responses, which have addressed my major concerns and questions.
> >
> > In light of this, I will upgrade my overall rating to a 7: Accept, and my soundness rating to a 3: Good. My high overall score is based off the significance and novelty of the authors' contribution.
> >
> > I suspect correlated noise variables (i.e., latent confounders) could be incorporated into the authors' analysis, which presents one exciting avenue for future work.

---

> > > ### Author Response · Authors · 2024-08-12
> > >
> > > We are grateful to the reviewer for the great suggestion. Indeed -- we agree that modeling the latent confounders via modifying the noise model is an effective of integrating them into the causal bandit model. We certainly agree that is an important next line of research in the causal bandit literature.

---

### Author Rebuttal · Authors · 2024-08-07

We thank the reviewers about the comments about our assumptions and empirical evaluations. We would like to clarify the following:

**Theoretical contributions:**  We remark that this paper significantly extends the scope of the causal bandit literature by entirely removing the assumption about knowing the graph topology and general soft interventions. All our other assumptions are either in line with or more relaxed compared to those in the existing literature. We have provided more details on these. We are glad that the reviewers have not pointed to any technical flaws in the analysis. We believe this paper provides a  major contribution in the trajectory of development in causal bandits, which has been moving away from stylized assumptions. We certainly agree that there are still important issues to address (e.g., hidden confounders) that remain for future work.

**Additional empirical evaluations:** With the additional results, overall we have provided evaluations to assess the key theoretical findings of the paper. These include:
Scaling behavior with respect to the three key parameters in the results, i.e., $d, L$.
Sublinear regret and fast convergence.
Computational advantage, which is a bottleneck for almost all the existing approaches to causal bandits with even stronger assumptions.

We remark that due to computational challenges, causal bandit algorithms are generally focused on small graph sizes. Additionally, the hierarchical graph is the one that can mimic the worst-case regret for algorithms with given parameters. The hierarchical graphical models that we are evaluating have $L=9$ layers and $N=19$ nodes, which is the largest model considered in the literature (for instance the JMLR paper [14], the AISTATS paper [17], and the JSAIT paper [24] consider hierarchical with no more than 3 layers; JMLR [14] and JSAIT [24] also consider simpler graph with at most $N=10$ nodes; and [22], which focuses on $do$ intervention with unknown graph, provides experiments for a simple binary tree with $N = 20$ nodes.)

**Scaling with $L$:** Scaling of the regret with respect to the causal depth $L$ is depicted in Figure 1 in the attached document for setting of a hierarchical graph with $d=2$ and $L$ varies in the range $1$ to $9$. The theoretical results (regret upper and lower bounds) predict that the regret grows at the range $(2c)^L$ (i.e., exponential in $L$). The empirical results in Figure 1 corroborate that the cumulative regret scales exponentially with length $L$, and the actual regret closely tracks the upper bound’s trend.

**Scaling with $d$:** Scaling of the regret with respect to the maximum in-degree $d$ is depicted in Figure 2 in the attached document based on a hierarchical graph with $L=2$. We note that we increase the number of sufficient exploration parameters $T_1$ and $T_2$ to ensure the results for all the degrees. Theoretical predictions suggest that our algorithm’s regret scales as $d^{3/2}$ (i.e., polynomial growth in $d$). For the choice of $L$, the lower bound becomes linear in $L$. Figure 2 demonstrates that our regret is super-linear and tracks the polynomial trend of the regret upper bound (i.e., our achievable regret).

**Sublinear regret:** Figure 3 of the attached document shows the cumulative regret for a larger graph with $L=6$ and $d=2$. In this scenario, both GA-LCB with and without graph information exhibit sublinear regret. However, the GA-LCB with unknown graph information incurs higher cumulative regret due to the need for additional exploration required for estimating the topology.

**Computational cost:** As discussed in Section 3.3, a key contribution of our algorithm is its significant computational advantage compared to the existing causal bandit algorithms (which also rely on stronger assumptions). First, our algorithm circumvents the computational complexity of computing the upper confidence bounds (which generally requires solving an optimization problem) by iteratively computing the value through the causal depth. To highlight the advantage, we note that the UCB-based causal bandit algorithms are computationally viable for only $L\leq 2$ or for special graphs such as chain graphs [24]. In contrast, the computational complexity of our algorithm scales linearly with $L$ (especially scales as with $\mathcal{O}(Ld^3)$). This has allowed us to easily implement algorithms for causal paths as long as $L=9$. Additionally, our algorithm includes a phase elimination step that removes sub-optimal arms from the potential best intervention set. Since we need to calculate UCBs for each possible intervention, the phase elimination step prevents unnecessary UCB computation. This results in significant computational savings as the time horizon increases.
We have also provided more empirical analysis of the computational complexity in Appendix A. Figure 4 (Appendix A) illustrates the computational differences between various algorithms. It is evident that our algorithm significantly reduces computation time, even for small graphs. In contrast, the other algorithm is not scalable with respect to both degree $d$ and length $L$.

---

### Decision · Program_Chairs · 2024-09-25

**Decision:**

Accept (poster)

**Comment:**

The paper studies causal bandit with unknown causal graph structure and soft interventions. The reviewers acknowledge that the paper addresses an important problem extension in causal bandits with significant results. The issues addressed by the reviewers are mostly addressed by the authors during the rebuttal stage. Overall, the paper provides solid contributions to the study of causal bandits, and I recommend accepting the paper.